# Transcription-coupled repair of DNA–protein cross-links depends on CSA and CSB

Christopher J. Carnie [1,2,8] ✉, Aleida C. Acampora[3,8], Aldo S. Bader [1,2], Chimeg Erdenebat[3], Shubo Zhao[3], Elnatan Bitensky[4], Diana van den Heuvel[5], Avital Parnas[4], Vipul Gupta[2], Giuseppina D'Alessandro [1,2], Matylda Sczaniecka-Clift[2], Pedro Weickert[3], Fatih Aygenli[6], Maximilian J. Götz[3], Jacqueline Cordes[3], Isabel Esain-Garcia[1,7], Larry Melidis [1,7], Annelotte P. Wondergem[5], Simon Lam [1,2], Maria S. Robles[6], Shankar Balasubramanian[1,7], Sheera Adar[4], Martijn S. Luijsterburg [5], Stephen P. Jackson [1,2] ✉ & Julian Stingele [3] ✉

Covalent DNA–protein cross-links (DPCs) are toxic DNA lesions that block replication and require repair by multiple pathways. Whether transcription blockage contributes to the toxicity of DPCs and how cells respond when RNA polymerases stall at DPCs is unknown. Here we find that DPC formation arrests transcription and induces ubiquitylation and degradation of RNA polymerase II. Using genetic screens and a method for the genome-wide mapping of DNA–protein adducts, DPC sequencing, we discover that Cockayne syndrome (CS) proteins CSB and CSA provide resistance to DPC-inducing agents by promoting DPC repair in actively transcribed genes. Consequently, CSB- or CSA-deficient cells fail to efficiently restart transcription after induction of DPCs. In contrast, nucleotide excision repair factors that act downstream of CSB and CSA at ultraviolet light-induced DNA lesions are dispensable. Our study describes a transcription-coupled DPC repair pathway and suggests that defects in this pathway may contribute to the unique neurological features of CS.

Covalent DNA–protein cross-links (DPCs) are bulky DNA adducts that are highly toxic to cells because they can impede DNA replication[1–4]. DPCs can arise by 'trapping' of enzymatic reaction intermediates such as topoisomerase cleavage complexes or DNA methyltransferase 1 (DNMT1) that can be stabilized by topoisomerase poisons or the chemotherapeutic drug 5-aza-dC (5-aza-2′-deoxycytidine), respectively[5–7]. Non-enzymatic DPCs can be induced by platinum-based drugs and reactive endogenous metabolites such as formaldehyde[6,8].

DPC repair requires DPC proteolysis by specialized proteases (such as SPRTN in higher eukaryotes) or the proteasome[9–13], and its

[1]Cancer Research UK Cambridge Institute, University of Cambridge, Cambridge, UK. [2]The Gurdon Institute and Department of Biochemistry, University of Cambridge, Cambridge, UK. [3]Gene Center and Department of Biochemistry, Ludwig-Maximilians-Universität München, Munich, Germany. [4]Department of Microbiology and Molecular Genetics, The Institute for Medical Research Israel–Canada, The Faculty of Medicine, The Hebrew University of Jerusalem, Jerusalem, Israel. [5]Department of Human Genetics, Leiden University Medical Center, Leiden, the Netherlands. [6]Institute of Medical Psychology and Biomedical Center, Faculty of Medicine, Ludwig-Maximilians-Universität München, Munich, Germany. [7]Yusuf Hamied Department of Chemistry, University of Cambridge, Cambridge, UK. [8]These authors contributed equally: Christopher J. Carnie, Aleida C. Acampora. ✉e-mail: chris.carnie@cruk.cam.ac.uk; steve.jackson@cruk.cam.ac.uk; stingele@genzentrum.lmu.de

importance is highlighted by hypomorphic *SPRTN* mutations causing premature ageing and cancer predisposition in patients with Ruijs–Aalfs syndrome[14–17]. DPC proteolysis can be initiated upon collision with the DNA replication machinery[18] causing SPRTN activation[13,19] and replication-coupled ubiquitylation, targeting DPCs for proteasomal degradation[13,20]. Additionally[3,10–12], global-genome (GG) DPC repair occurs through DPC SUMOylation and subsequent SUMO-dependent polyubiquitylation by RNF4 or TOPORS, which triggers proteasomal degradation[21–25] or cleavage by SPRTN[26].

DPCs stall T7 bacteriophage RNA polymerase in vitro[27] but it is unknown whether DPCs affect transcription in mammalian cells. At ultraviolet (UV) light-induced DNA lesions, RNA polymerase II (RNAPII) stalling activates transcription-coupled nucleotide excision repair (TC-NER)[28,29], leading to RNAPII degradation, local and global shut-down of transcription and lesion excision by the nucleotide excision repair (NER) machinery[30,31]. TC-NER is initiated by CSB, which recognizes stalled RNAPII and recruits the CRL4–CSA E3 ubiquitin ligase complex[30] that, with ELOF1 (refs. 32–34), promotes ubiquitylation of RPB1 (refs. 31,35), the largest subunit of RNAPII. This stabilizes mono-ubiquitylated UVSSA, which recruits the TFIIH complex and XPA, subsequently engaging the ERCC1–XPF and XPG endonucleases[30]. These mediate a dual incision, excising a stretch of single-stranded DNA. The resultant gap is then closed by DNA synthesis and ensuing ligation[30]. Additionally, UV lesions can be repaired by GG-NER, which is initiated by lesion sensors such as XPC, leading to recruitment of TFIIH, XPA and the incision endonucleases as per TC-NER[30,36,37].

Several human diseases are caused by loss-of-function mutations in NER genes. Mutations in *ERCC8* (encoding CSA) or *ERCC6* (encoding CSB) cause Cockayne syndrome (CS), *UVSSA* mutations cause UV-sensitive syndrome and mutations in any of the *XP* genes cause xeroderma pigmentosum[30]. CS patients suffer from cachexia, neurodegeneration and kidney failure[38–40]—features not shared with xeroderma pigmentosum, which is characterized by extreme sensitivity to UV light and an increased risk of skin cancer[41]. This contrast implies that the aetiological lesions targeted by CSA and CSB extend beyond 'classical' NER substrates. Indeed, endogenous formaldehyde induces transcription stress and drives CS phenotypes in mice and could contribute to CS aetiology[42].

In this Article, we identify a transcription-coupled DPC repair pathway dependent on CSB and CSA but not on canonical NER. We demonstrate that CSA and CSB are important for DPC repair in actively transcribed genes and are thus critical for transcription recovery following DPC induction and cellular tolerance of DPCs. These results suggest that loss of DPC repair capacity caused by mutations in CSB/*ERCC6* or CSA/*ERCC8* may contribute to the unique pathological features of CS.

## Results

### CS proteins are required for DPC tolerance
To identify factors involved in DPC repair, we conducted genome-scale clustered regulary interspaced short palindromic repeats interference (CRISPRi) screens in K562 cells in the presence or absence of formaldehyde. To help distinguish DPC-specific effects from those caused by other formaldehyde-induced lesions, we screened in parallel with 5-aza-dC, which specifically cross-links DNMT1 to DNA (Fig. 1a). Using a false discovery rate (FDR) cut-off of 0.1, we identified 93 protein-coding genes whose downregulation conferred sensitivity to formaldehyde and 17 that conferred resistance (Fig. 1b, Extended Data Fig. 1a and Supplementary Table 1). As observed in other formaldehyde-based CRISPR screens[43–45], interfering with expression of *ADH5* or *ESD* (encoding formaldehyde-detoxifying enzymes[45,46]; Extended Data Fig. 1b), resulted in severe formaldehyde sensitivity (Fig. 1b). The 5-aza-dC screen revealed 177 protein-coding genes whose downregulation conferred sensitivity and 51 that conferred resistance (Fig. 1c, Extended Data Fig. 1a and Supplementary Table 2). Repressed expression of *DCK*, *CMPK1* or *SLC29A1* conferred 5-aza-dC resistance (Fig. 1c) as

expected, because *SLC29A1* encodes a nucleoside transporter required for 5-aza-dC uptake, while DCK and CMPK1 phosphorylate 5-aza-dC, required for its incorporation into DNA (Extended Data Fig. 1c)[47–49].

As reported previously[42–44,46], downregulation of TC-NER factors—CSB, CSA, XPA, XPF and XPG—caused formaldehyde sensitivity (Fig. 1b). Strikingly, however, only CSA or CSB loss conferred 5-aza-dC hypersensitivity, while depletion of other TC-NER factors showed no effect (Fig. 1c). These data suggested that CSA and CSB may function in DPC repair independently of canonical TC-NER. To confirm these results, we performed clonogenic survival assays in wild-type (WT), *XPA^{-/-}* and *CSB^{-/-}* diploidized HAP1 cells. *CSB^{-/-}* and *XPA^{-/-}* cells were hypersensitive to formaldehyde, but only *CSB^{-/-}* cells were hypersensitive to 5-aza-dC (Fig. 1d,e). Furthermore, doxycycline-induced expression of CSB in *CSB^{-/-}* cells restored their resistance to formaldehyde, 5-aza-dC and illudin S (Fig. 1f–i and for non-induced controls, see Extended Data Fig. 1d–f), an alkylating agent that causes DNA lesions specifically repaired by TC-NER[50]. In contrast, inducing the expression of a CSB ATPase-dead mutant (CSB^{K538R})[51] in *CSB^{-/-}* cells failed to restore formaldehyde, 5-aza-dC or illudin S tolerance to the same extent as WT (CSB^{WT}; Fig. 1f–i and for non-induced controls, see Extended Data Fig. 1d–f). In line with our findings in HAP1 cells, the patient-derived CSB-deficient fibroblast cell line CS1AN was more sensitive to formaldehyde and 5-aza-dC than control fibroblasts (MRC5), as were *CSB^{-/-}* immortalized human hTERT–RPE1 cells (Extended Data Fig. 1g–j).

The absence of 5-aza-dC hypersensitivity in *XPA^{-/-}* cells suggested that downstream NER factors are not required for cellular DPC tolerance (Fig. 1e). Furthermore, in RPE1 cells, XPC loss, which compromises transcription-independent GG-NER, did not sensitize cells to formaldehyde or 5-aza-dC, even in the absence of CSB (Extended Data Fig. 1i–k and ref. 42), despite conferring UVC hypersensitivity (Extended Data Fig. 1l). Accordingly with our other findings, unlike after UVC radiation, we did not observe excised DNA oligonucleotides after formaldehyde treatment (Extended Data Fig. 1m–n). Even complete inactivation of NER by *XPA* knockout in *XPC^{-/-}* cells caused hypersensitivity to formaldehyde and illudin S but not 5-aza-dC (Fig. 1j–l)[37,39].

Taken together, these data demonstrate that CSB plays an important role in DPC tolerance that is distinct from its well-established role in TC-NER, which is not required for DPC tolerance.

### CSB acts in parallel to established DPC repair pathways
To elucidate the role of CSB in DPC tolerance, we explored its relationship with established DPC repair mechanisms. We edited endogenous *SPRTN* in WT and *CSB^{-/-}* RPE1 cells to generate patient-mimicking *SPRTN-ΔC* alleles that compromise the repair capacity of SPRTN[26] (Extended Data Fig. 2a). Combined loss of CSB and SPRTN activity caused proliferation defects and heightened formaldehyde sensitivity compared with *SPRTN-ΔC*, *CSB^{-/-}* or WT cells (Fig. 2a and Extended Data Fig. 2b). Concordantly, small interfering RNA (siRNA)-mediated CSB depletion increased formaldehyde sensitivity in both WT and *SPRTN-ΔC* cells (Fig. 2b and Extended Data Fig. 2c). Similarly, siRNA-mediated SPRTN depletion further sensitized both WT and *CSB^{-/-}* cells to formaldehyde (Fig. 2c and Extended Data Fig. 2d). Additionally, siRNA-mediated RNF4 depletion—compromising GG DPC repair[21,23]—from *CSB^{-/-}* RPE1 or HAP1 cells caused additional sensitivity to both formaldehyde and 5-aza-dC (Fig. 2d–g and Extended Data Fig. 2e,f). These findings demonstrate that CSB confers DPC tolerance through a pathway independent of established SPRTN- and RNF4-dependent repair mechanisms.

### CSB promotes recovery from DPC-induced transcription arrest
Since CSB acts as a sensor of stalled RNAPII for TC-NER, we explored whether DPC formation triggers CSB recruitment to RNAPII. Immunoprecipitation of elongating RNAPII (RPB1 CTD-pS2) complexes[52] following formaldehyde treatment revealed clear co-precipitation of both CSB and CSA (Fig. 3a). Furthermore, formaldehyde-induced

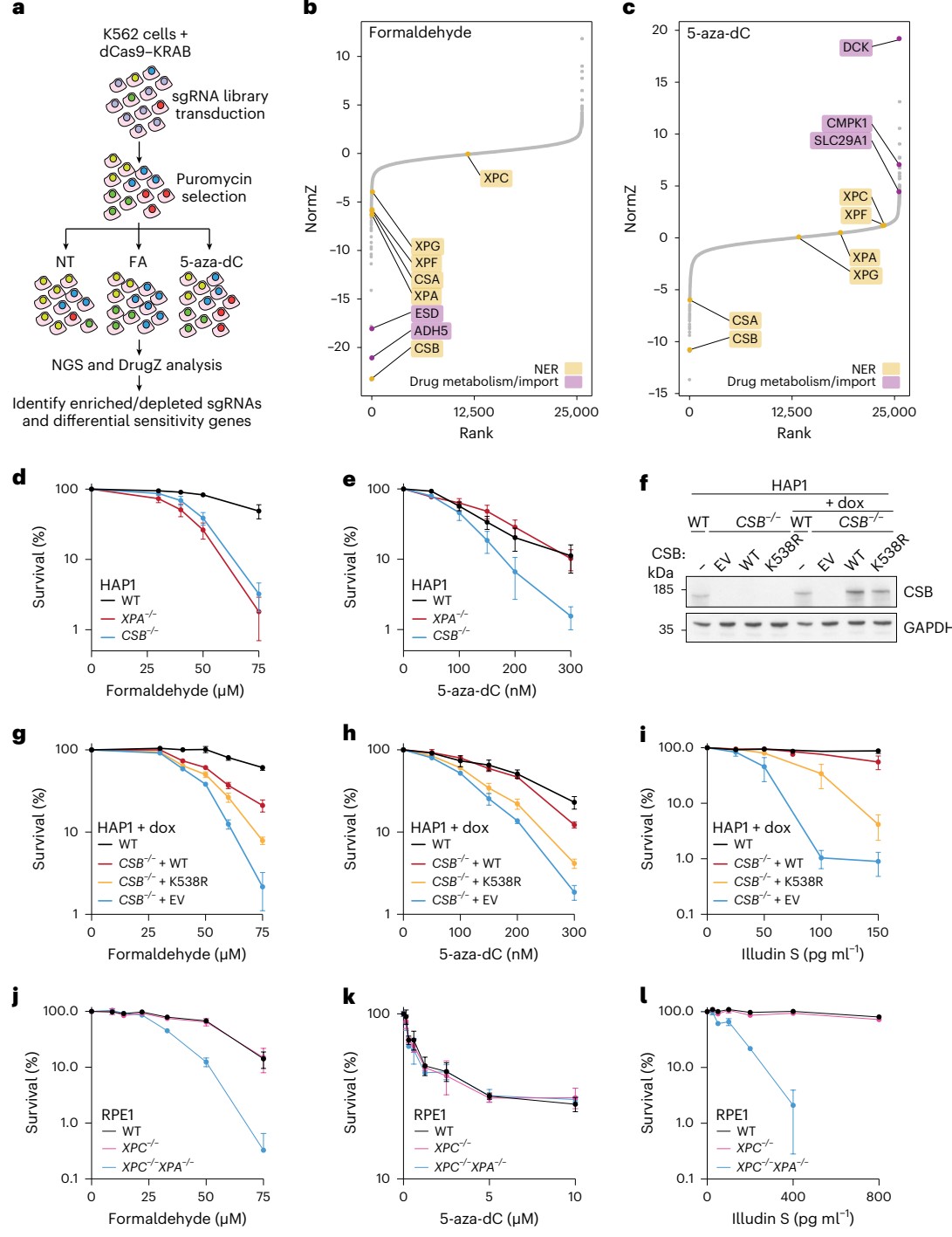

**Fig. 1 | *ERCC6*/CSB mediates cellular tolerance of DPCs. a**, Schematic of formaldehyde (FA) and 5-aza-dC CRISPRi screens compared to non-treated conditions (NT) in K562 cells. **b**, Rank plot showing normalized Z-scores (NormZ) scores from DrugZ analysis and selected hits from the formaldehyde CRISPRi screen in **a**. **c**, The same as for **b** but for the 5-aza-dC CRISPRi screen. **d**,**e**, Clonogenic survival assays in WT, *CSB*⁻/⁻ and *XPA*⁻/⁻ HAP1 cells treated with formaldehyde (**d**) or 5-aza-dC (**e**). Error bars ± s.e.m., *n* = 4 replicates. **f**, Doxycycline (dox)-inducible expression of CSB or CSB^K538R in *CSB*⁻/⁻ TET3G

HAP1 cells; representative of three independent experiments. **g**–**i**, Clonogenic survival assays in WT or *CSB*⁻/⁻ TET3G cells with doxycycline-induced expression of CSB, CSB^K538R or the empty vector (EV) treated with formaldehyde (**g**), 5-aza-dC (**h**) or illudin S (**i**). Symbols and error bars denote mean ± s.e.m., *n* = 3 replicates. **j**–**l**, Alamar blue cell viability assays in the indicated RPE1 cell lines treated with formaldehyde (**j**), 5-aza-dC (**k**) or illudin S (**l**). Symbols and error bars denote mean ± s.d., *n* = 3 replicates. Source numerical data and unprocessed blots are available in Source data.

co-immunoprecipitation of CSA with RNAPII was abrogated in *CSB*⁻/⁻ cells. To test whether CSA recruitment by CSB results in polyubiquityla-tion of the RNAPII subunit RPB1 (refs. 31,35), we enriched ubiquitylated proteins using the ubiquitin-binding protein Dsk2 (ref. 53) followed by western blotting against RPB1. As recently reported[42], we observed RPB1

polyubiquitylation after formaldehyde treatment in G1-synchronized RPE1 cells (Fig. 3b); this was partially CSB dependent (Fig. 3c). On treat-ing S-phase RPE1 cells with 5-aza-dC, however, only weakly polyubiq-uitylated RPB1 was detectable and this was not appreciably increased compared with S-phase cells treated with deoxycytidine (Fig. 3b).

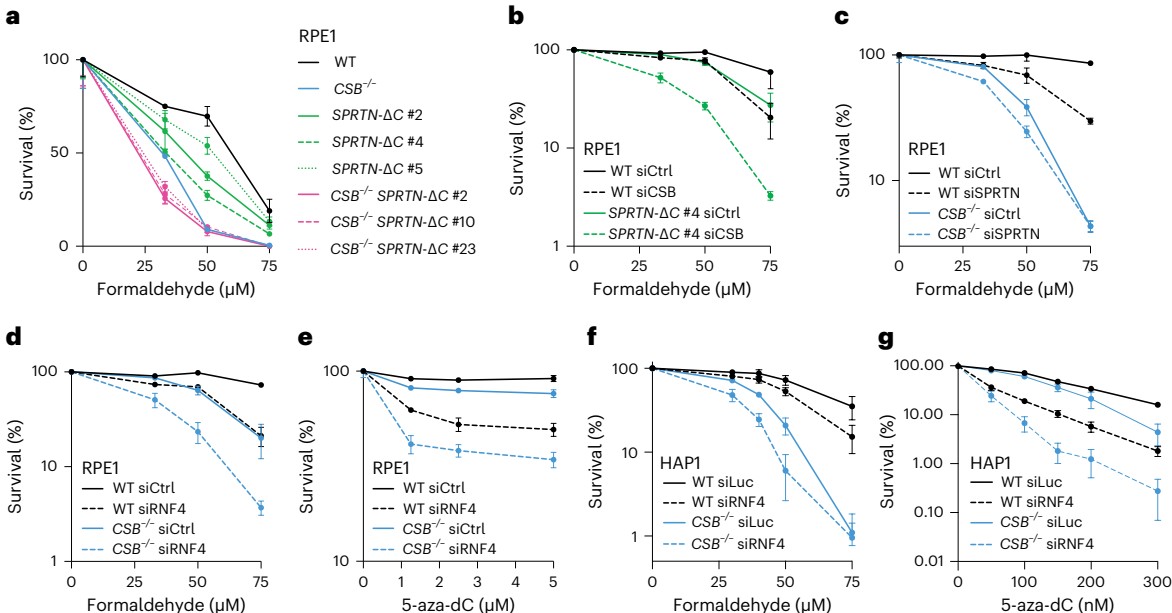

**Fig. 2 | CSB acts in parallel to known DPC repair pathways in promoting DPC tolerance. a**, Alamar blue cell viability assays with the indicated CSB- and SPRTN-deficient RPE1 cell line clones treated with formaldehyde. **b,c**, Alamar blue assays with the indicated cell lines treated with formaldehyde after siCSB or a non-targeting siRNA control (siCtrl) in WT versus *SPRTN-ΔC* cells (**b**) or siSPRTN in WT versus *CSB⁻/⁻* RPE1 cells (**c**). **d,e**, Alamar blue assays in the WT versus *CSB⁻/⁻* RPE1 cells upon RNF4 depletion, treated with formaldehyde (**d**) or 5-aza-dC (**e**). In **a–e**, symbols and error bars denote mean ± s.d., *n* = 3 replicates. **f,g**, Clonogenic survival assays in WT or *CSB⁻/⁻* HAP1 cells upon siRNA-mediated depletion of RNF4, treated with formaldehyde (**f**) or 5-aza-dC (**g**). Symbols and error bars denote mean ± s.e.m., *n* = 3 replicates. Source numerical data are available in Source data.

We speculated that the amount of 5-aza-dC-induced DNMT1 DPCs, which form at methylated CpG sites[54–56], may be too low in actively transcribed genes to result in substantial RNAPII ubiquitylation. Indeed, constitutively overexpressed enhanced green fluorescent protein (eGFP)-tagged DNMT1 (GFP–DNMT1) in U2OS cells (Extended Data Fig. 3a) resulted in observable 5-aza-dC-induced RPB1 polyubiquitylation (Fig. 3d). Furthermore, we observed close proximity between GFP–DNMT1 and RPB1 CTD-pS2 upon 5-aza-dC treatment measured by a proximity ligation assay (PLA) (Extended Data Fig. 3b,c), probably corresponding to collisions between elongating RNAPII and GFP–DNMT1–DPCs.

UV-induced RPB1 polyubiquitylation is linked to active TC-NER and also to RPB1 proteasomal degradation, mediated via Cullin-dependent E3 ubiquitin ligases including CSA[31–33,35,57]. Therefore, we tested whether formaldehyde-induced RPB1 polyubiquitylation also triggers its degradation (in the presence of cycloheximide to block protein synthesis). RPB1 levels were assessed using an antibody against RPB1 CTD-pS5, a modification enriched at transcription start sites (TSSs) but also found on elongating RNAPII[58]. Following formaldehyde treatment, RPB1 was mildly destabilized; this degradation was abrogated by inhibition of Cullin-dependent ubiquitylation with MLN4924 (Fig. 3e and Extended Data Fig. 3d), as observed after UVC (Fig. 3f and Extended Data Fig. 3e). However, while UVC-induced RPB1 degradation was entirely CSB and CSA dependent (Extended Data Fig. 3f–i), we only observed a partial reduction in formaldehyde-induced RPB1 degradation in *CSB⁻/⁻* or *CSA⁻/⁻* cells compared with WT controls (Fig. 3g–j and additionally observed using an anti-RPB1 CTD-pS2 antibody in Extended Data Fig. 3j–m). These findings are consistent with the diminished RPB1 polyubiquitylation observed following formaldehyde treatment of *CSB⁻/⁻* cells (Fig. 3c).

Collectively, our results suggested an important role for CSB in resolving acute DPC-induced transcription stress. To better understand the effects of DPCs on transcription in mammalian cells, we determined transcription rates by measuring incorporation of 5-ethynyluridine (EU) into nascent RNA[59] upon formaldehyde treatment. RNA synthesis

was inhibited in a dose-dependent manner, similarly to inhibition of DNA synthesis as measured by EdU incorporation (Extended Data Fig. 4a–c). We did not detect global effects of 5-aza-dC on transcription rates in either RPE1 or HAP1 cells (Extended Data Fig. 4d,e). Importantly, following formaldehyde removal, RNA synthesis recovered over 16 h in WT cells (Extended Data Fig. 4f,g), as observed after UVC irradiation (Extended Data Fig. 4h). Strikingly, recovery of transcription after formaldehyde treatment was markedly delayed in *CSB⁻/⁻* RPE1 and HAP1 cells (Fig. 4a and Extended Data Fig. 5a,b). Recovery was delayed in nucleolar and nucleoplasmic regions (Fig. 4a–d), suggesting that CSB supports recovery of RNA polymerase I (RNAPI)- and RNAPII-dependent transcription after DPC induction. Furthermore, *CSB⁻/⁻* RPE1 cells expressing GFP–CSB^K538R exhibited a partial transcription recovery defect following formaldehyde treatment when compared with *CSB⁻/⁻* cells expressing GFP–CSB^WT (Extended Data Fig. 5c,d).

To corroborate the role of CSB in transcription recovery after formaldehyde treatment, we performed quantitative PCR with reverse transcription (RT–qPCR) against various transcripts using an external chicken messenger RNA spike-in control to enable absolute quantification. Compared with WT cells, *CSB⁻/⁻* cells exhibited delayed expression recovery across all tested transcripts (Fig. 4e), suggesting that CSB loss compromises the production of mature transcripts following DPC induction. We next assessed whether these defects were affected by a prolonged block on new initiation in *CSB⁻/⁻* cells. We observed delayed reappearance of initiating, hypophosphorylated RPB1 (Fig. 4f), and also prolonged expression of the damage-induced inhibitor of transcription initiation ATF3 in *CSB⁻/⁻* cells (Fig. 4g).

To better understand the cause of formaldehyde-induced transcription arrest, we performed genome-wide sequencing of nascent transcripts after formaldehyde treatment. Sequencing of 4-thiouridine (4SU)-labelled nascent transcripts revealed no difference between untreated WT and *CSB⁻/⁻* cells. Immediately after formaldehyde treatment, nascent transcription shifted to TSSs, with no apparent nascent transcription within gene bodies (Fig. 4h,i). This observation indicates that DPCs impair progression of RNAPII in gene bodies,

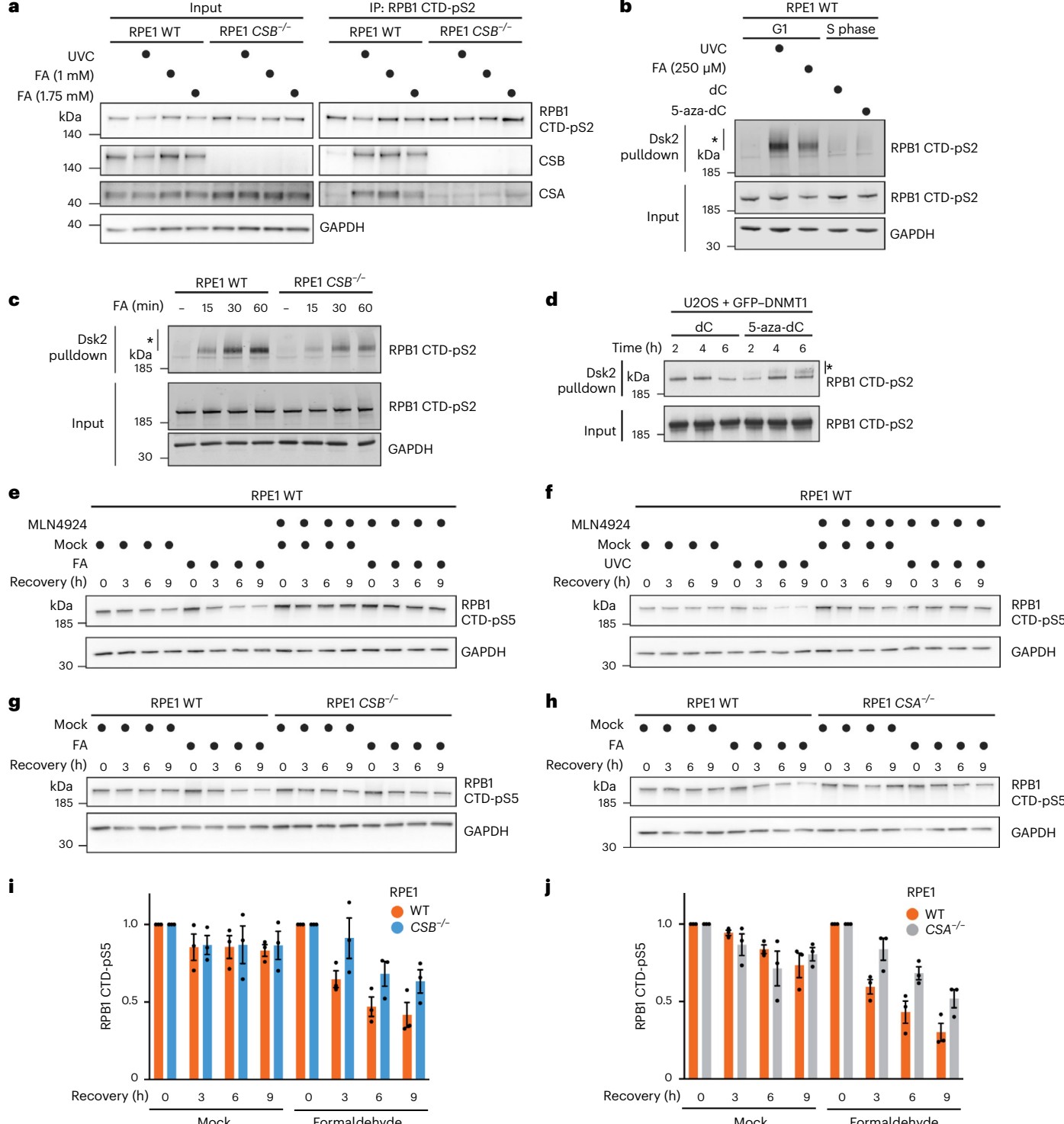

**Fig. 3 | CSB initiates a pathway that supports transcription recovery following DPC induction. a**, Immunoprecipitation (IP) of chromatin-bound elongating RNAPII (RPB1 CTD-pS2) following treatment with UVC or formaldehyde (FA) at the indicated doses, representative of three independent experiments. **b**, Dsk2 pulldown of ubiquitylated proteins from RPE1 cells synchronized in G1 by serum starvation and treated with UVC or FA, or from cells released from a thymidine block into S-phase in the presence of deoxycytidine (dC) or 5-aza-dC, representative of three independent experiments. **c**, Dsk2 pulldown in WT and *CSB*⁻/⁻ RPE1 cells synchronized in G1 by serum starvation and treated with 250 µM FA for the indicated times, representative of four independent experiments. **d**, Dsk2 pulldown in U2OS GFP–DNMT1 cells released from a single thymidine block into dC or 5-aza-dC for the indicated

times, representative of three independent experiments. **e**,**f**, RPB1 degradation in cycloheximide-treated RPE1 cells following formaldehyde (**e**) or UVC (**f**) treatment in the presence or absence of MLN4924, an inhibitor of Cullin-dependent ubiquitylation, representative of three independent experiments. **g**,**h**, RPB1 degradation in cycloheximide-treated WT and *CSB*⁻/⁻ (**g**) or *CSA*⁻/⁻ (**h**) RPE1 cells at the indicated timepoints after a pulsed FA treatment. For **g** and **h**, GAPDH blot images are also shown alongside blots in Extended Data Fig. 3j,k, respectively, due to detection of RPB1 CTD-pS2 and RPB1 CTD-pS5 from the same experiment. **i**,**j**, Quantification of **g** and **h**, respectively; error bars ± s.e.m., *n* = 3 replicates. In **b**–**d**, *denotes polyubiquitylated RPB1 CTD-pS2. Source numerical data and unprocessed blots are available in Source data.

rather than causing an acute effect on transcription initiation. After 9 h post-formaldehyde treatment, we observed recovery of nascent transcription in the gene bodies analysed in WT cells. By contrast, nascent transcription in $CSB^{-/-}$ cells was strongly delayed at this timepoint (Fig. 4i–j). Notably, nascent transcription was still shifted to the TSS at 9 h post-formaldehyde treatment, suggesting partial recovery in a fraction of the cells, leading to a mixed profile. Metagene analysis for gene groups of different lengths showed a similar recovery defect in $CSB^{-/-}$ cells for shorter (25–50 kb) and longer genes (>100 kb; Fig. 4j).

Taken together, these data suggest that DPCs cause transcription stress that triggers a CSB-dependent response, which enables elongating RNAPII to overcome DPCs throughout gene bodies and resume productive transcription.

## NER is not required for transcription recovery after formaldehyde

Next, we explored whether CSA was also required for transcription recovery after DPC induction. Compared with controls, we observed a substantial delay but eventual transcription recovery following release from formaldehyde treatments in both $CSA^{-/-}$ and $CSB^{-/-}$ RPE1 cells (Fig. 5a and Extended Data Fig. 5e). In contrast, UVC irradiation caused an irreversible transcription arrest in these cells at the dose used (Fig. 5b).

We next addressed whether the transcription recovery and hypersensitivity phenotypes of formaldehyde-treated $CSB^{-/-}$ and $CSA^{-/-}$ cells might arise solely from deficient RPB1 ubiquitylation. We first assessed the role of ELOF1, which promotes RPB1 ubiquitylation upon UV treatment[32,33]. In addition to their expected illudin S hypersensitivity, $ELOF1^{-/-}$ cells were hypersensitive to formaldehyde but only mildly more sensitive to 5-aza-dC than WT cells (Extended Data Fig. 5f–h). Furthermore, after formaldehyde treatment, $ELOF1^{-/-}$ cells displayed a less obvious transcription recovery defect than $CSB^{-/-}$ cells (Fig. 5c). To more directly explore the impact of RPB1 ubiquitylation, we used HeLa cells with a lysine-to-arginine mutation at the RPB1 ubiquitylation site (RPB1$^{K1268R}$)[31,35]. RPB1$^{K1268R}$ cells displayed mild formaldehyde and 5-aza-dC hypersensitivity in addition to illudin S hypersensitivity (Extended Data Fig. 5i–k). However, while RPB1$^{K1268R}$ cells displayed a transcription recovery defect following formaldehyde treatment, the phenotype was less pronounced than in $CSB^{-/-}$ cells (Fig. 5d). Together, these observations demonstrate that RPB1 K1268 ubiquitylation contributes towards the response to DPC-induced transcription arrest, but also underlines a role for CSB and CSA in transcription recovery and cell viability following DPC induction beyond RPB1 ubiquitylation.

Our genetic data indicated that CSB and CSA are required for DPC tolerance but other canonical GG/TC-NER factors are not (Fig. 1c,e,k and Extended Data Fig. 1j). Therefore, we assessed the requirement of various NER factors acting downstream of RPB1 ubiquitylation for transcription recovery post-formaldehyde treatment. $UVSSA^{-/-}$ cells were hypersensitive to both formaldehyde and 5-aza-dC (Extended Data Fig. 5l,m), displayed partially delayed transcription recovery following formaldehyde release and did not recover transcription after UVC irradiation (Fig. 5e,f and Extended Data Fig. 6a). Interestingly however, RPB1 polyubiquitylation or degradation was not overtly

impaired in $UVSSA^{-/-}$ cells (Extended Data Fig. 6b–d). To examine the effect of abrogating NER entirely, we tested transcription recovery in $XPC^{-/-}$, $XPC^{-/-}/CSB^{-/-}$ and $XPC^{-/-}/XPA^{-/-}$ RPE1 cells. Unlike in $XPC^{-/-}/CSB^{-/-}$ cells, we observed no transcription recovery defect in $XPC^{-/-}/XPA^{-/-}$ cells following formaldehyde release (Fig. 5g and Extended Data Fig. 6e). By contrast, recovery after UVC treatment was strongly affected by XPA loss (Fig. 5h). Next, we examined the roles of the NER nucleases ERCC1–XPF and XPG. Despite $ERCC1^{-/-}$ or $XPG^{-/-}$ cells (Extended Data Fig. 6f,g) displaying formaldehyde hypersensitivity and an intermediate 5-aza-dC sensitivity phenotype compared with $CSB^{-/-}$ cells (Extended Data Fig. 6h,i), neither ERCC1 nor XPG loss compromised recovery of transcription following formaldehyde release (Fig. 5i and Extended Data Fig. 7a). This contrasted with the situation following UVC treatment (Fig. 5j) and is consistent with the absence of a transcription recovery defect in formaldehyde-treated $XPA^{-/-}$ cells (Fig. 5g).

We concluded that the upstream TC-NER factors CSB and CSA are required for transcription recovery when DPCs block RNAPII but operate in a pathway distinct from classical TC-NER. On the basis of the intermediate phenotypes observed in $ELOF1^{-/-}$ and RPB1$^{K1268R}$ cells, RPB1 K1268 ubiquitylation contributes to this pathway but does not account for all of the phenotypes exhibited by CSB- and CSA-deficient cells. The transcription recovery phenotype observed in these cells after formaldehyde treatment correlated closely with 5-aza-dC sensitivity. We therefore surmise that the lesions causing formaldehyde sensitivity in $XPA^{-/-}$, $ERCC1^{-/-}$ or $XPG^{-/-}$ cells are probably not only DPCs, but also other cross-linking damages.

## Transcription-coupled DPC repair depends on CSB

Our results suggested a transcription-coupled DPC repair pathway coordinated by CSB and CSA, but confounding effects caused by other types of formaldehyde-induced lesions could not be excluded. Therefore, we used the recently established purification of x-linked proteins (PxP) approach coupled to mass spectrometry (PxP–MS)[26] to determine which formaldehyde-cross-linked proteins were repaired during a 6 h release from DPC induction (Extended Data Fig. 7c and Supplementary Tables 3 and 4). Consistent with previous findings[26], these proteins included histones and other chromatin-associated proteins (Extended Data Fig. 7c,d). Notably, CSB loss did not delay DPC resolution at a global level (Extended Data Fig. 7d), implying that CSB may rather function in DPC repair at specific—presumably transcriptionally active—loci.

To test this, we developed a DPC-sequencing (DPC-seq) approach, enabling genome-wide mapping of formaldehyde-induced DPCs, based on the established KCl–sodium dodecyl sulfate (SDS) precipitation assay[60]. In brief, cells were treated with formaldehyde, then collected directly or following recovery and lysed in denaturing SDS-containing buffer. Following sonication, proteins and cross-linked protein–DNA complexes were precipitated using KCl. After additional rounds of re-solubilization and re-precipitation, cross-linked DNA was treated with proteinase K, followed by DNA library preparation and next-generation sequencing (NGS) (Fig. 6a). Using read coverage as a surrogate for DPC presence, DPC-seq enabled us to monitor DPC formation and repair across the genome.

---

**Fig. 4 | CSB supports transcription recovery following DPC induction.**
**a–c**, Quantification of recovery of RNA synthesis (RRS) assays paired with NPM1 staining to enable stratified quantification of relative EU intensity in nuclear (**a**), nucleolar (**b**) and nucleoplasmic (**c**) regions in WT or $CSB^{-/-}$ cells treated with formaldehyde and released into fresh medium for the indicated times. Error bars ± s.e.m., $n$ = 3 replicates. **d**, Representative images from RRS assays in **a–c**. Scale bars, 10 μm. **e**, RT–qPCR for the indicated targets normalized to a chicken spike-in mRNA in WT or $CSB^{-/-}$ cells treated with formaldehyde and released into fresh medium for 6 h. Values were normalized to transcript levels in untreated conditions. Error bars ± s.e.m., $n$ = 3 replicates. *$P$ < 0.05 based on multiple two-sided Mann–Whitney tests; $P$ = 0.029 (SIRT1), $P$ = 0.029 (TBP), $P$ = 0.029 (EXO1) and $P$ = 0.029 (ERCC4). **f**, Western blot analysis of hyperphosphorylated

(elongating) and hypophosphorylated (initiating) RPB1, denoted by asterisk and hash symbol, respectively, upon formaldehyde treatment for the indicated timepoints, representative of three independent experiments. **g**, Western blot analysis of ATF3 induction and degradation in WT or $CSB^{-/-}$ RPE1 cells upon treatment with formaldehyde for the indicated times, or with UVC and allowed to recover for 24 h, representative of three independent experiments. **h**, Nascent RNA-seq heat maps of 3–100 kb genes in WT and $CSB^{-/-}$ cells treated with formaldehyde and released for 9 h. **i**, An example genome browser plot from **h** of the $MAP3K14$ locus. **j**, Metagene profiles of normalized (Norm.) read counts from **h** of genes 25–50 kb, 50–100 kb and >100 kb in length, shown in full (left), ±5 kb around the TSS (centre) or beginning from 5 kb into the gene body (right). Source numerical data and unprocessed blots are available in Source data.

In G1-synchronized RPE1 cells, DNA quantification following KCl–SDS precipitation confirmed robust DPC induction by formaldehyde, with precipitated DNA reducing after 6 h recovery in drug-free media (Extended Data Fig. 8a). Upon sequencing of these samples, we determined DPC coverage across gene bodies by mapping our data onto the human genome (Fig. 6b,c). Sequence reads dropped substantially 6 h post-formaldehyde release, indicating repair (Fig. 6b,c).

Interestingly, our analyses revealed that formaldehyde-induced DPCs formed preferentially at TSSs (Fig. 6b). Since TSSs are relatively nucleosome free but enriched for RNAPII, our findings implied that formaldehyde may cause DNA cross-linking of RNAPII at TSSs. Indeed, our PxP–MS data revealed the induction of cross-linked RNAPII subunits after formaldehyde treatment (Extended Data Fig. 7d). To assess the formation of RNAPII DPCs directly, we performed Cleavage Under

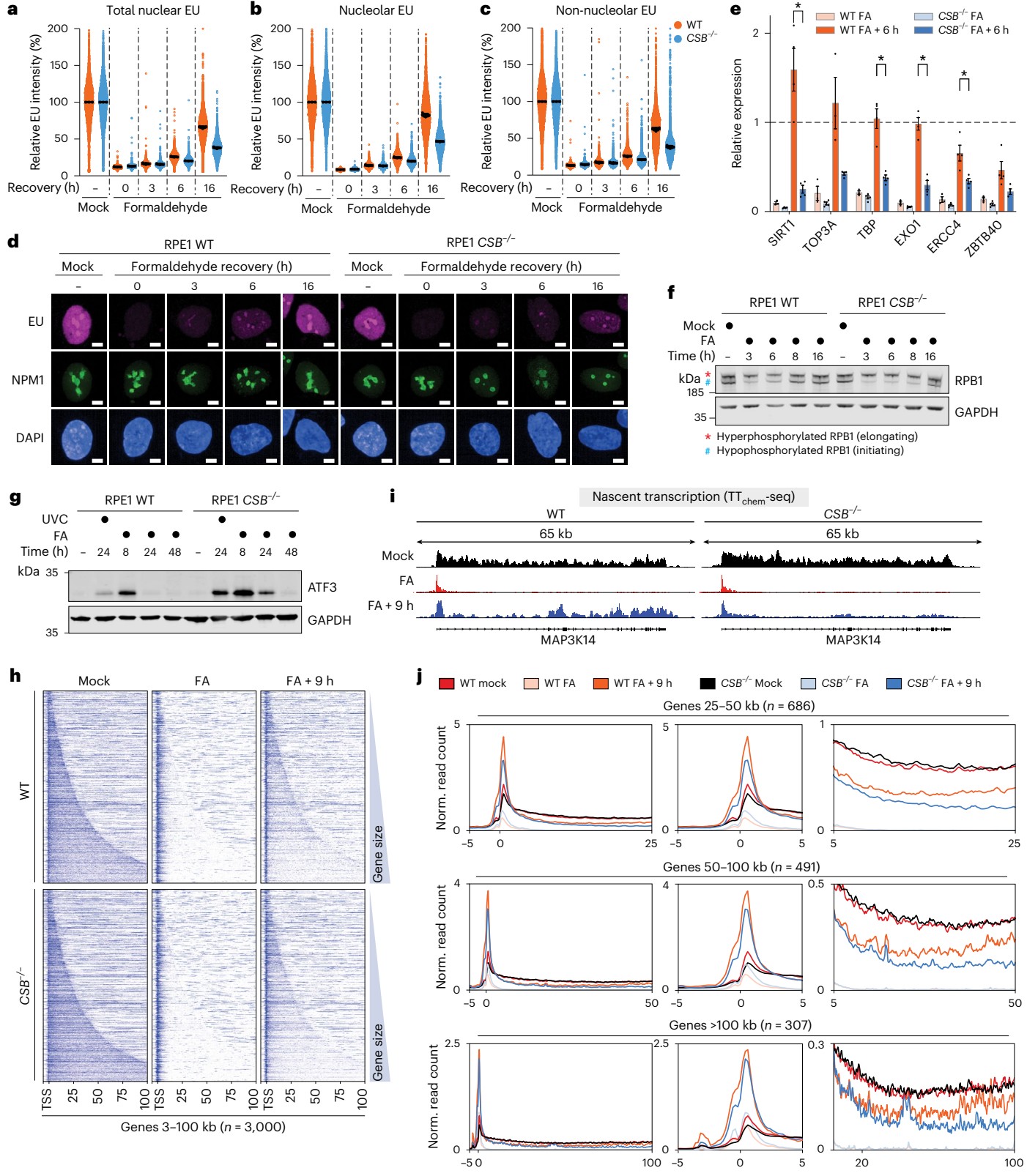

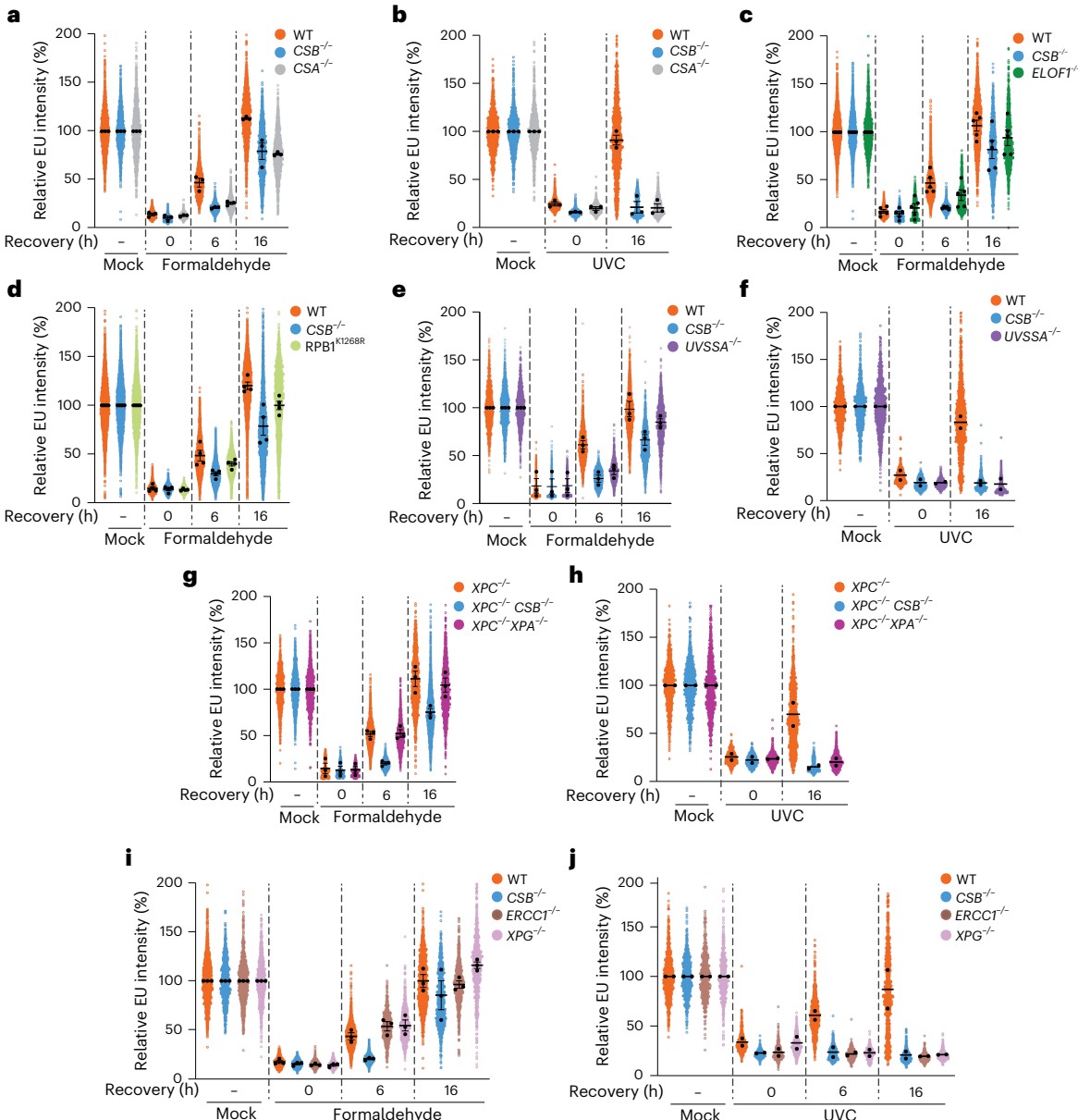

**Fig. 5 | Upstream TC-NER factors support transcription recovery after DPC induction. a,b**, RRS assays in WT, *CSB*$^{-/-}$ or *CSA*$^{-/-}$ RPE1 cells treated with formaldehyde (**a**) or UVC (**b**) and released for the indicated times. **c**, The same as for **a** but with WT, *CSB*$^{-/-}$ and *ELOF1*$^{-/-}$ RPE1 cells. **d**, The same as for **a** but with WT, *CSB*$^{-/-}$ and RPB1$^{K1268R}$ HeLa cells. **e,f**, The same as for **a** and **b** but with WT, *CSB*$^{-/-}$ and *UVSSA*$^{-/-}$ RPE1 cells. **g,h**, The same as for **a** and **b** but with *XPC*$^{-/-}$, *XPC*$^{-/-}$/*CSB*$^{-/-}$ and *XPC*$^{-/-}$/*XPA*$^{-/-}$ RPE1 cells. **i,j**, The same as for **a** and **b** but with WT,

*CSB*$^{-/-}$, *ERCC1*$^{-/-}$ and *XPG*$^{-/-}$ RPE1 cells. Mean intensities of replicates ($n = 3$ for **a**, **b**, **e**, **g** and **i**; $n = 5$ for **c**; and $n = 4$ for **d**) are displayed as black dots with a mean of those averages shown, error bars ± s.e.m. For **f**, **h** and **j**, $n = 2$ and mean intensities of individual replicates are shown as black dots and the mean of those values is shown as a black line. Note that for **c**, data from WT and *CSB*$^{-/-}$ cells are shared with some replicates from other images shown in this figure. Source numerical data are available in Source data.

Targets and Tagmentation (CUT&Tag) by using antibodies against RPB1 CTD-pS5 or CTD-pS2 under high salt conditions and without fixation to enrich for cross-linked RNAPII. We observed substantial formaldehyde-induced RNAPII cross-links at TSSs that were resolved over a 6 h recovery period in a CSB-independent manner (Extended Data Fig. 8b,c). Consistent with this, PxP–western blot (WB) demonstrated CSB-independent resolution of formaldehyde-induced RNAPII DPCs (Extended Data Fig. 8d). Importantly, the DPC-seq peak observed at the TSS overlapped strongly with the formaldehyde-cross-linked RNAPII identified by our stringent CUT&Tag approach (Extended Data Fig. 8e,f). Of note, while establishing DPC-seq, we noticed that this TSS peak was particularly affected by shearing caused by manual handling

and sonication (Extended Data Fig. 8g,h), which should be considered when applying this technique.

Given that formaldehyde causes the formation of histone DPCs in particular[26], we next considered the impact of chromatin accessibility on DPC induction and repair. Comparing our DPC-seq profiles with existing chromatin accessibility datasets from assay for transposase-accessible chromatin with sequencing (ATAC-seq) experiments in RPE1 cells (Gene Expression Omnibus (GEO): GSE209659), we found that formaldehyde induced substantially more DPCs in inaccessible, histone-rich chromatin, but repair occurred irrespective of accessibility (Fig. 6d). To test whether DPC-seq can be used to assess the contribution of different DPC repair pathways, we monitored

DPC repair while inhibiting the proteasome, which degrades DPCs in replication-coupled and GG pathways[13,20–23,26]. Upon treating cells with the proteasome inhibitor MG132 throughout a 6 h release from formaldehyde, we found that proteasome inhibition caused a global delay in DPC repair that was most pronounced in highly accessible chromatin (Extended Data Fig. 8i).

Next, we sought to understand whether transcription affects DPC repair rates. By using existing RPB1 chromatin immunoprecipitation sequencing (ChIP-seq) data[61], we compared DPC coverage upon formaldehyde treatment and release with RNAPII occupancy[60] and found that, in general, genes with higher RPB1 occupancy exhibited the most dramatic drop in DPC coverage over the 6 h recovery period (Fig. 6e and Extended Data Fig. 9a). Since this was strongly suggestive of transcription-coupled DPC repair, we next performed DPC-seq in the presence of the transcription inhibitor flavopiridol. Flavopiridol strongly reduced DPC recovery specifically at highly transcribed genes (Fig. 6f), defined by RPB1 occupancy using existing data[61]. We were thus able to identify genes with significantly increased DPC coverage in flavopiridol-treated cells 6 h after treatment, specifying these genes as undergoing transcription-dependent DPC repair (Fig. 6g). Directly comparing DNA accessibility and RPB1 occupancy revealed a subpopulation of genes with high transcriptional activity and high accessibility (Fig. 6h). Strikingly, this subpopulation contained around 62% of the genes whose DPC repair was significantly slowed by flavopiridol (Fig. 6i). Together, these findings demonstrate transcription-coupled DPC repair.

We next used DPC-seq to assess the impact of CSB loss on DPC repair after formaldehyde treatment and release. While there was little difference in DPC induction between WT and $CSB^{-/-}$ cells (Extended Data Fig. 9b), we identified a subset of genes that, after 6 h of recovery following formaldehyde treatment, displayed a statistically significant increase in DPC coverage in $CSB^{-/-}$ cells compared with WT cells (Fig. 7a). Within these loci, the most dramatic differences between WT and $CSB^{-/-}$ cells occurred within gene bodies—a phenomenon that became most striking when we quantified fold-changes in DPC coverage between $CSB^{-/-}$ and WT cells across the length of CSB-dependent genes (Fig. 7b,c). These findings demonstrated that CSB loss causes DPCs' persistence across the gene body, but not particularly upstream or downstream of the gene. In addition, the DPC enrichment in $CSB^{-/-}$ compared with WT cells was not seen initially after formaldehyde induction but only after a 6 h recovery from formaldehyde treatment, showing that it specifically reflected impacts on DPC resolution, not formation.

To investigate the features that promote transcription-dependent and CSB-dependent DPC repair, we compared the relative levels of different genomic features in genes whose DPC repair was delayed upon transcription inhibition or CSB knockout, respectively.

Consistent with a role for CSB in transcription-coupled DPC repair, the transcription-dependent and CSB-dependent gene sets bore strong similarities in feature enrichment compared with genes unaffected by either perturbation (Fig. 7d). In CSB-dependent gene sets, protein-coding genes were additionally enriched (Fig. 7d and Extended Data Fig. 9c). This analysis further revealed that while gene length and DNA accessibility (based on ATAC-seq data) both correlated with transcription and CSB dependence, the strongest enriched feature of both gene sets was transcriptional activity (Fig. 7d and Extended Data Fig. 9d–f).

In agreement with our observation that 62% of genes whose DPC repair was transcription dependent were highly transcriptionally active and highly accessible (Fig. 6i), we found that 50% of genes at which repair was CSB dependent were also found in this population (Fig. 7e). Correspondingly, only 19% of genes whose repair was not CSB dependent fell into this high-accessibility, high-activity category, while 45% of such genes displayed low transcriptional activity and low accessibility (Fig. 7e). To further explore this, we analysed DPC-seq coverage in genes with differential transcriptional activity (as in Extended Data Fig. 9a). This showed that CSB loss did not affect DPC coverage in inactive genes, but as transcriptional activity increases, $CSB^{-/-}$ cells showed increasing enrichment of DPCs 6 h after release from formaldehyde treatment (Extended Data Fig. 9g). Next, we compared our CSB-dependent and transcription-dependent DPC-seq gene data directly. We separated genes into transcription-dependent and transcription-independent classes based on our flavopiridol DPC-seq experiments (Fig. 6g) and determined the respective DPC coverage in WT and $CSB^{-/-}$ cells 6 h after release from formaldehyde. These analyses revealed that CSB loss compromised DPC repair specifically in genes whose DPC repair is transcription dependent (Fig. 7f).

Taken together, our DPC-seq data provide direct evidence for CSB-dependent transcriptional DPC repair in human cells.

## Discussion

In this study, we describe a transcription-coupled DPC repair pathway. Our results suggest that DPCs act as physical barriers to RNAPII progression, resulting in recruitment of CSB and CSA, which together promote DPC repair and transcription recovery independently of classical TC-NER.

Our DPC-seq data established that formaldehyde induces DPCs around the TSS (Fig. 6b), a region that is typically relatively nucleosome free[62]. This observation, corroborated by our modified, stringent CUT&Tag approach (Extended Data Fig. 8b,c), suggests that RNAPII itself forms DPCs at the TSS. Importantly, the repair of these DPCs is CSB independent, reminiscent of a recent study demonstrating CSB-independent RNAPII degradation at promoters[63]. By contrast, our DPC-seq and nascent RNA sequencing (RNA-seq) data demonstrate

---

**Fig. 6 | DPC-seq enables genome-wide mapping of DPC induction and resolution. a**, An overview of the DPC-seq methodology. Cultured cells are treated with 1.75 mM formaldehyde for 1 h then either collected or released for 6 h to recover. Cells are lysed with SDS buffer and sonicated. DPC DNA is then precipitated with high KCl and purified. **b**, Metagene profile of DPC-seq in formaldehyde-treated RPE1 cells with or without 6 h recovery, coverage calculated as reads per million (RPM) with the TSS and TES (transcription end site) indicated. **c**, A genome browser plot showing DPC-seq coverage at a specific region of chromosome 6 with RPB1 ChIP-seq coverage added for comparison (GEO: GSE141798)[61]. **d**, DPC-seq coverage per gene, with or without 6 h recovery after treatment, in genes with low, medium or high DNA accessibility as determined by ATAC-seq (GEO: GSE209659). Statistics via paired two-sided Wilcoxon test. ***$P < 0.001$; $P$ values are $P < 2.2 \times 10^{-16}$, $P < 2.2 \times 10^{-16}$ and $P < 2.2 \times 10^{-16}$ for 0 h versus 6 h recovery in low-, medium- or high-accessibility, respectively. The box plot shows upper (Q3) and lower (Q1) quartile boundaries and line at the median. Lower whisker (minimum) is Q1 – 1.5 × interquartile range (IQR) and upper whisker (maximum) is Q3 + 1.5 × IQR. **e**, DPC-seq coverage per gene in samples recovered for 6 h after formaldehyde treatment versus non-recovered samples, coloured by RNAPII occupancy. **f**, $\log_2$ fold change of DPC-seq coverage per gene 6 h/0 h after FA treatment, with or without flavopiridol treatment, in genes grouped by RNAPII occupancy. Statistics via paired two-sided Wilcoxon test. *$P < 0.05$, **$P < 0.01$ and ***$P < 0.001$; $P$ values are 0.9809, 0.03714, $<2.2 \times 10^{-16}$ and $<2.2 \times 10^{-16}$ for comparisons in low, mid-low, mid-high and high transcriptional activity gene sets, respectively. The box plot shows upper (Q3) and lower (Q1) quartile boundaries and the line at the median. The lower whisker (minimum) is Q1 – 1.5 × IQR and the upper whisker (maximum) is Q3 + 1.5 × IQR. **g**, DPC-seq coverage per gene 6 h after formaldehyde treatment in WT RPE1 cells that are either not treated (NT) or treated with flavopiridol. Green highlights genes with significantly higher DPC coverage in flavopiridol-treated cells, indicating they undergo transcription-dependent DPC repair. **h**, Per gene RNAPII occupancy versus DNA accessibility, as determined via ATAC-seq in RPE1 cells (GEO: GSE209659). **i**, The same as **g** but only showing genes that show transcription-dependent DPC repair (as defined in **g**) and a group of matched size that do not show transcription-dependent repair, including the percentage of each group that are present in the shown quadrants. For all DPC-seq analyses, $n = 3$ biological replicates.

that CSB is important for DPC repair in gene bodies (Figs. 4i and 7b–d), probably corresponding to either cross-linked elongating RNAPIIs or polymerases stalled at cross-linked nucleosomes. It is conceivable that the bulkiness of DPCs prevents canonical TC-NER from acting, perhaps due to steric hinderance of XPA or TFIIH recruitment.

Alternatively, RNAPII cross-linking might impede its backtracking or removal, thus compromising ERCC1–XPF and XPG positioning.

This model explains why XPA, ERCC1 and XPG are all dispensable for transcription recovery after formaldehyde treatment (Fig. 5g,i) but also raises an important question: how is the DPC lesion eventually

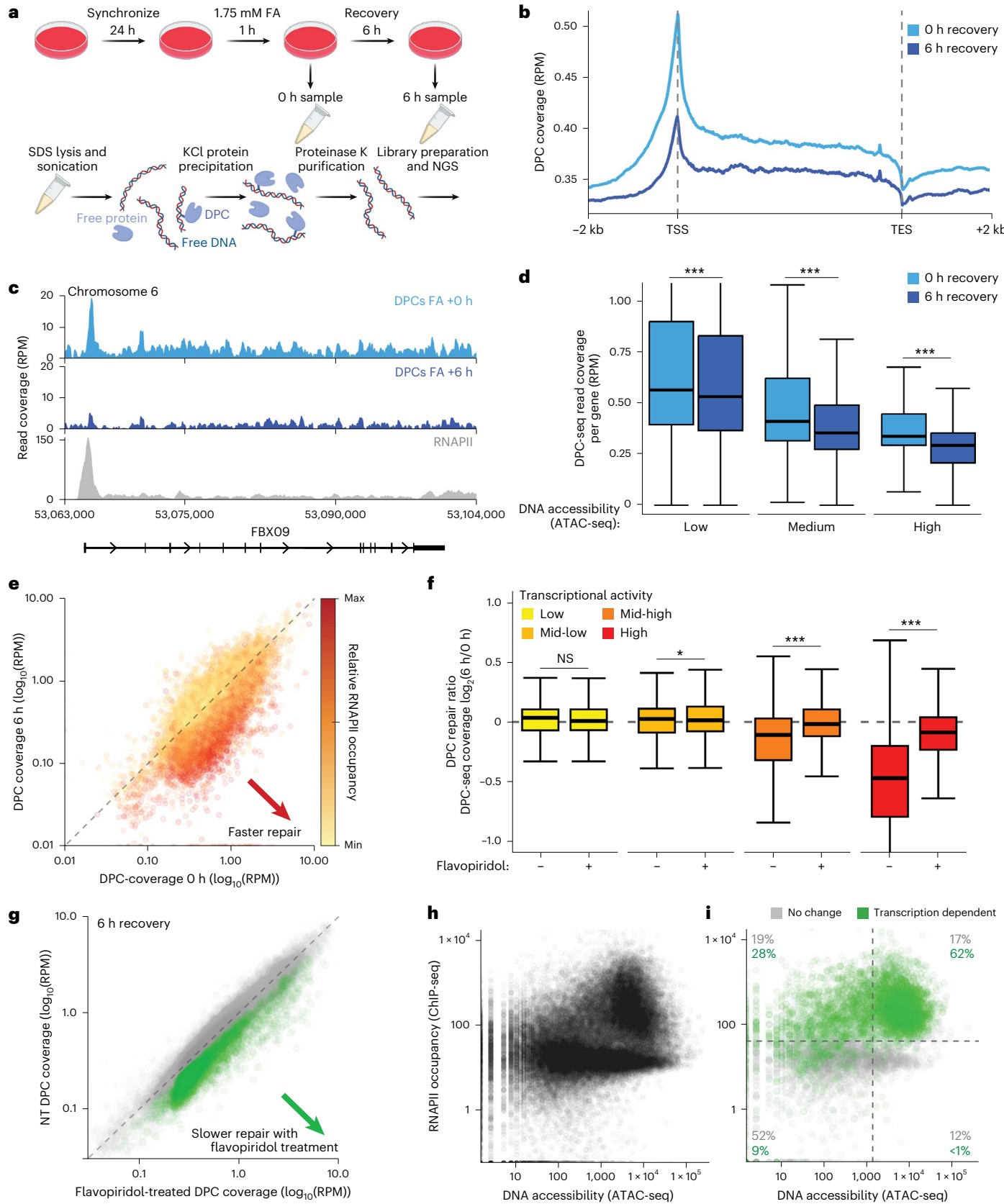

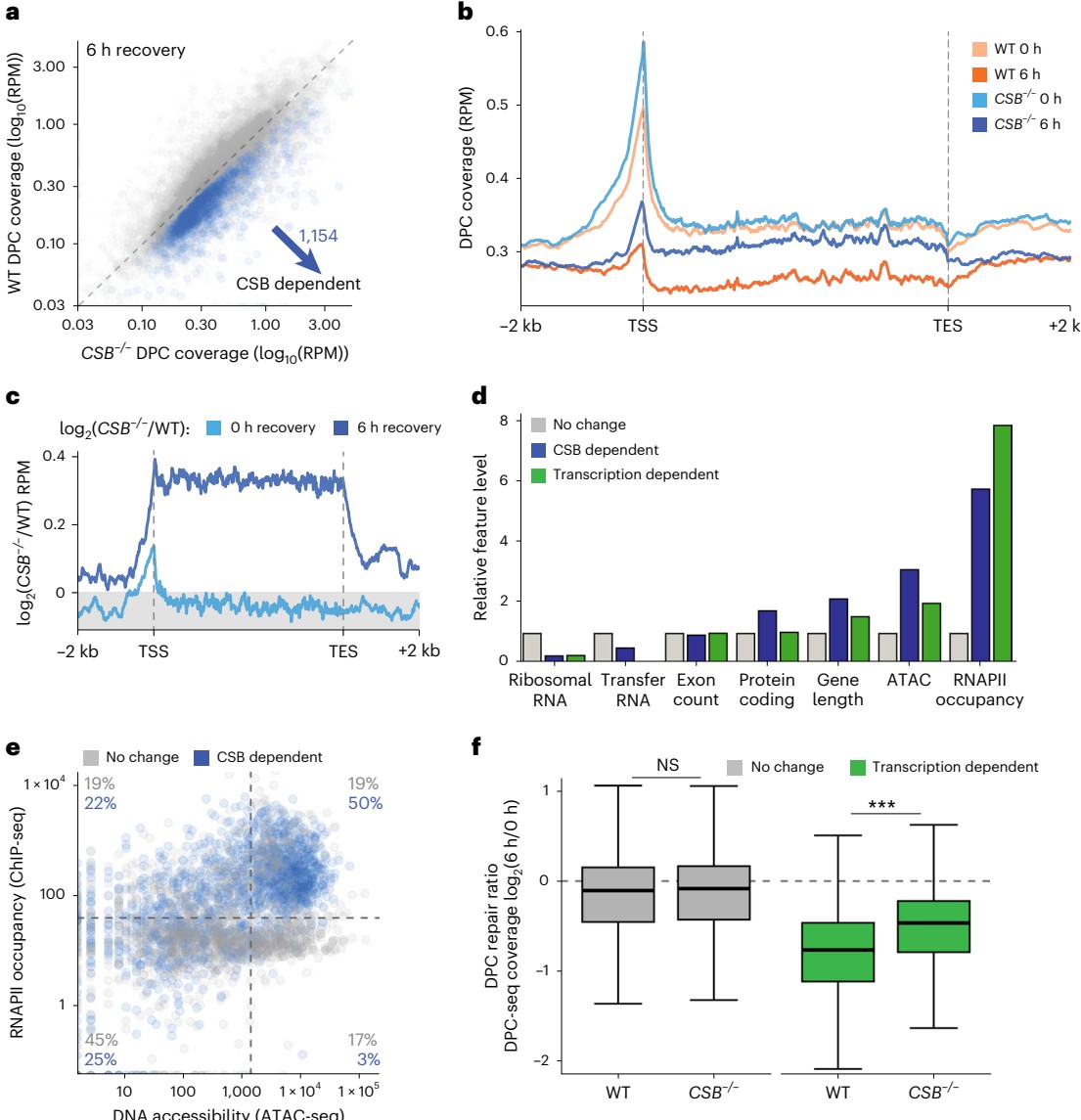

**Fig. 7 | CSB is required for the repair of DPCs at transcriptionally active loci. a**, DPC-seq coverage per gene in WT RPE1 cells versus $CSB^{-/-}$ cells 6 h after formaldehyde treatment. Blue highlights genes with significantly higher DPC coverage in $CSB^{-/-}$ cells, indicating they undergo CSB-dependent DPC repair. **b**, Metagene profile of DPC-seq coverage in WT versus $CSB^{-/-}$ RPE1 cells treated with formaldehyde and with or without 6 h recovery. Metagene specifically shows CSB-dependent genes from **a**. **c**, The same as **b** but showing $log_2$-fold change coverage for $CSB^{-/-}$/WT RPE1 cells with or without 6 h recovery after formaldehyde treatment. **d**, The level of different genomic features in CSB-dependent genes from **a** and transcription-dependent genes relative to the non-changing group. rRNA, ribosomal RNA; tRNA, transfer RNA. **e**, Per gene RNAPII occupancy versus DNA accessibility, as determined via ATAC-seq in RPE1

cells (GEO: GSE209659) showing CSB-dependent genes from **a** and non-changing genes, including the percentage of each group that are present in the shown quadrants. **f**, $Log_2$-fold change of DPC-seq coverage per gene 6 h/0 h after FA treatment in WT RPE1 cells versus $CSB^{-/-}$ cells, in genes grouped according to whether they show transcription-dependent DPC repair. Statistics via paired two-sided Wilcoxon test. ***$P < 0.001$; $P$ values are 0.109 and $<2.2 × 10^{-16}$ for comparisons in no change and transcription-dependent gene sets, respectively. The box plot shows the upper (Q3) and lower (Q1) quartile boundaries and line at the median. Lower whisker (minimum) is $Q1 − 1.5 × IQR$ and upper whisker (maximum) is $Q3 + 1.5 × IQR$. For all DPC-seq analyses, $n = 3$ biological replicates. Source numerical data are available in Source data.

removed? An involvement for SPRTN is unlikely since *SPRTN-ΔC* cells bear almost no transcription recovery defect after formaldehyde treatment, even in the absence of CSB (Extended Data Fig. 7b). Other proposed DPC proteases[64] did not score as hits in our CRISPR screens, although some of these could function redundantly among themselves and/or with SPRTN. The requirement for the CRL4−CSA E3 ubiquitin ligase complex for transcription restart after DPC induction indicates a key role for polyubiquitylation in transcription-coupled DPC repair, further supported by our DPC-seq experiments showing DPC repair defects upon proteasome inhibition (Extended Data Fig. 8i). Two non-mutually exclusive scenarios that seem plausible are as follows:

(1) CRL4−CSA polyubiquitylates RPB1 to facilitate its removal from the damage site and enable repair and (2) CRL4−CSA polyubiquitylates the DPC, targeting it for proteasomal degradation.

Indeed, our data indicate at least two distinct functions of CSB and CSA in transcription-coupled DPC repair. Loss of ELOF1, which compromises RPB1 polyubiquitylation at K1268 (refs. 32,33) or RPB1 containing the K1268R mutation that prevents its polyubiquitylation[31,35], caused intermediate transcription recovery defects following formaldehyde treatment when compared with CSB loss (Fig. 5c,d). Furthermore, polyubiquitylation and degradation of RPB1 upon formaldehyde treatment were reduced but not completely abrogated

upon CSB or CSA loss (Fig. 3c,g–j). These findings suggest that CSB/CSA-mediated RPB1 polyubiquitylation contributes to but does not comprise the entirety of CSB/CSA's function in transcription-coupled DPC repair. One possibility is that RPB1 ubiquitylation is critical when the DPC comprises RNAPII itself, but is less important for the repair of other formaldehyde-induced DPCs. Interestingly, loss of UVSSA, which is recruited to RNAPII through interaction with CSA and stabilized by Cullin-dependent RNAPII ubiquitylation[31–33,35,52], had no observable effect on RPB1 degradation but still affected transcription recovery (Fig. 5e). Thus, one possibility is that UVSSA helps direct CSA-dependent ubiquitylation to the RNAPII-blocking DPC. Once the DPC is degraded, a peptide is likely to remain cross-linked to DNA and it remains to be determined whether such peptide adducts can then be bypassed by the polymerase[65] and/or are removed by other, as-yet unknown, mechanisms.

Together, our findings reveal previously unrecognized cellular functions of CS proteins in transcription-coupled DPC repair. That this function is independent of classical TC-NER is striking because it might help explain pathological disparities between CS and xeroderma pigmentosum. Compared with xeroderma pigmentosum, CS is a more severe and multi-faceted disease, with a complex and incompletely understood etiology. In addition to the well-documented DNA repair defect, dysregulation of the transcriptional landscape[31,66–69] and defective RNAPII processing[35,70] have been suggested to contribute to CS. Importantly, endogenous formaldehyde has recently been shown to precipitate CS features in mice[42]. While this study, in line with ours, reported that XPA (and therefore TC-NER) contributes to cellular formaldehyde tolerance, it also showed that CSB loss confers a more severe phenotype than XPA loss in mice lacking the formaldehyde-detoxifying enzyme ADH5, indicating an important functional distinction between XPA and CSB in formaldehyde tolerance in vivo[42]. These observations might be explained by our finding that cells lacking XP proteins are proficient in the repair of formaldehyde-induced DPCs, while cells lacking CS proteins are not.

This work, along with two complementary studies reporting similar findings[71,72], represents an important step forward in the understanding of both the mechanisms of DPC repair and the cellular etiology of Cockayne Syndrome. The discovery of transcription-coupled DPC repair poses the question as to how this repair mechanism interplays with the established replication-coupled and GG DPC repair pathways. It will be important to investigate how pathway choice is regulated and to determine the contribution of each mechanism across different cell cycle stages, tissues and differentiation states. DPC-seq could be instrumental in answering such questions.

## Availability of biological materials

Newly established cell lines described in this study are available from the corresponding authors upon reasonable request, subject to establishment of a suitable material transfer agreement where relevant.

### Reporting summary

Further information on research design is available in the Nature Portfolio Reporting Summary linked to this article.

## Online content

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

## Methods

### Cell culture and transfections

For details of cell lines used, see Supplementary Table 5. *CSB*[−/−] HAP1 cells complemented with doxycycline-inducible expression constructs were established by first transducing *CSB*[−/−] HAP1 cells with pT2A-hygro-containing lentivirus, which bears the TET3G-T2A element (VectorBuilder). After hygromycin-B selection (200 µg ml[−1]; Thermo Scientific), single cell-derived clones were established and validated. One clone was transduced with lentiviral particles containing pTREG3-HA-EV-puro, pTREG3-HA-CSB-puro (all synthesized by VectorBuilder) or pTREG3-HA-CSB[K538R]-puro. Cells were selected with 1 µg ml[−1] puromycin and expression of proteins was induced using 1 µg ml[−1] doxycycline for 48 h. Generation of *CSB*[−/−] RPE1 cells expressing GFP–EV, GFP–CSB[WT] and GFP–CSB[K538R] was performed using lentiviral delivery and 3 µg ml[−1] puromycin selection, using pGFP-lentiPuro constructs generated by VectorBuilder.

RPE1 *SPRTN-ΔC* and RPE1 *CSB*[−/−]/*SPRTN-ΔC* double mutant genome-edited cell lines were generated from RPE1-TetOn-Cas9-PuroS-TP53[−/−] and -TP53[−/−]/CSB[−/−] (referred to as RPE1 WT and *CSB*[−/−] throughout this manuscript, respectively), by co-transfection with lenti multi-guide plasmid containing gRNA_SPRTN-ΔC#1 and gRNA_SPRTN-ΔC#2 and px330 (Addgene, 82580) containing gRNA_SPRTN-ΔC#2 (Supplementary Table 6) (1 µg + 1 µg DNA). At 96 h after transfection, puromycin (1 µg ml[−1]) was added and increased to 2 µg ml[−1] the following day. After selection, single cell-derived clones were selected and expanded. Editing efficiency was assessed by western blotting.

To generate *XPG*[−/−] RPE1 cells, RPE1-TetOn-Cas9-PuroS-TP53[−/−] cells were transfected with Cas9-2A-EGFP (pX458; Addgene #48138) containing sgXPG-4. Single clones were grown and verified by western blot and Sanger sequencing.

GFP–/GFP–DNMT1-expressing U2OS cells were established by transfecting U2OS cells with pEGFP–C1 or pEGFP–C1–DNMT1 (synthesized by VectorBuilder) using Lipofectamine 2000. At 36 h after transfection, cells were selected with 1 mg ml[−1] G418. Single cell-derived clones were selected, expanded and GFP/GFP–DNMT1 expression was validated by live cell imaging and western blotting.

SiRNA transfections were performed using Lipofectamine RNAiMAX (Thermo Fisher Scientific) according to the manufacturer's instructions. The details of siRNAs used in this study can be found in Supplementary Table 6.

### CRISPRi screens

A pooled single guide RNA library was generated by combining the human protein-coding genome-wide CRISPRi-v2 library (containing 5 gRNA per gene; Addgene #83969) with multiple CRISPRi non-coding libraries (10 gRNA per gene; Addgene #86538, #86539, #86544, #86548 and #86549), covering the all long non-coding RNA genes expressed in K562 cells. Sublibraries were amplified in E. cloni cells (10G Elite duo, Lucigen) and mixed according to the relative amount of gRNA per individual sublibrary. Library transduction was performed as described[73]. Briefly, cells were transduced by spinoculation and transduced cells were selected with 0.75 µg ml[−1] puromycin. A reference sample was collected 48 h post-transduction, the multiplicity of infection was calculated at day 6 and drug treatment was started at day 12. Every other day, cells were counted and replated at 500× representation with or without formaldehyde or 5-aza-dC at half-maximum inhibitory concentration (155 µM and 55 nM, respectively). Cell pellets were collected for genomic DNA (gDNA) extractions and library preparation/NGS at day 2 (initialome), day 10 (essentialome) and day 24 (endpoint). gDNA was extracted using the QIAamp DNA Blood Maxi Kit (Qiagen). Library preparation was then performed by subjecting 561 µg gDNA from the reference sample and 5.84 µg from experimental samples to 28 PCR cycles with NEBNext Ultra II Q5 Master Mix (New England Biolabs), using custom indexed primers (Supplementary Table 6) for simultaneous amplification of the backbone and addition of Illumina adaptors. PCR products were purified by excision and extraction from 1% agarose using the QIAquick Gel Extraction Kit (Qiagen). Sample concentrations were checked by a NanoDrop and diluted to a final concentration of 20 nM. Samples were multiplexed at 20 nM and sequenced on an Illumina HiSeq1500 using the CRISPRi_F1 primer. Enriched/depleted gRNAs in the endpoint samples compared to the 48 h reference sample were determined using DrugZ[74].

### Alamar blue assays

RPE1 cells (1,000 per well) were seeded in triplicate in 24-well plates and treated the next day with compounds (methanol-free formaldehyde, Fisher Scientific; 5-aza-dC, Sigma-Aldrich and illudin S, Santa Cruz). After 5 days, growth medium was replaced with 0.5 ml Alamar blue cell viability reagent (36 µg ml[−1] resazurin in phosphate-buffered saline (PBS)) and plates were incubated for 1 h at 37 °C. Viability was assessed by using fluorescence (560 nm excitation/590 nm emission).

For assays with siRNA-transfected cells, cells were transfected 48 h before seeding and treated with the appropriate compounds the following day (72 h after transfection).

### Colony formation assays

Cells were seeded at 400–1,000 per well in 6-well plates in technical triplicate. After 24 h, drugs (5-aza-dC, formaldehyde or illudin S) were added in fresh media. Formaldehyde-containing media were replaced with fresh media after 24 h. After 6–8 days, cells were fixed in 0.25% (wt/vol) crystal violet with 25% ethanol and washed with distilled water. Plates were scanned and colonies counted using FIJI. Technical triplicates were averaged and treated as one biological replicate. To assess UVC sensitivity, 4,000 cells per dish were seeded in triplicate in 6 cm dishes. The following day, cells were irradiated in PBS with UVC and PBS was replaced with fresh media. After 7 days, cells were stained with crystal violet. For assays with siRNA transfections, cells were first transfected in 6-well plates and clonogenic survival assays were seeded after 48 h. Excess cells were re-seeded for collection of western blot lysates 24 h later.

### RNAPII co-immunoprecipitation

Elongating RNAPII immunoprecipitation was performed as previously described[52]. Cells were treated with formaldehyde (1 mM or 1.75 mM) or UVC (20 J m[−2]) and collected after 1 h. Chromatin enrichment was performed on ice in IP-130 buffer (30 mM Tris pH 7.5, 130 mM NaCl, 2 mM MgCl$_2$, 0.5% Triton X-100 (Sigma-Aldrich) and cOmplete ethylenediaminetetraacetic acid (EDTA)-free protease inhibitor cocktail (Roche)) for 30 min, followed by centrifugation at 10,000g for 10 min and removal of the supernatant. Cell pellets (chromatin fractions) were lysed in IP-130 with 500 U ml[−1] Benzonase (Merck), and 2 µg RNAPII-S2 antibody (Abcam; Supplementary Table 7) for 3 h at 4 °C. Protein complexes were immunoprecipitated by 1.5 h incubation with Protein A Agarose Beads (Sigma-Aldrich). Beads were washed 6× with IP-130 and immunoprecipitates were eluted by boiling in 2× NuPAGE lithium dodecyl sulfate (LDS) sample buffer.

### Dsk2 pulldown

RPE1 cells were synchronized in 1% fetal bovine serum (FBS)-containing media for 24 h before formaldehyde (250 µM) or UVC (20 J m[−2]) treatment and collection 1 h later or as indicated. For pulldowns with dC/5-aza-dC, cells were grown to 60% confluency and synchronized at the G1/S boundary with 2 mM thymidine (Sigma-Aldrich) for 20–24 h. Cells were released into S-phase or 30 min and treated with 10 µM dC/5-aza-dC. Cells were collected by scraping in PBS and centrifugation at 500g for 5 min. Pellets were snap frozen on dry ice and stored at −80 °C.

Preparation of GST–Dsk2 affinity resin and subsequent Dsk2 pulldowns were performed essentially as previously described[53].

Cells were lysed in TENT buffer (50 mM Tris/HCl ph 7.5, 2 mM EDTA, 150 mM NaCl and 1% Triton X-100) supplemented with cOmplete Protease Inhibitor Cocktail tablets and 2mM N-ethylmaleimide (Sigma-Aldrich). Samples were incubated on ice for 10 min and sonicated extensively before centrifugation at full speed for 5 min at 4 °C. Supernatants were transferred to fresh tubes, protein concentrations were determined by the Bradford assay and standardized in TENT buffer. Ten per cent was removed as an input and prepared for western blotting.

Next, 0.5 ml GST–Dsk2 beads per 1 mg protein were equilibrated in ice-cold TENT buffer before distribution across samples. Pulldowns were performed overnight at 4 °C with rotation. Beads were washed three times by centrifugation (500$g$ for 5 min at 4 °C) and resuspension in TENT buffer, then washed once with cold PBS. Elution was performed by boiling in 2× SDS loading dye containing 1 mM dithiothreitol (DTT) at 95 °C for 5 min.

### RPB1 degradation and ATF3 induction and degradation assays

For RPB1 degradation experiments, cells were pre-treated with 100 μg ml$^{-1}$ cycloheximide for 1 h, then treated with 1.75 mM formaldehyde for 1 h or 20 J m$^{-2}$ UVC and allowed to recover for the indicated time or, for chronic treatments, treated with 250 μM formaldehyde for the indicated times. Where indicated, cells were pre-treated with or without 2 μM NEDDi (MLN4924) for 1 h before UVC/formaldehyde treatment. To assess disappearance/recovery of hypophosphorylated RPB1, cells were synchronized in 1% FBS-containing medium and treated with 250 μM formaldehyde for the indicated times. For ATF3 induction and degradation experiments, cells were synchronized in medium containing 1% FBS and treated with 250 μM formaldehyde for the indicated times, or treated with 9 J m$^{-2}$ UVC and allowed to recover for 24 h. Cells were lysed in 1× NuPAGE LDS sample buffer, followed by SDS–polyacrylamide gel electrophoresis and western blotting with the indicated antibodies.

### RRS assays

For recovery of RNA synthesis (RRS) assays, adherent cells on coverslips or in 24-well imaging plates were synchronized with 1% FBS-containing media for 24 h. Cells were then treated with 1.75 mM (RPE1) or 0.75 mM (HAP1) formaldehyde, or irradiated with UVC 20 J m$^{-2}$. After 1 h, formaldehyde-treated cells were washed twice with PBS and allowed to recover for the indicated times before pulse labelling with 400 μM EU (Jena Bioscience) for 1 h, followed by a 30 min chase with Dulbecco's modified Eagle medium. Cells were fixed with 4% formaldehyde in PBS for 15 min. For EU incorporation upon 5-aza-dC treatment, cells were synchronized using a double thymidine block, based on two cycles of overnight incubation with 2 mM thymidine (Sigma-Aldrich) and a 9 h release in between. Cells were released into S-phase and treated with 5-aza-dC for the indicated time periods, or 30 min in the case of HAP1 cells. In the final 30 min of treatment, EU was added followed by medium chase and fixation as above. Click-iT labelling and analysis was performed as described below.

### EU and EdU detection by Click-iT

For Click-iT labelling, cells were permeabilized with 0.5% Triton X-100 in PBS for 15 min at room temperature (RT) and blocked in 1.5% bovine serum albumin (Thermo Scientific) in PBS for 15 min. Click reactions were performed for 1 h with 60 μM AF488-Azide (Jena Bioscience), 4 mM CuSO$_4$ (Sigma-Aldrich), 10 mM (+)-sodium L-ascorbate (Sigma-Aldrich) and 0.5 μg ml$^{-1}$ 4,6-diamidino-2-phenylindole (Thermo Scientific) in a 50 mM Tris buffer. Coverslips/imaging plates were washed three times with PBS and mounted in Prolong Gold Antifade Mountant (Thermo Scientific; coverslips) or stored in PBS (imaging plates). Images were acquired from coverslips using a Zeiss LSM710 and software ZEN 2009 (Carl Zeiss) version 5.5.0.443 and analysed using CellProfiler. Images from 24-well imaging plates were acquired using

a PerkinElmer Opera Phenix and analysed using Harmony software. For representative images, brightness was adjusted using FIJI (ImageJ) software. For comparisons between cell lines, per-nucleus EU intensity values were normalized to the mean EU intensity of untreated cells. For visualization purposes, graphs were plotted to exclude outlier cells exceeding 200% normalized EU intensity.

### PLA

U2OS GFP–DNMT1 cells in 96-well CellCarrier Ultra (PerkinElmer) imaging plates were synchronized at the G1/S boundary and released into S-phase as described for RRS assays with 5-aza-dC. At 30 min after release into S-phase, cells were treated with 10 μM dC or 5-aza-dC. Then, 1 h later, cells were pre-extracted in ice-cold PBS with 0.2% Triton X-100 on ice for 2 min and fixed with 4% formaldehyde for 15 min. PLA was performed with anti-RPB1 CTD-pS2 and anti-GFP, using the Duolink In Situ PLA Probe Anti-Mouse Minus and Anti-Rabbit Plus probes and the DuoLink In Situ Detection Reagents FarRed Kit (Sigma-Aldrich) as per the manufacturer's instructions. Images were acquired on a Zeiss 880 confocal microscope analysed using CellProfiler.

### RT–qPCR with exogenous RNA spike-in for absolute quantification

Following treatment, cells were collected, counted and lysed in TRIzol reagent (Invitrogen). Exogenous 'spike-in' RNA from the DT40 chicken cell line was added to the lysed cells in proportion to the cell count. Total RNA extraction was then performed with the GENEzol TriRNA Pure Kit (Geneaid). RNA concentration and purity were measured by a NanoDrop 2000c (Thermo Scientific). The complementary DNA was prepared using qScript cDNA Synthesis Kit (QuantaBio, 95047). Expression level of genes was determined by real-time PCR conducted on a Bio-Rad CFX384 system using iTaq Universal SYBR Green Supermix (Bio-Rad). Gene expression was normalized to the chicken transcript cRPL4 for absolute quantification. All primers were validated for specificity and linearity.

### TT$_{chem}$-seq

Nascent RNA-seq (TT$_{chem}$-seq) was performed as previously described[75]. Cells were mock treated or exposed to 1 mM formaldehyde for 1 h. Following drug removal, cells were allowed to recover as indicated and labelled with 1 mM 4SU (Glentham Life Sciences) during the last 30 min. Immediately after 4SU labelling, cells were washed with PBS and lysed in TRIzol reagent. RNA was isolated by TRIzol–chloroform isolation and ethanol precipitation. RNA was rDNAse-treated according to the manufacturer's protocol (Macherey-Nagel) followed by phenol/chloroform isolation and ethanol precipitation. Then, 200 μg RNA (in a total volume of 116 μl) was fragmented by adding 4 μl 5 M NaOH and incubating on ice for 30 min, then stopped by addition of 80 μl of 1 M Tris pH 7 and cleaned up twice using the Micro Bio-Spin P-30 Gel Columns (Bio-Rad, 7326223) according to the manufacturer's instructions. Biotinylation of 4SU residues was performed in a total volume of 250 μl, containing 10 mM Tris–HCl pH 7.4, 1 mM EDTA and 5 mg of MTSEA biotin-XX linker (Biotium, BT90066) for 30 min at RT in the dark. RNA was purified by phenol–chloroform extraction, denatured by 10 min incubation at 65 °C and added to 150 μl μMACS Streptavidin MicroBeads (Milentyl, 130-074-101). RNA was incubated with beads for 15 min at RT and beads were applied to a μColumn in the magnetic field of a μMACS magnetic separator. The beads were washed twice with 55 °C pulldown wash buffer (100 mM Tris–HCl pH 7.4, 10 mM EDTA, 1 M NaCl and 0.1% Tween-20). Biotinylated RNA was eluted twice by addition of 100 mM DTT and cleaned up using the RNeasy MinElute kit (Qiagen, 74204) using 1,050 μl ethanol (≥99%) per 200 μl reaction after addition of 700 μl of RLT buffer to precipitate RNA of less than 200 nucleotides. At least 30 ng of the purified 4SU-labelled RNA was then used as input for the TruSeq Stranded Total RNA kit (Illumina, 20020596) for library preparation. The libraries were amplified according to the

manufacturer's instructions. The library was amplified with ten PCR cycles and quality-control checked on the TapeStation (Agilent) using the High Sensitivity DNA Kit before pooling and paired-end sequencing on the NextSeq 550 (Illumina) system.

A sequencing quality profile was generated using FastQC (version 0.11.9). If needed, sequences were trimmed using TrimGalore (version 0.6.5). Reads were aligned to the human genome 38 using STAR (version 2.7.7a/gcc-8.3.1) with genome GRCh38_no_alt_analysis_set and sjdbGTFfile Homo_sapiens.GRCh38.94.correctedContigs.gtf (https://ftp.ncbi.nlm.nih.gov/genomes/all/GCA/000/001/405/GCA_000001405.15_GRCh38/seqs_for_alignment_pipelines.ucsc_ids/GCA_000001405.15_GRCh38_no_alt_analysis_set.fna.gz). Bam files were converted into stranded TagDirectories and University of California Santa Cruz (UCSC) genome tracks using Hypergeometric Optimization of Motif EnRichment tools (version 4.8.2)[76]. Example genome tracks were generated in Integrative Genomics Viewer (version 2.4.3). A list of 49,948 genes was obtained from the UCSC genome database (https://genome.ucsc.edu/cgi-bin/hgTables) selecting the known canonical table containing the canonical TSSs per gene. To prevent contamination of binding profiles, only genes of at least 3 kb in size and non-overlapping with at least 2 kb between genes (9,944 genes) were included.

TT$_{chem}$-seq aggregated profiles were defined using the Annotate-Peaks.pl tool of Hypergeometric Optimization of Motif EnRichment without normalization of read counts. Genes were selected on size and strongest TT$_{chem}$ signal in the first 3 kb of genes in untreated WT cells. To compare different aggregate profiles, total reads per individual profile were normalized to nascent transcript levels, quantified by EU labelling performed in parallel to the TT$_{chem}$-seq experiment.

## PxP

First, $1 \times 10^6$ cells were seeded in 10-cm dishes and 24 h later synchronized with 1% FBS-containing medium. After 24 h, cells were treated with formaldehyde (1.75 mM, Fisher Scientific). Formaldehyde-treated cells were collected or washed twice with PBS, and recovered in fresh medium for 3 h or 6 h. Cell pellets were frozen at −80 °C and processed the next day. For PxP, cells were washed and resuspended in PBS at $2.5 \times 10^4$ cells μl$^{-1}$. Then, 10 μl of cell suspension was lysed in 1× NuPAGE LDS sample buffer (Thermo Scientific) as input sample for western blot. The remaining cell suspension was pre-warmed for 45 s at 45 °C and mixed with an equal volume of low-melt agarose (2% in PBS, Bio-Rad) and immediately cast into plug molds (1703713, Bio-Rad). Plugs were placed at 4 °C for 5 min, then transferred into 1 ml ice-cold lysis buffer (1× PBS, 0.5 mM EDTA, 2% sarkosyl, cOmplete EDTA-free protease inhibitor cocktail (Merck) and 0.04 mg ml$^{-1}$ Pefabloc SC (Merck)). Cells were lysed on a rotating wheel at 4 °C for 4 h. For electro-elution, plugs were transferred to wells of 10-well SDS−polyacrylamide gel electrophoresis gels (12%, 1.5 mm Novex WedgeWell or BOLT gels, Thermo Fisher). Electrophoresis was performed in 300 ml 3-(N-morpholino)propanesulfonic acid buffer at 20 mA per gel for 60 min in a Mini Gel Tank (Thermo Fisher). Following electro-elution, plugs were retrieved and transferred to tubes containing 1 ml 1×TBS (for PxP−MS) on a rotating wheel at 4 °C for 10 min. This wash was repeated once.

For western blotting, NuPAGE LDS sample buffer was added to plugs after electro-elution, before melting at 99 °C for 25 min.

## PxP sample preparation for MS

For analysis by MS, 100 μl denaturation solution (4 M urea, 100 mM Tris−HCl (pH 7.5), 2 mM DTT and 10 mM chloroacetamide) were added per 100 mg plug in a 1.5 ml tube. After vortexing, samples were incubated at 37 °C for 30 min with agitation (1,500 rpm). Then, 20 μl trypsin solution (2 M urea, 50 mM Tris−HCl (pH 7.5), 1 mM DTT, 5 mM chloroacetamide and 20 μg ml$^{-1}$ trypsin (Sigma-Aldrich)) was added per sample. After overnight incubation at 25 °C with rapid agitation (1,500 rpm), samples were centrifuged at high speed for 10 min.

Supernatants were transferred to fresh tubes. Trifluoroacetic acid (TFA) was added at 1% (final concentration).

## Purification and desalting of peptides on three layers of styrenedivinylbenzene reverse-phase sulfonate StageTips

StageTips were equilibrated by adding 100 μl of 100% acetonitrile (ACN), 100 μl of Solution 2 (30% methanol, 0.2% TFA and 0.2% TFA solution respectively). After the addition of each solution, tips were centrifuged at 600g at RT until no liquid was visible at the tip. Thereafter, samples were loaded into StageTips. Following centrifugation at 600g, samples were sequentially washed with 100 μl isopropanol (twice) and 100 μl 0.2% TFA (twice). After centrifugation at 600g in each washing step, samples were eluted with 60 μl elution buffer (1.25% ammonium hydroxide (NH$_4$OH) and 80% ACN) into PCR tubes. Eluted peptides were dried using a SpeedVac centrifuge (Eppendorf, Concentrator plus) at 45 °C for 25 min and then samples were resuspended in A* Buffer (2% ACN/0.1% TFA).

## LC−MS/MS measurements

LC−MS/MS analysis was performed on the Orbitrap Exploris 480 mass spectrometer coupled to a Thermo Scientific Vanquish Neo ultra high performance liquid chromatography system. First, 200 ng peptide per sample was loaded into a house-packed −50 cm reversed-phase column (75 μm diameter; packed with ReproSil-Pur C18-AQ 1.9 μm resin), and eluted with a gradient starting at 5% buffer B (80% ACN) and increased to 30% in 75 min with a flow rate of 0.300 μl min$^{-1}$, 60% in 5 min and 95% in 10 min. For DIA (data-independent acquisition) measurement, the full scan range was set to 350−1,000 m/z at a resolution of 120,000. The full MS automatic gain control was set to 300% at a maximum injection time of 45 ms. Higher energy collisional dissociation (30%) was used for precursor fragmentation and the resulting fragment ions were analysed in 84 DIA windows at a resolution of 15,000 and an automatic gain control of 1,000. The windows were fixed size with 7.7 m/z width and 1 m/z overlap.

## Raw MS data analysis

Raw files from MS were processed in DIA-NN 1.8.2 beta 22 using an in silico library (N-terminal methionine excision, cysteine carbamidomethylation and one missed cleavage are enabled). Peptide length range was 7−35, and precursor m/z range was 350−1,000. match-between-runs was unchecked. Besides these, default parameters were used.

## Statistical analysis of MS data

Statistical analysis of MS data was performed in R (version 4.2.2). Due to an overall lower number of identified proteins, one replicate (no. 3) out of four was excluded from downstream analysis. Label-free quantitation intensities were log$_2$ transformed and filtered for proteins identified in the three remaining replicates at the 0 h timepoint in both WT and CSB$^{-/-}$ cells. Values were quantile normalized between replicates using the R package preprocessCore (version 1.60.0) and missing values were imputed using the MinDet method from the R package MSnbase (2.24.0)[77]. Differentially abundant proteins were identified using a moderated t-test using the R package limma[78] with Benjamini−Hochberg FDR correction[79]. Proteins with a log$_2$-fold change >1 and a FDR ≤0.01 were considered significant.

## DPC-seq

Cells were seeded in 6-well plates at 160,000 cells per well in 10% FBS-containing media in technical triplicate and 24 h synchronized with 1% FBS-containing media. After 24 h, cells were treated with 1.75 mM formaldehyde for 1 h. Next, cells were washed twice with PBS and either collected immediately or released into drug-free media for 6 h. Cells were scraped in 150 μl 2% SDS, 20 mM Tris−HCl pH 7.5, transferred to 1.5 ml tubes snap frozen in liquid nitrogen and stored at −80 °C. For KCl−SDS precipitation, samples were thawed at 55 °C

for 5 min (1,200 rpm shaking) and sonicated using Covaris Focused ultrasonicator E220evo in 130 μl tubes (microTUBE AFA Fiber Pre-Slit Snap-Cap 6 × 16 mm; 1× cycle and 120 s). Samples were transferred to 1.5 ml tubes and 270 μl 2% SDS, 20 mM Tris–HCl pH 7.5 was added. DNA extraction was performed on 10% of the total lysate using the GeneJET Genomic DNA Purification Kit (Thermo Scientific) and considered 'Input'. For KCl–SDS precipitation, 400 μl of KCl buffer (200 mM KCl and 20 mM Tris–HCl pH 7.5) was added, incubated on ice for 5 min and centrifugated full speed at 4 °C (5 min). Supernatants (soluble DNA) were transferred to fresh tubes for quantification. Pellets (protein and cross-linked protein–DNA complexes) were washed three times according to the following protocol: addition of 400 μl KCl buffer, incubation at 55 °C for 5 min (1,200 rpm shaking), incubation on ice (5 min) and full-speed centrifugation 4 °C (5 min). Next, pellets were resuspended in 400 μl KCl buffer + Proteinase K (0.2 mg ml$^{-1}$) and incubated at 55 °C for 45 min (800 rpm shaking). Then, 10 μl UltraPure bovine serum albumin (Thermo Scientific) was added followed by cooling on ice for 5 min and centrifugation at maximum speed at 4 °C for 5 min. Next, supernatants containing cross-linked DNA were collected. Soluble and cross-linked samples were treated with 0.2 mg ml$^{-1}$ DNAse-free RNAse A (Sigma-Aldrich) for 30 min at 37 °C. DNA concentrations were determined using the Qubit dsDNA HS Assay Kit (Thermo Scientific). Relative DPC amounts between samples were calculated as a ratio of cross-linked DNA to total DNA (cross-linked plus soluble DNA). Then, 50 ng DNA was concentrated via ethanol precipitation with 300 mM sodium acetate, 1 μl glycogen and 2.5× ethanol and resuspended in 20 μl nuclease-free water. DNA was run on 1% agarose gels, stained with SYBR-Gold and the DNA smear of 400–1,000 bp was excised. Gel slices were immersed in 500 μl gel-extraction buffer (10 mM Tris pH 8.0, 1 mM EDTA and 0.02% SDS) and rotated at 4 °C overnight. Gel slices and buffer were loaded into Spin-X columns (Corning Costar, CLS8160) and centrifuged at 14,000g for 10 min at 4 °C. Eluted DNA was ethanol precipitated as before and resuspended in 50 μl nuclease-free water. DNA was then subjected to library preparation via the NEBNext Ultra II DNA library prep kit (NEB, E7645L) using a 1/10 adaptor dilution and seven PCR cycles. Libraries were analysed via a Qubit and Tapestation, pooled at equimolar concentrations and sequenced on an Illumina NovaSeq with PE50 cycles. For all DPC-seq experiments, three biological replicates were performed, each consisting of pooled technical triplicates. A step-by-step protocol for DPC-seq can be found at Nature Protocol Exchange[80].

### DPC-seq analysis
Fastq files were generated using bcl2fastq2 (v2.20), low-quality reads were filtered out using fastp[81] (v0.23.2) and aligned to the hg38 human genome via Bowtie2 (ref. [82]) (v2.4.5). Alignments were sorted and indexed using Samtools[83] (v1.16.1). Read coverages were calculated from alignments using Deeptools[84] (v3.5.0) bamCoverage or bedtools[85] (v2.30.0) coverage with GRCh38 as a reference. Per-gene coverage was normalized to reads per sample and gene length. Further analysis was conducted in R (v4.1.2) using custom scripts. Read coverages were compared via log$_2$-fold change and $t$-tests with Bonferroni correction. All box plots, dot plots and genome track plots were generated using ggplot2 (v3.4.0) whereas metagene line plots and heat maps were generated using Deeptools. Comparisons with other datasets from ATAC-seq (GSE209659) and RNAPII ChIP-seq (GSE141798) was completed by analysing read coverage of these datasets in the same way as with DPC-seq data. All code for upstream processing, downstream analysis and plot generation are available at ref. [86].

### NER excision assay
Chemiluminescent excision assays were performed as previously described[87]. Following treatments, $4.5 \times 10^6$ cells were collected and low-molecular-weight DNA was extracted. Spike-in DNA was added as an internal control (50 fmol of a 50 nt oligomer). High-molecular-weight DNA was removed by reverse size selection with 1.2× PCR clean up beads (MagBio), and samples were subjected to biotinylation with biotin-16-ddUTP (Merck). Samples were separated on 12% urea–polyacrylamide gel, transferred to hybond N+ nylon membrane (Amersham) and biotin-labelled DNA was detected using streptavidin–horseradish peroxidase (Abcam, ab7403) and detected using enhanced chemiluminescence.

### Stringent CUT&Tag
CUT&Tag was performed in biological triplicate as previously described[88] with minor modifications. Per condition, 200,000 untreated, formaldehyde-treated or released cells were resuspended in high salt wash buffer (20 mM HEPES pH 7.5, 600 mM NaCl, 0.5 mM spermidine in nuclease-free water with a Roche Complete Protease Inhibitor EDTA-free tablet) and prepared with concanavalin A-coated magnetic beads (Bangs Laboratories, BP531). Primary antibodies (Supplementary Table 7) were incubated 1:50 overnight at 4 °C. Cells were washed with high salt Dig-wash buffer (0.05% digitonin, 20 mM HEPES pH 7.5, 600 mM NaCl and 0.5 mM spermidine in nuclease-free water with a Roche Complete Protease Inhibitor EDTA-free tablet). The guinea pig anti-rabbit secondary antibody (Supplementary Table 7) was added 1:50 and incubated for 2 h at 25 °C. Cells were washed three times with 300 mM NaCl Dig-wash buffer, and pA-Tn5 tagmentation and subsequent steps were performed in 300 mM NaCl as per the original protocol (Dig-300 buffer; 0.01% digitonin, 20 mM HEPES pH 7.5, 300 mM NaCl and 0.5 mM spermidine in nuclease-free water with a Roche Complete Protease Inhibitor EDTA-free tablet). Libraries were prepared as described previously[88], quantified with the NEBNext Library Quant Kit for Illumina (NEB, 7630S) and sequenced on an Illumina NextSeq 2000 (P3, 100 cycles).

### CUT&Tag data processing
Illumina sequencing paired-end output files were demultiplexed using demux Illumina version 3.0.9 using the flags; -c -d -i -e -t 1 -r 0.01 -R -l 9. Resultant fq.gz files underwent sequencing quality control using FastQC v0.11.8, and their summary was visualized by MultiQC v1.11. Bases with a quality score <20 were trimmed from both reads using cutadapt (cutadapt -q 20). Fastq files were aligned to the combined hg38 and *Escherichia coli* genomes using bwa 0.7.17-r1188 with only reads in the whitelist regions of hg38 continuing the process pipeline. Duplicates were removed using Picard version 2.20.3 (Picard MarkDuplicates). BigWig files were created using deepTools version 3.5.1 bamCoverage using the mapped BAM files and the flags --binSize 3 --normalizeUsing CPM --extendReads, for fragments of up to 600 nt (using −maxFragmentLength). Profile plots were generated using deeptools 3.5.1 on positive-sense non-overlapping genes.

### Statistics and reproducibility
Genome-wide CRISPRi screens were performed with three independent cell populations per conditions. DPC-seq experiments were performed in biological triplicate, where each biological replicate consists of three pooled technical replicates. The TT$_{chem}$-seq experiment was performed once. CUT&Tag experiments were performed in biological triplicate. PxP–MS was performed in four indepent replicates but one was excluded from analysis because fewer peptides overall were detected in the excluded sample. All other experiments were performed a minimum of twice but usually three to four times independently, as indicated in the figure legends. Statistical analysis was performed using two-sided tests as appropriate; details and $P$ values can be found in the legends. For data visualization purposes, in quantification of RRS assays, outlier EU intensity values exceeding 200–210% of the mean were excluded, but raw data including these values can be found in Source data. No statistical method was used to predetermine sample sizes. The experiments were not randomized.

The investigators were not blinded to allocation during experiments and outcome assessment.

## Data availability

All raw and processed data relating to the CRISPRi screens, TT$_{chem}$-seq, DPC-seq and CUT&Tag experiments described in Figs. 1, 4, 6 and 7 and Extended Data Figs. 8 and 9 have been uploaded to ArrayExpress under accession number E-MTAB-12912. As described, DPC-seq data were compared to publicly available RPB1 ChIP-seq and ATAC-seq data accessible through the GEO with the accession numbers GSE141798 (ref. 61) and GSE209659, respectively. The UCSC genome database was used to access Human Genome 38 for read alignments in TT$_{chem}$-seq. The MS proteomics data have been deposited to the ProteomeXchange Consortium via the PRIDE[89] partner repository with the dataset identifier PXD047668. DrugZ analysis outputs from CRISPRi screens shown in Fig. 1 and Extended Data Fig. 1 are provided in Supplementary Tables 1 and 2. All other raw data supporting the findings related to this study are available from the corresponding authors upon reasonable request. Source data are provided with this paper.

## Code availability

All analytical code for both upstream processing and downstream analysis and plot generation of DPC-seq data are publicly available at ref. 86.

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

## Acknowledgements

We thank A. Tufegžić-Vidaković (Medical Research Council Laboratory of Molecular Biology, Cambridge, UK) for the gift of the pGEX3–Dsk2 plasmid and for helpful advice. We thank T. Ogi for the gift of the RPB1–K1268R and *CSB*$^{-/-}$ HeLa cells. We thank K. Harnish and N. Lawrence of the Gurdon Institute Imaging and Scientific Facilities, University of Cambridge for their valuable assistance. We thank K. Dry for editorial assistance. A.C.A., P.W., F.A. and J.C. are supported by the International Max-Planck Research School for Molecular Life Sciences and S.Z. was supported by the Ludwig-Maximilians-Universität–China Scholarship Council Program. Research in the laboratory of J.S. is supported by European Research Council (ERC Starting Grant 801750 DNAProteinCrosslinks), Alfried Krupp Prize for Young University Teachers awarded by Alfried Krupp von Bohlen und Halbach Stiftung, European Molecular Biology Organization (YIP4644), The Vallee Foundation and by the Deutsche Forschungsgemeinschaft (German Research Foundation) (project IDs 213249687–SFB 1064 and 393547839–SFB 1361). Research in the S.P.J. laboratory is funded by Cancer Research UK (CRUK) Discovery Award DRCPGM\100005, CRUK core grant A:29580 and ERC Synergy Award 855741 (DDREAMM). The laboratory was also supported by core funding grants C6946/A24843 and WT203144 to the Gurdon Institute. C.J.C. and M.S.-C. were funded by CRUK Discovery Award DRCPGM\100005; V.G., S.L. and G.D. by ERC Synergy Award 855741; and A.S.B. by CRUK RadNet grant C17918/A28870. S.P.J. receives a salary from the University of Cambridge. M.S.L. was supported by the ERC Consolidator Grant STOP-FIX-GO (grant agreement no. 101043815) and the Netherlands Scientific Organization (Exacte en Natuurwetenschappen grant OCENW.KLEIN.090). S.A. was supported by Israel Cancer Research Fund grant 20-203-RCDA and Israel Science Foundation grant 482/22. S.A. is the recipient of the Jacob and Lena Joels Memorial Foundation Senior Lectureship. Work in the M.S.R. laboratory was supported by the Deutsche Forschungsgemeinschaft (INST 86/1800-1 FUGG, project 428041612 and Project ID 213249687 - SFB 1064) and Munich's Institutional Strategy LMU excellent within the framework of the German Excellence Initiative. Work in the S.B. laboratory was supported by the CRUK core funding (C9681/A29214), programme funding from CRUK (C9545/A19836) and Herchel Smith funds (University of Cambridge). S.B. is a Wellcome Trust Senior Investigator (209441/Z/17/Z). For the purpose of open access, the author has applied a Creative Commons Attribution (CC BY) public copyright licence to any Author Accepted Manuscript version arising from this submission.

## Author contributions

C.J.C., A.C.A., A.S.B., C.E., S.Z., E.B., D.v.d.H., A.P., G.D., M.S.-C., P.W., F.A., M.J.G., J.C., I.E.-G. and A.P.W generated cell lines and performed experiments. A.S.B., V.G., D.v.d.H., M.J.G., L.M. and S.L. performed bioinformatic analyses. S.A. supervised work by E.B., M.S.L. supervised work by D.v.d.H. and A.P.W., S.B. supervised work by I.E.-G. and L.M., and M.S.R. supervised work by F.A. S.P.J. supervised C.J.C., A.S.B., V.G., G.D., M.S.-C. and S.L., and J.S. supervised A.C.A., P.W., M.J.G., J.C. and S.Z. C.J.C., A.C.A., A.S.B., S.P.J. and J.S. wrote the manuscript. C.J.C., S.P.J. and J.S. coordinated and supervised the project.

## Competing interests

S.B. is a founder and shareholder of Biomodal Ltd., Inflex Ltd. and RNAvate Ltd. L.M. is a consultant for Inflex Ltd. S.P.J works part time as Chief Research Officer at Insmed Innovation UK Ltd. S.P.J is a founding partner of Ahren Innovation Capital LLP and is co-founder,

a board member and Chair of the Scientific Advisory Board of Mission Therapeutics Ltd. S.P.J. is a consultant and shareholder of Inflex Ltd. The authors declare no other competing interests.

## Additional information

**Extended data** is available for this paper at https://doi.org/10.1038/s41556-024-01391-1.

**Correspondence and requests for materials** should be addressed to Christopher J. Carnie, Stephen P. Jackson or Julian Stingele.

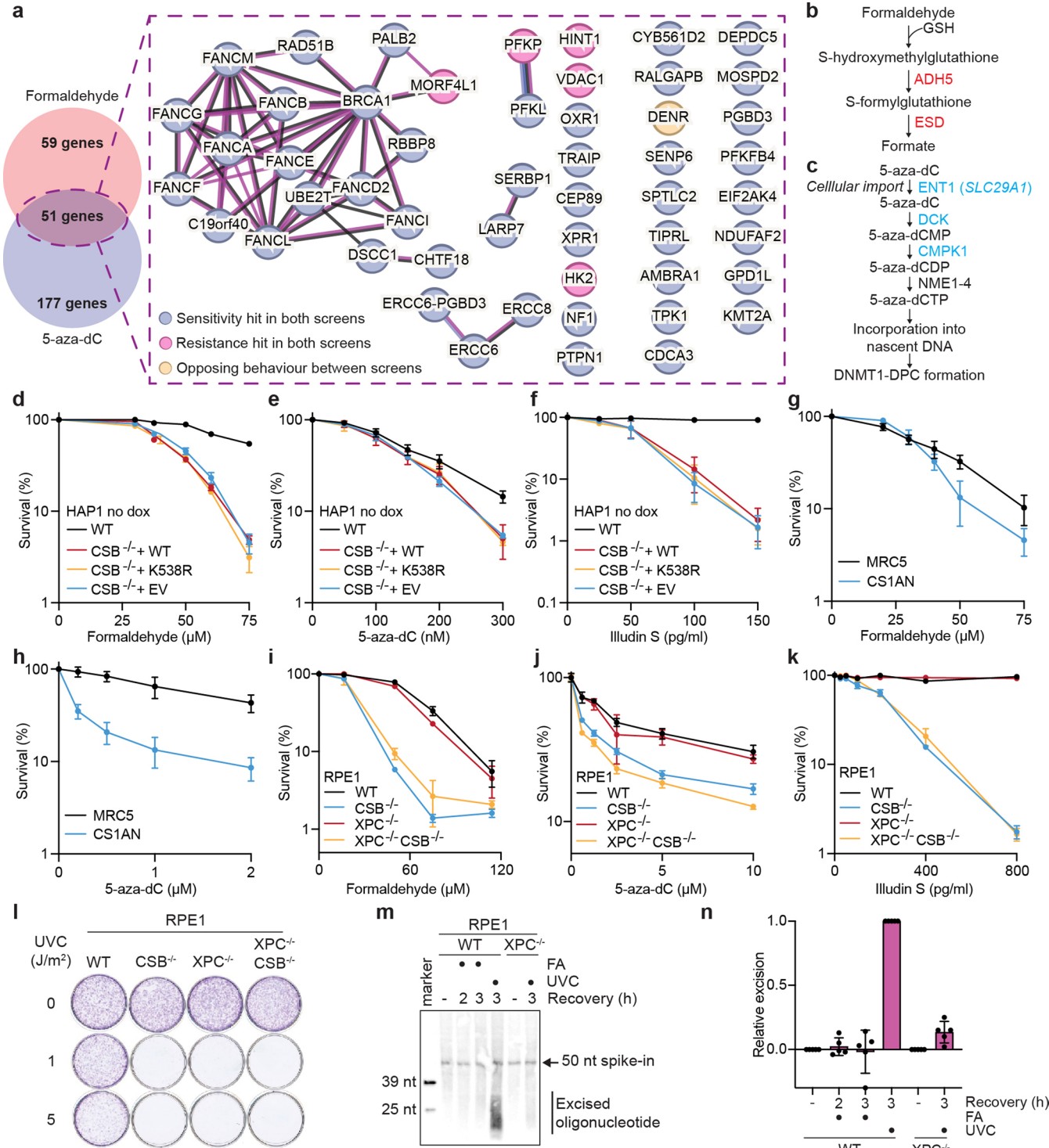

**Extended Data Fig. 1 | *ERCC6*/CSB mediates cellular tolerance of DPCs.**
(**a**) Venn diagram and STRING analysis at medium confidence (cutoff = 0.400) showing common hits (resistance or sensitivity) from formaldehyde and 5-aza-dC CRISPRi screens in K562 cells. (**b**) Schematic depiction of formaldehyde detoxification by ADH5 and ESD. (**c**) Schematic showing 5-aza-dC uptake into cells and subsequent phosphorylation events, mediated by DCK and CMPK1, that are necessary for 5-aza-dC incorporation into nascent DNA. (**d-f**) Clonogenic survival assays in WT or *CSB*$^{-/-}$ TET3G cells in the absence of doxycycline, treated with formaldehyde (d), 5-aza-dC (e) or Illudin S (f); data are presented as mean ± SEM, n = 3 replicates. (**g-h**) Clonogenic survival assays in MRC5 lung

fibroblasts and CSB-deficient fibroblasts (CS1AN) treated with formaldehyde (g) or 5-aza-dC (h); data are presented as mean ± SEM, n = 3 replicates. (**i-k**) Alamar blue viability assays in WT, *CSB*$^{-/-}$, *XPC*$^{-/-}$ or *CSB*$^{-/-}$/*XPC*$^{-/-}$ RPE1 cells treated with formaldehyde (i), 5-aza-dC (j) or Illudin S (k); data are presented as mean ± SD, n = 3 replicates. (**l**) Colony formation assay in the cell lines from (i-k) treated with UVC at the indicated doses. (**m**) Representative oligonucleotide excision assay in WT and *XPC*$^{-/-}$ RPE1 cells treated with formaldehyde (FA) or UVC and released from treatment as indicated. A 50nt oligonucleotide was spiked in as an internal control. (**n**) Quantification of (m); data are presented as mean ± SD, n = 3 replicates. Source numerical data are available in source data.

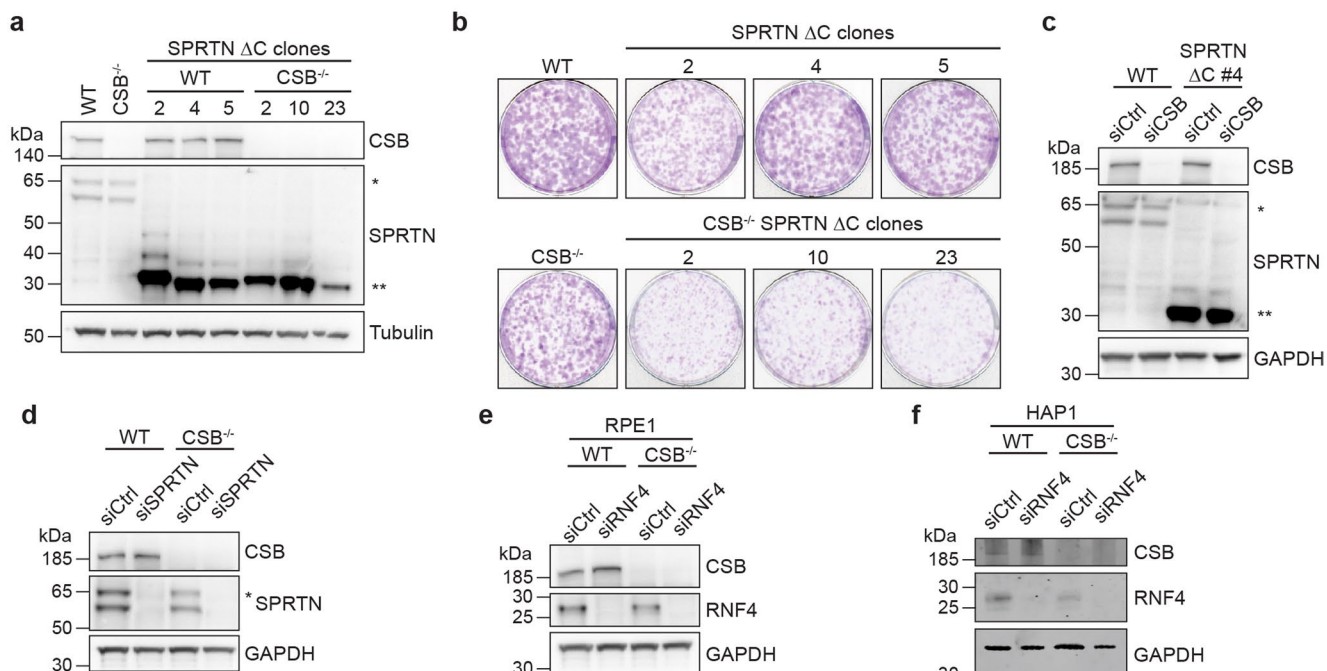

**Extended Data Fig. 2 | CSB acts in parallel to known DPC repair pathways in promoting DPC tolerance.** (**a**) Western blot for CSB and SPRTN in WT, *CSB*[−/−], *SPRTN-ΔC* and *SPRTN-ΔC/CSB*[−/−] RPE1 cell line clones; data are representative of 2 independent experiments. (**b**) Colony formation assay in untreated cell lines from (a). (**c**) siRNA-mediated depletion of CSB in WT and *SPRTN-ΔC* RPE1 cells. (**d**) siRNA-mediated depletion of SPRTN in WT and *CSB*[−/−] RPE1 cells.

(**e-f**) Depletion of RNF4 by siRNA transfection in WT and *CSB*[−/−] RPE1 (e) or HAP1 (f) cells. (c-f) data are representative of 3 independent experiments. (a) and (c): the monoubiquitylated form of SPRTN is denoted by an asterisk (*) and the truncated SPRTN-ΔC protein product is denoted by two asterisks (**). Unprocessed blots are available in source data.

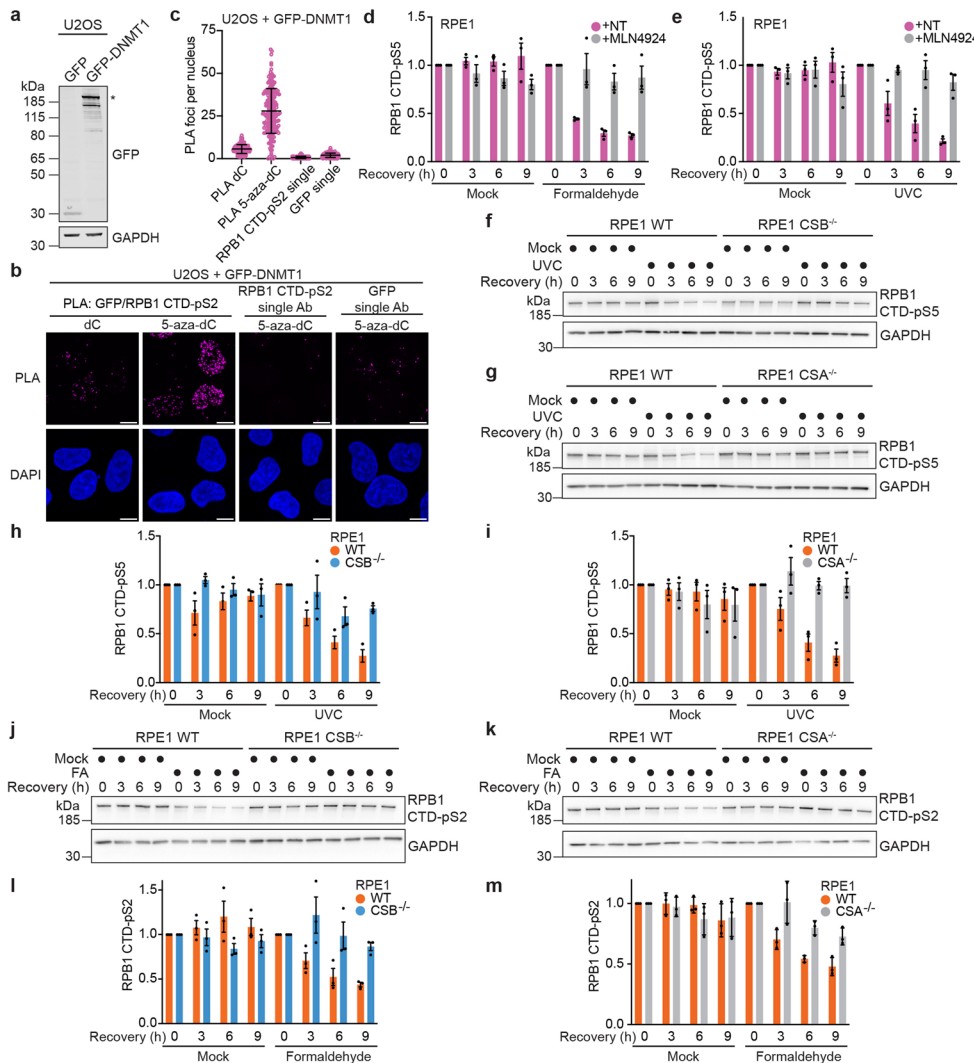

**Extended Data Fig. 3 | CSB initiates a pathway that supports transcription recovery following DPC induction.** (**a**) Constitutive expression of GFP-DNMT1 compared to GFP in U2OS cells; full length GFP-DNMT1 is denoted by an asterisk (*). (**b**) Representative images from Proximity ligation assays (PLA) between GFP and RPB1 CTD-pS2 in U2OS cells constitutively expressing GFP-DNMT1 following release from thymidine block into 10 μM dC or 5-aza-dC for 1 h; scale bars = 10 μm. (**c**) Quantification of (b); data presented as mean ± SD, n = 3 replicates. (**d-e**) Quantification of Fig. 3e, f; degradation of RPB1 in RPE1 cells treated with FA (d) or UVC (e) in the presence or absence of MLN4924; data presented as mean ± SEM, n = 3 replicates. (**f-g**) RPB1 degradation in cycloheximide-treated

WT and $CSB^{-/-}$ (f) or $CSA^{-/-}$ (g) RPE1 cells at the indicated time points after a UVC treatment. (**h-i**) Quantification of (f-g), respectively; data presented as mean ± SEM, n = 3 replicates. (**j-k**) RPB1 degradation in cycloheximide-treated WT and $CSB^{-/-}$ (j) or $CSA^{-/-}$ (k) cells treated with formaldehyde for the indicated time points. For (j) and (k), GAPDH blot images are also shown alongside blots in Fig. 3g, h, respectively, due to detection of RPB1 CTD-pS2 and RPB1 CTD-pS5 from the same experiment. (**l-m**) Quantification of (j) and (k), respectively; data presented as mean ± SEM, n = 3 replicates. Source numerical data and unprocessed blots are available in source data.

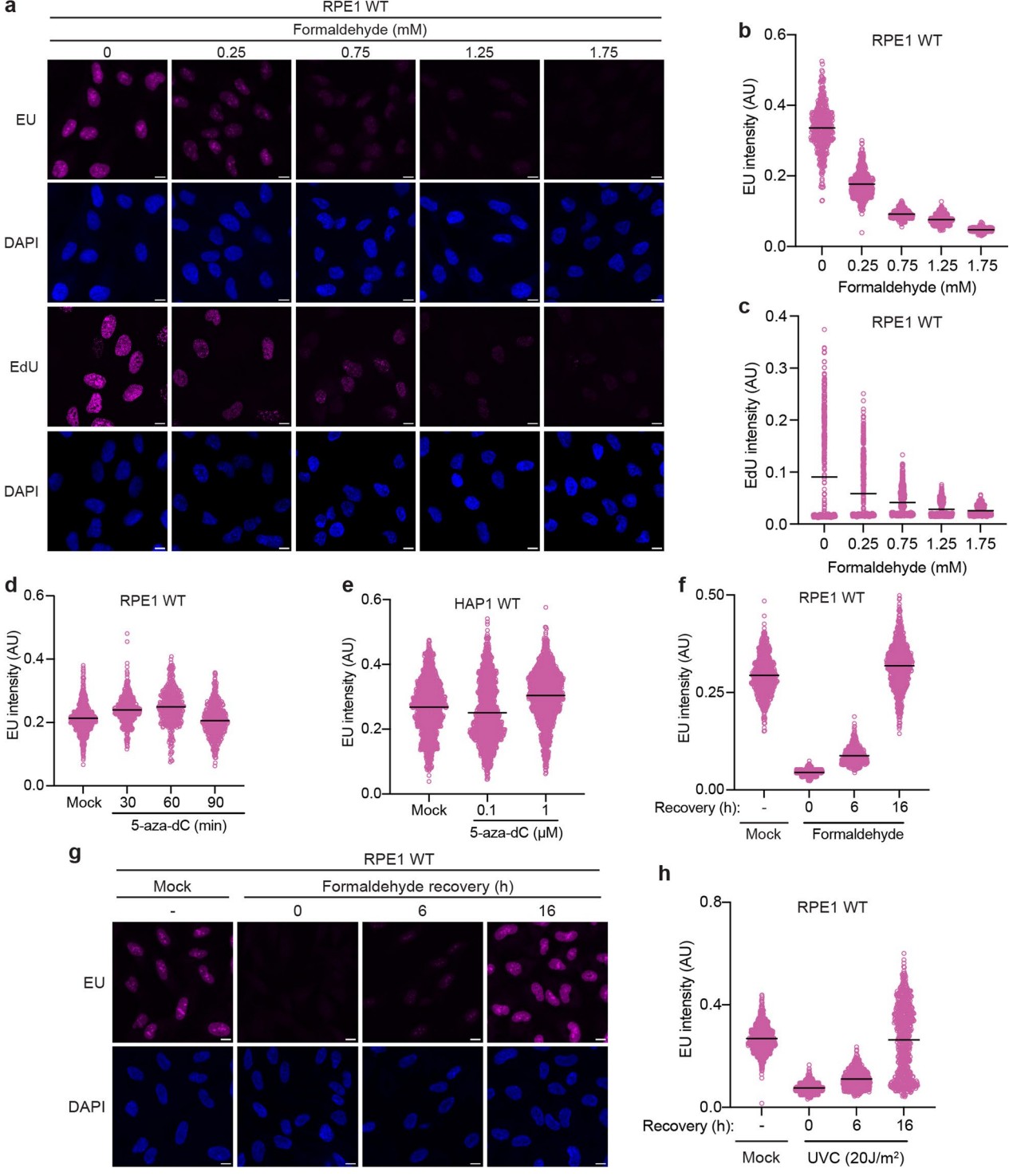

**Extended Data Fig. 4 | Formaldehyde induces global transcription arrest.**
(**a**) Representative images showing formaldehyde-induced inhibition of
transcription (EU incorporation) and DNA replication (EdU incorporation) in
WT RPE1 cells; scale bars = 10 µm. (**b-c**) Quantification of (a) for EU (b) and EdU
(c). (**d-e**) RRS assays in RPE1 (d) or HAP1 (e) cells treated with 5-aza-dC for the
indicated times. (**f-g**) Quantification (f) and representative images (**g**) of RRS
assays in RPE1 cells treated with formaldehyde and allowed to recover for the
indicated times; scale bars = 10 µm. (**h**) RRS assay in RPE1 cells treated with UVC
and allowed to recover for the indicated times. For (b-f) and (h), data are shown
from one biological replicate, representative of 2 (e) or 3 (b-d, f, h) independent
experiments with the mean value shown in black. Source numerical data are
available in source data.

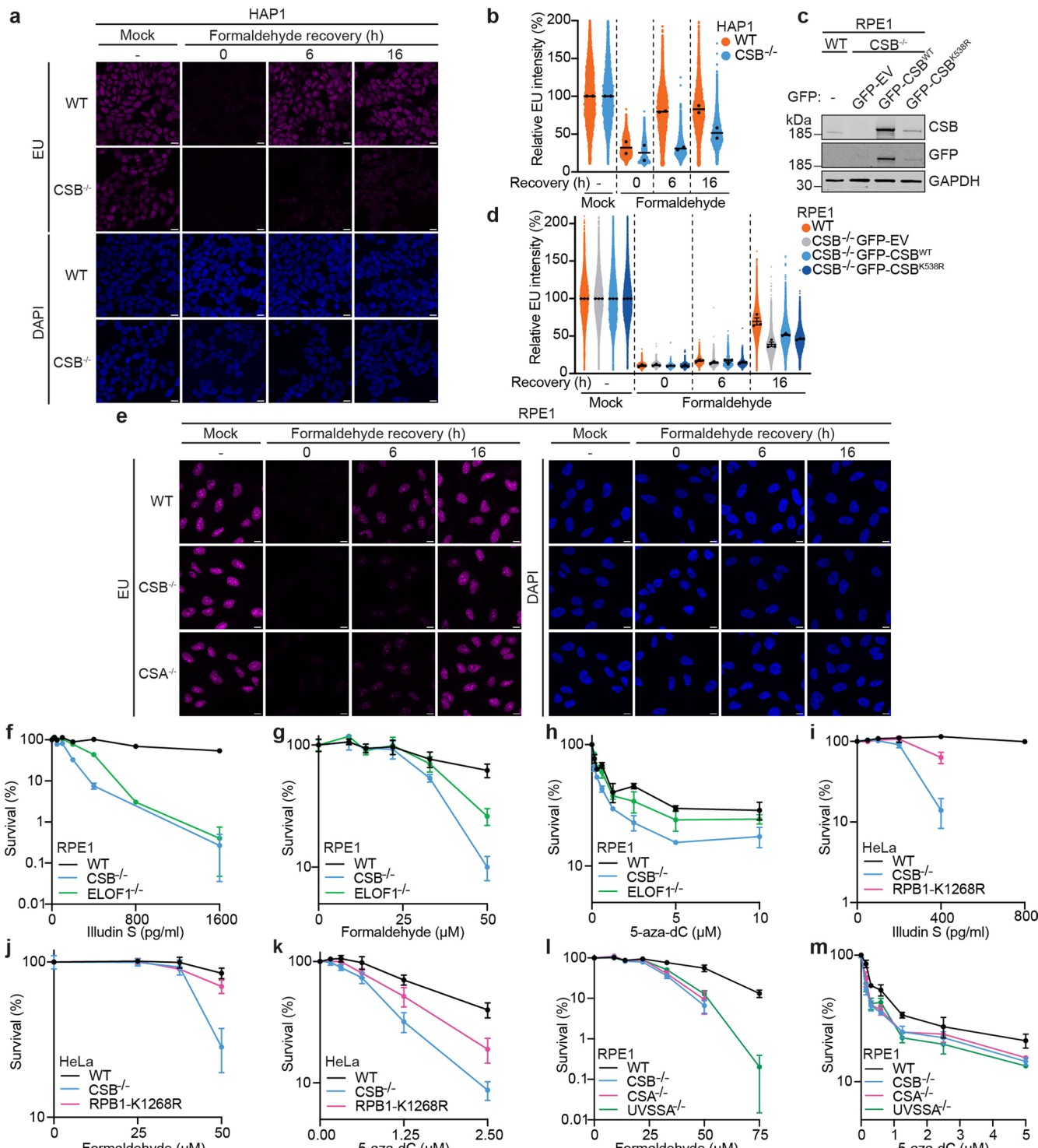

**Extended Data Fig. 5 | Upstream TC-NER factors support transcription recovery after DPC induction.** (**a-b**) Representative images (a) and quantification (b) from RRS assays in WT and $CSB^{-/-}$ HAP1 cells treated with formaldehyde and released for the indicated times; scale bars = 10 μm. Mean intensities of replicates are displayed as black dots with a mean of those averages shown; n = 2 replicates. (**c**) Western blot analysis of CSB and GFP-CSB expression in the indicated RPE1 cell lines. (**d**) RRS assays in cell lines from (c) treated with formaldehyde and released for the indicated times. Mean intensities of replicates are displayed as black dots with a mean of those averages shown as line; n = 3 replicates, error bars ± SEM. (**e**) Representative images from RRS assays in WT,

$CSB^{-/-}$ and $CSA^{-/-}$ RPE1 cells treated with formaldehyde and released for the indicated times; scale bars = 10 μm. (**f-h**) Alamar blue cell viability assays in WT, $CSB^{-/-}$ or $ELOF1^{-/-}$ RPE1 cells treated with Illudin S (f), formaldehyde (g) or 5-aza-dC (h) at the indicated doses; data presented as mean ± SD, n = 3 replicates. (**i-k**) Alamar blue viabilities as in (f-h) but with WT, $CSB^{-/-}$ and RPB1-K1268R HeLa cells; data presented as mean ± SD, n = 3 replicates. (**l-m**) Alamar blue assays in WT, $CSB^{-/-}$, $CSA^{-/-}$ and $UVSSA^{-/-}$ RPE1 cells treated with formaldehyde (FA) (c) or 5-aza-dC (d); data presented as mean ± SD, n = 3 replicates. Uncropped western blot images are provided in the source data. Source numerical data and unprocessed blots are available in source data.

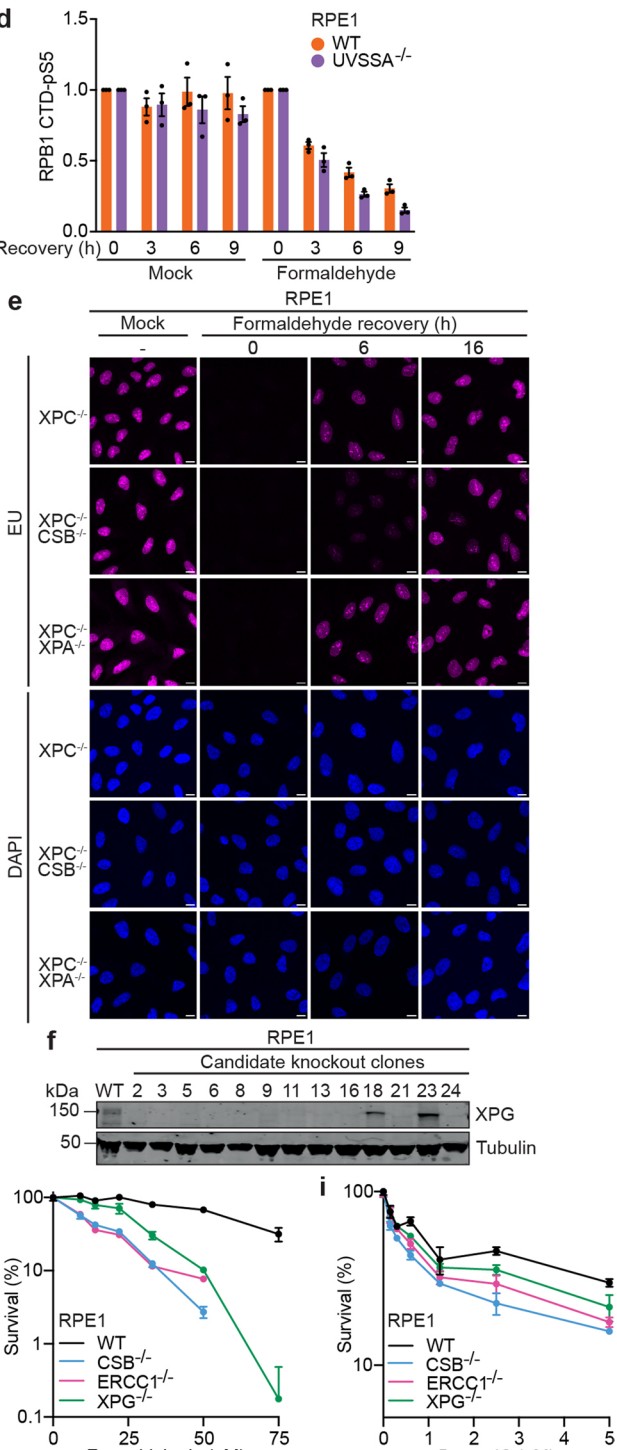

**Extended Data Fig. 6 | Downstream NER factors are dispensable for transcription recovery after DPC induction.** (**a**) Representative images from RRS assays in WT, *CSB*[−/−] and *UVSSA*[−/−] RPE1 cells treated with formaldehyde; scale bars = 10 μm, images representative of 3 independent experiments. (**b**) Dsk2 pulldown in G1-synchronised WT, *CSB*[−/−] and *UVSSA*[−/−] RPE1 cells treated with formaldehyde for the indicated times; data representative of 3 independent experiments. (**c**) Formaldehyde-induced RPB1 degradation in cycloheximide-treated WT and *UVSSA*[−/−] cells released from FA treatment for the indicated times. (**d**) Quantification of (**c**); error bars ± SEM, n = 3 replicates. (**e**) Representative

images from RRS assays in *XPC*[−/−], *XPC*[−/−]/*CSB*[−/−] and *XPC*[−/−]/*XPA*[−/−] RPE1 cells treated with formaldehyde; scale bars = 10 μm. (**f**) Western blot analysis of candidate XPG knockout RPE1 clones. (**g**) Sanger sequencing result from genotyping of RPE1 *XPG*[−/−] clone 21 used following PCR amplification of the targeted gDNA sequence. (**h-i**) Alamar blue assays in WT, *CSB*[−/−], *ERCC1*[−/−] and *XPG*[−/−] RPE1 cells treated with formaldehyde (**e**) or 5-aza-dC (**f**); data presented as mean ± SD, n = 3 replicates. Source numerical data and unprocessed blots are available in source data.

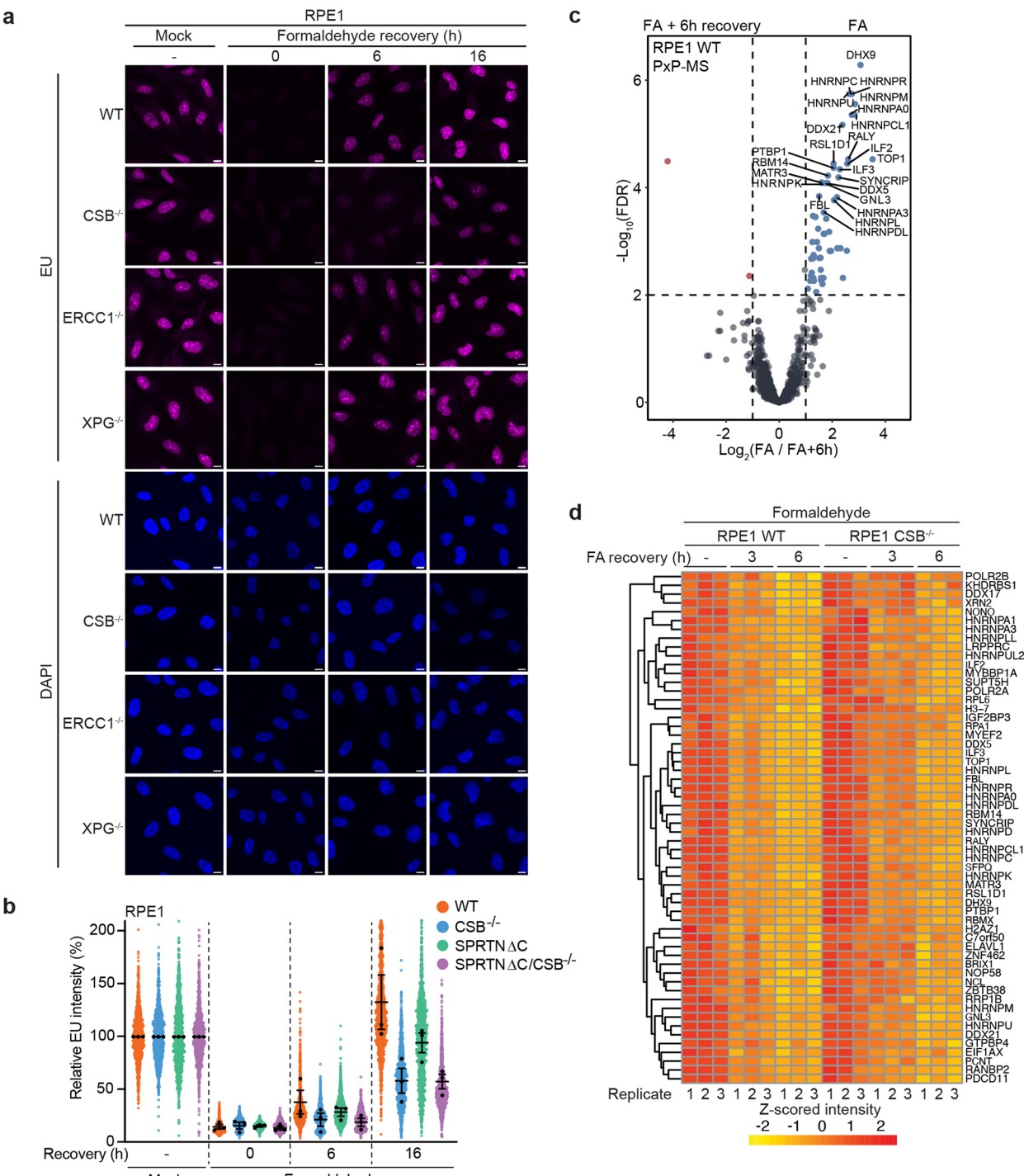

**Extended Data Fig. 7 | Global DPC induction and repair is not affected by CSB loss.** (**a**) Representative images from RRS assays in WT, *CSB*−/−, *ERCC1*−/− and *XPG*−/− RPE1 cells treated with formaldehyde, representative of 3 independent experiments. (**b**) RRS assays in WT *SPRTN-ΔC*, *CSB*−/− and *SPRTN-ΔC/CSB*−/− RPE1 cells treated with formaldehyde and released for the indicated times. Mean intensities of replicates are displayed as black dots with a mean of those averages shown as line; n = 3 replicates error bars ± SEM. (**c**) Volcano plot from PxP-MS

in WT RPE1 cells treated with formaldehyde compared with cells treated with formaldehyde and released for 6 h. (**d**) Heat map displaying Z-scored intensities for DNA-crosslinked proteins identified by PxP-MS after formaldehyde treatment and release for the indicated times in WT and *CSB*−/− RPE1 cells. Intensities are displayed for three replicates per condition. Source numerical data are available in source data.

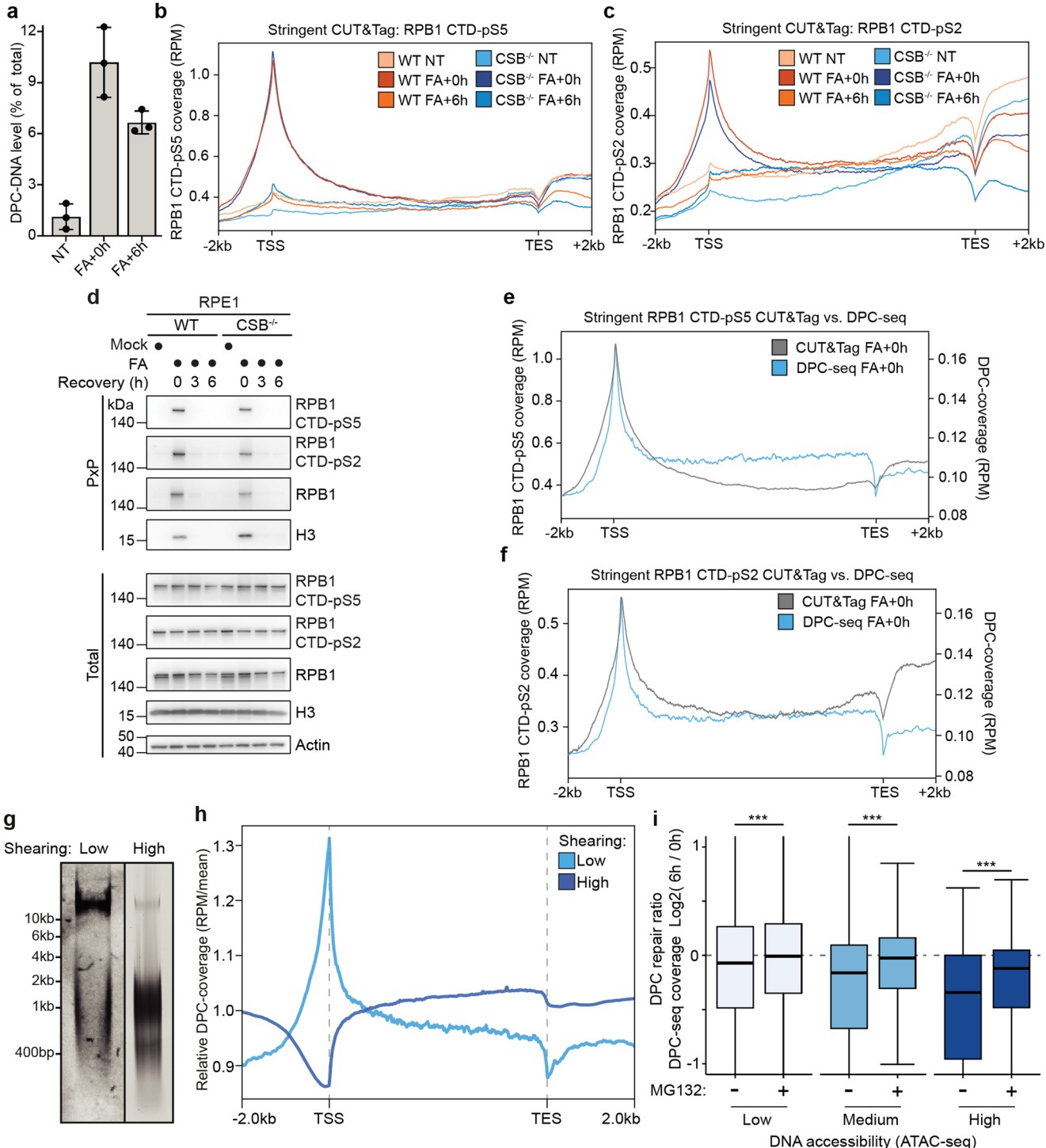

**Extended Data Fig. 8 | Formaldehyde induces RNAPII-DPCs at transcription start sites.** (**a**) Enriched DPC-DNA quantification by Qubit as a % of total DNA per sample in either untreated RPE1 cells, treated with 1.75 mM formaldehyde for 1 h or treated and then recovered for 6 h, data presented as mean ± SD, n = 3 replicates. (**b-c**) Metagene profiles from stringent CUT&Tag with antibodies against RPB CTD-pS5 (b) and -pS2 (c) in WT or *CSB*⁻/⁻ RPE1 cells treated with formaldehyde, or treated and then released for 6 hours; n = 3 biological replicates. (**d**) PxP-WB with the indicated antibodies in WT and *CSB*⁻/⁻ RPE1 cells treated with formaldehyde and released for the indicated times; data representative of 3 independent experiments. (**e-f**) Overlaid metagene profiles from formaldehyde-treated WT RPE1 cells subjected to DPC-seq or stringent CUT&Tag against RPB1 CTD-pS5 (e) or -pS2 (f); n = 3 biological replicates for each approach. (**g**) Representative gel image of low (left lane) and high (right

lane) shearing in DPC-seq sample preparation. (**h**) Metagene profiles of DPC-seq after formaldehyde treatment in RPE1 cells from samples in (g); n = 3 biological replicates. (**i**) Log2 fold change of DPC-seq coverage per gene 6 h/0 h after FA treatment, with or without MG132 treatment, in genes with low, medium or high DNA accessibility as determined by ATAC-seq (GEO: GSE209659), n = 3 biological replicates, statistics via two-sided paired Wilcoxon test *** p < 0.001; p-values are <2.2 × 10⁻¹⁶, <2.2 × 10⁻¹⁶ and <2.2 × 10⁻¹⁶ for comparisons in low, medium and high DNA accessibility, respectively. Box-plot shows upper (Q3) and lower (Q1) quartile boundaries and line at the median. Lower whisker (minimum) = Q1 − 1.5 x interquartile range (IQR), upper whisker (maximum) = Q3 + 1.5 x IQR. For all DPC-seq analyses, n = 3 biological replicates. Source numerical data and unprocessed blots are available in source data.

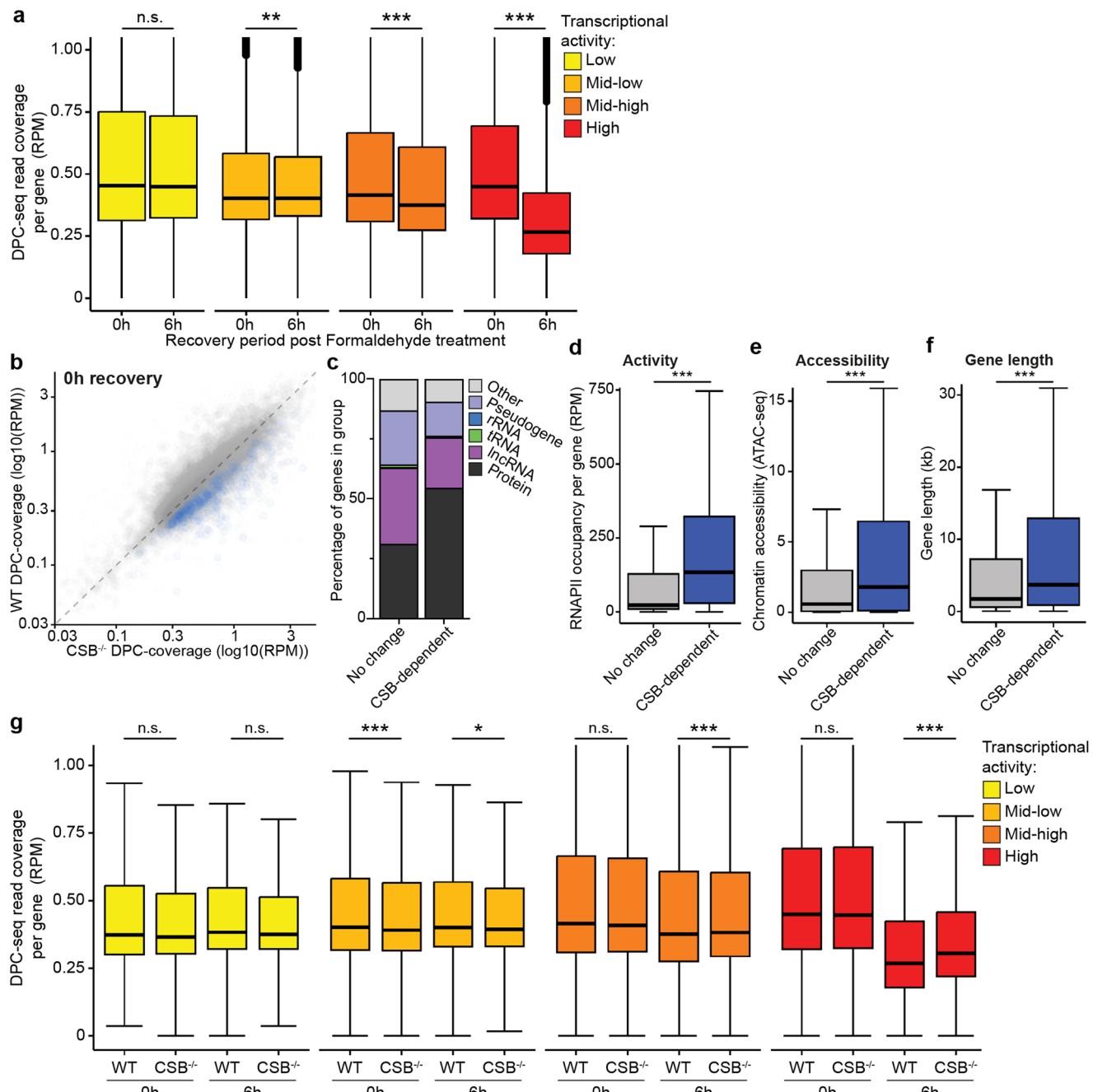

**Extended Data Fig. 9 | CSB is required for the repair of DPCs at transcriptionally active loci.** (**a**) DPC-seq coverage per gene, with or without 6 h recovery after treatment, in genes grouped by transcriptional activity, as determined by RNAPII ChIP-seq (GEO: GSE141798)[59], statistics via two-sided paired Wilcoxon test ** p < 0.01 *** p < 0.001; p-values are 0.1458, 0.008734, <2.2 × 10[−16] and <2.2 × 10[−16] for comparisons in low, mid-low, mid-high and high transcriptional activity gene sets, respectively. (**b**) DPC-seq coverage per gene in WT RPE1 cells vs *CSB*[−/−] cells immediately (0 h) after formaldehyde treatment, blue genes show significantly enriched DPC coverage in *CSB*[−/−] cells. (**c**) Percentage of different gene types in genes that are dependent on CSB for DPC repair or where repair is not significantly changing based on CSB status. (**d**) RNAPII occupancy of genes that are dependent or independent on CSB for DPC repair, statistics via two-sided unpaired Wilcoxon test *** p < 0.001; p < 2.2 × 10[−16].

(**e**) Same as (d) but for DNA accessibility; p < 2.2 × 10[−16]. (**f**) Same as (d) but for gene length; p = 7.457 × 10[−11]. (**g**) Box-plot of DPC-seq coverage per gene in WT and *CSB*[−/−] cells with 1 h 1.75 mM formaldehyde treatment with or without 6 h recovery split into quartiles of transcriptional activity, statistics via two-sided Dunn test (paired) * p < 0.05 *** p < 0.001; p-values are 0.2265 and 0.05998 (low transcriptional activity, 0 h and 6 h respectively), 0.0007959 and 0.03377 (mid-low transcriptional activity, 0 h and 6 h respectively), 0.5608 and 2.466 × 10[−10] (mid-high transcriptional activity, 0 h and 6 h respectively), 0.2227 and <2.2 × 10[−16] (high transcriptional activity, 0 h and 6 h respectively). (a, d-g) Box-plot shows upper (Q3) and lower (Q1) quartile boundaries and line at the median. Lower whisker (minimum) = Q1 − 1.5 x interquartile range (IQR), upper whisker (maximum) = Q3 + 1.5 x IQR. For all DPC-seq analyses, n = 3 biological replicates. Source numerical data are available in source data.

# Reporting Summary

## Statistics

For all statistical analyses, confirm that the following items are present in the figure legend, table legend, main text, or Methods section.

| n/a | Confirmed | |
|---|---|---|
| ☐ | ☒ | The exact sample size (*n*) for each experimental group/condition, given as a discrete number and unit of measurement |
| ☐ | ☒ | A statement on whether measurements were taken from distinct samples or whether the same sample was measured repeatedly |
| ☐ | ☒ | The statistical test(s) used AND whether they are one- or two-sided<br>*Only common tests should be described solely by name; describe more complex techniques in the Methods section.* |
| ☐ | ☒ | A description of all covariates tested |
| ☐ | ☒ | A description of any assumptions or corrections, such as tests of normality and adjustment for multiple comparisons |
| ☐ | ☒ | A full description of the statistical parameters including central tendency (e.g. means) or other basic estimates (e.g. regression coefficient) AND variation (e.g. standard deviation) or associated estimates of uncertainty (e.g. confidence intervals) |
| ☐ | ☒ | For null hypothesis testing, the test statistic (e.g. $F$, $t$, $r$) with confidence intervals, effect sizes, degrees of freedom and $P$ value noted<br>*Give P values as exact values whenever suitable.* |
| ☒ | ☐ | For Bayesian analysis, information on the choice of priors and Markov chain Monte Carlo settings |
| ☒ | ☐ | For hierarchical and complex designs, identification of the appropriate level for tests and full reporting of outcomes |
| ☒ | ☐ | Estimates of effect sizes (e.g. Cohen's *d*, Pearson's *r*), indicating how they were calculated |

*Our web collection on statistics for biologists contains articles on many of the points above.*

## Software and code

Policy information about availability of computer code

| Data collection | Descriptions of the software used to collect data relating to this study are included throughout the Methods section. |
|---|---|
| Data analysis | Descriptions of the code used to analyse data as part of this study are included throughout the Methods section. For data presentation and analysis purposes, the following software packages were used: Harmony 5.2, ImageJ2 2.9.0/1.53t, CellProfiler 4.2.5, FastQC 0.11.8 and 0.11.9, TrimGalore 0.6.5, STAR 2.7.7a/gcc-8.3.1, HOMER tools 4.8.2, IGV 2.4.3, DIA-NN 1.8.2 beta 22, R 4.2.2 and 4.1.2, preprocessCore 1.60.0, MSnbase 2.24.0 MinDet method, limma, fastp 0.23.2, Bowtie2 2.4.5, Samtool 1.16.1, Deeptools 3.5.0 and 3.5.1, bedtools 2.30.0, bamCoverage, ggplot2 3.4.0, demux Illumina 3.0.9, MultiQC 1.11, cutadapt, Picard 2.20.3. As stated in the Code Availability statement, all analytical code for both upstream processing and downstream analysis and plot generation of DPC-seq data are publicly available at https://github.com/aldob/DPC-Seq. |

For manuscripts utilizing custom algorithms or software that are central to the research but not yet described in published literature, software must be made available to editors and reviewers. We strongly encourage code deposition in a community repository (e.g. GitHub). See the Nature Portfolio guidelines for submitting code & software for further information.

# Data

Policy information about availability of data

All manuscripts must include a data availability statement. This statement should provide the following information, where applicable:
- Accession codes, unique identifiers, or web links for publicly available datasets
- A description of any restrictions on data availability
- For clinical datasets or third party data, please ensure that the statement adheres to our policy

All raw and processed data relating to the CRISPRi screens, TTchem-seq, DPC-seq and CUT&Tag experiments described in Figures 1, 4, 6 and 7 and Extended Data Figures 8 and 9 have been uploaded to ArrayExpress under accession number E-MTAB-12912. As described, DPC-seq data were compared to publicly available RPB1 ChIP-seq and ATAC-seq data accessible through the GEO with the accession numbers GSE14179861 and GSE209659, respectively. The UCSC genome database was used to access Human Genome 38 for read alignments in TTchem-seq. The mass spectrometry proteomics data have been deposited to the ProteomeXchange Consortium via the PRIDE85 partner repository with the dataset identifier PXD047668. DrugZ analysis outputs from CRISPRi screens shown in Fig. 1 and Extended Data Fig. 1 is provided in Supplementary Tables 1-2. Source data has been provided in Source Data. All other raw data supporting the findings related to this study are available from the corresponding authors upon reasonable request.

# Human research participants

Policy information about studies involving human research participants and Sex and Gender in Research.

| | |
|---|---|
| Reporting on sex and gender | n/a |
| Population characteristics | n/a |
| Recruitment | n/a |
| Ethics oversight | n/a |

Note that full information on the approval of the study protocol must also be provided in the manuscript.

# Field-specific reporting

Please select the one below that is the best fit for your research. If you are not sure, read the appropriate sections before making your selection.

☒ Life sciences  ☐ Behavioural & social sciences  ☐ Ecological, evolutionary & environmental sciences

For a reference copy of the document with all sections, see nature.com/documents/nr-reporting-summary-flat.pdf

# Life sciences study design

All studies must disclose on these points even when the disclosure is negative.

| | |
|---|---|
| Sample size | No sample size calculation was performed prior to any experiments. Most experiments shown that were aimed at answering new questions relating to the study were performed at least three times independently. Sample sizes were chosen based on technical difficulty and robustness of method, informed by both literature and the authors' experience. Where possible, aggregated data from replicate experiments have been shown in the figures. In other cases, findings from representative experiments have been shown. In the case of 'control' experiments aimed at validating known phenotypes (such as transcription recovery defects after UVC treatment in in NER-deficient cell lines), two replicate experiments were deemed sufficient to confirm these phenotypes. |
| Data exclusions | In RRS assays, normalised EU intensity values exceeding 200% were considered outliers and excluded from visualisation but not from subsequent calculation of relevant means. These assays aimed at identifying recovery towards 100% after conditions that brought values down to around or below 10%. The outliers above 200% were considered likely to largely arise from microscopy artefacts and were therefore excluded to enable proper visualisation and interpretation of the actual window of biological events. For PxP-MS data analysis, one replicate (#3) out of four was excluded from downstream analysis, due to an overall lower number of identified proteins. |
| Replication | This study has benefited from the collaboration of two independent research groups with supporting collaborations with several others. Between the two primary groups, the phenotypes representing the key findings (RRS assays and cell viability assays) of the study were independently reproducible between both groups. These approaches were repeated at least 3 times independently between the two main groups, and in some cases cross-validated in additional cell lines (RPE1 and HAP1). |
| Randomization | No randomization of samples was relevant to the study since all experiments were internally controlled. |
| Blinding | Blinding was not relevant to the study because as outlined in the Methods section, all quantification and analyses of cell survival and microscopy-based assays were performed using software to apply identical image processing and analysis parameters across samples within each experiment. |

# Reporting for specific materials, systems and methods

We require information from authors about some types of materials, experimental systems and methods used in many studies. Here, indicate whether each material, system or method listed is relevant to your study. If you are not sure if a list item applies to your research, read the appropriate section before selecting a response.

## Materials & experimental systems

| n/a | Involved in the study |
|-----|-----------------------|
| ☐ | ☒ Antibodies |
| ☐ | ☒ Eukaryotic cell lines |
| ☒ | ☐ Palaeontology and archaeology |
| ☒ | ☐ Animals and other organisms |
| ☒ | ☐ Clinical data |
| ☒ | ☐ Dual use research of concern |

## Methods

| n/a | Involved in the study |
|-----|-----------------------|
| ☒ | ☐ ChIP-seq |
| ☒ | ☐ Flow cytometry |
| ☒ | ☐ MRI-based neuroimaging |

## Antibodies

| Antibodies used | Details of antibodies used can be found in Supplementary Table 7. Rat anti-SPRTN (6F2) is a custom antibody generated at LMU Munich. To request this antibody, please contact Julian Stingele. |
|---|---|
| Validation | The following antibodies were validated by western blot in knockout cells or with siRNA-mediated depletion: anti-CSB (Abcam ab96089), anti-CSA (Abcam ab137033), anti-SPRTN (Custom, produced at LMU Munich, see Zhao et al (2021) Nucleic Acids Research), anti-RNF4 (R&D Systems AF7964 and Proteintech 17810-1-AP), anti-XPG (Bethyl A301-484A-2). |
| | The following antibodies were not validated by us but behaved as expected (based on available literature) and are commonly used in Western blots: anti RNAPII-CTDpSer5 4H8 (Abcam ab5408), Anti-RNAPII-CTDpSer2 3E10 (Millipore 04-1571), Anti-RNAPII-pCTDpSer2 (Novus BiologicalsNB100-1805), anti-RNAPII-NTD (Cell Signaling 14958), anti-GAPDH 14C10 (Cell Signaling 2118), anti-GFP (Roche 11814460001), anti-ATF3 (Abcam ab207434), anti-Tubulin (Sigma-Aldrich T6074 and T6199). |
| | The following antibodies were used for immunofluorescnce and/or PLA experiments, consistent with prior use in the literature and within the manufacturers' validation statements: anti-GFP (Roche 11814460001), Anti-RNAPII-CTDpSer2 (Novus Biologicals NB100-1805), anti-NPM1 (Abcam ab10530). |
| | The following antibody was validated for use in immunoprecipitation, in agreement with previous literature and the manufacturer's website: anti-RNAPII-CTDpSer2 (Abcam ab5095). |
| | The following antibodies were validated for use in CUT&Tag experiments: anti-RNAPII-CTDpSer5 (Cell Signaling 13523), anti-RNAPII-CTDpSer2 (Cell Signaling 13499). |

## Eukaryotic cell lines

Policy information about cell lines and Sex and Gender in Research

| Cell line source(s) | K-562 cells are human chronic myeloid leukaemia (CML) cells from a female patient, HAP1 cells are derived from KBM-7 cells, which are CML cells from a human male. RPE-1 cells are human female cells derived from the retinal pigment epithelium, MRC5 cells are lung fibroblasts derived from a human male, CS1AN cells are human skin fibroblasts from a female with Cockayne syndrome, and U2OS cells are osteosarcoma cells from a human female. HeLa cells are cervical cancer cells from a human female. Individual cell line sources are as follows: |
|---|---|
| | K562 dCas9-KRAB - Kind gift from Jonathan Weissman |
| | HAP1 WT -  Horizon Discovery (Cat no. C631) |
| | HAP1 CSB-/- - Horizon Discovery (Cat no. HZGHC000422c011) |
| | HAP1 XPA-/- - Horizon Discovery (Cat no. HZGHC000433c001) |
| | HAP1 CSB-/- +EV TET3G - This study |
| | HAP1 CSB-/- +CSB-WT TET3G - This study |
| | HAP1 CSB-/- +CSB-K538R TET3G - This study |
| | RPE1-TetOn-Cas9-PuroS-TP53-/- ('WT') - Van der Weegen et al, NCB 2021 |
| | RPE1-TetOn-Cas9-PuroS-TP53-/--CSB-/- ('CSB-/-') - Van der Weegen et al, NCB 2021 |
| | RPE1-TetOn-Cas9-PuroS-TP53-/--XPC-/- ('XPC-/-') - Van der Weegen et al, NCB 2021 |
| | RPE1-TetOn-Cas9-PuroS-TP53-/--XPC-/--XPA-/- ('XPC-/-XPA-/-') - Van den Heuvel et al, PNAS 2023 |
| | RPE1-TetOn-Cas9-PuroS-TP53-/--XPC-/--CSB-/- ('XPC-/-CSB-/-') - Van den Heuvel et al, PNAS 2023 |
| | RPE1-TetOn-Cas9-PuroS-TP53-/--SPRTN ΔC clone 2  ('SPRTN ΔC#2') - This study |
| | RPE1-TetOn-Cas9-PuroS-TP53-/--SPRTN ΔC clone 4  ('SPRTN ΔC#4') - This study |
| | RPE1-TetOn-Cas9-PuroS-TP53-/--SPRTN ΔC clone 5  ('SPRTN ΔC #5') - This study |
| | RPE1-TetOn-Cas9-PuroS-TP53-/--CSB-/-SPRTN ΔC clone 2  ('SPRTN ΔC#2') - This study |
| | RPE1-TetOn-Cas9-PuroS-TP53-/--CSB-/-SPRTN ΔC clone 10  ('SPRTN ΔC#10') - This study |
| | RPE1-TetOn-Cas9-PuroS-TP53-/--CSB-/-SPRTN ΔC clone 23  ('SPRTN ΔC #23') - This study |
| | RPE1-TetOn-Cas9-PuroS-TP53-/--ELOF1-/- ('ELOF1-/-') - Van der Weegen et al, NCB 2021 |

RPE1-TetOn-Cas9-PuroS-TP53-/-CSA-/- ('CSA-/-') - Van der Weegen et al, NCB 2021
RPE1-TetOn-Cas9-PuroS-TP53-/-UVSSA-/- ('UVSSA-/-') - Van der Weegen et al, NCB 2021
RPE1-TetOn-Cas9-PuroS-TP53-/-ERCC1-/- ('ERCC1-/-') - Apelt et al, JEM 2020
RPE1-TetOn-Cas9-PuroS-TP53-/-XPG-/- ('XPG-/-') - This study
RPE1-TetOn-Cas9-PuroS-TP53-/-CSB-/- ('CSB-/-+GFP-EV') - This study
RPE1-TetOn-Cas9-PuroS-TP53-/-CSB-/- ('CSB-/-+GFP-CSB-WT') - This study
RPE1-TetOn-Cas9-PuroS-TP53-/-CSB-/- ('CSB-/-+GFP-CSB-K538R') - This study
MRC5 - ATCC (CCL-171)
CS1AN - Kind gift from Alan Lehmann
U2OS - ATCC (HTB-96)
U2OS GFP - This study
U2OS GFP-DNMT1 - This study
HeLa WT - Kind gift from Tomoo Ogi
HeLa CSB-/- - Kind gift from Tomoo Ogi
HeLa RPB1-K1268R - Kind gift from Tomoo Ogi

Authentication

None of the cell lines have been formally authenticated.

Mycoplasma contamination

The cell lines in this study have not all been formally confirmed as mycoplasma-free.

Commonly misidentified lines
(See ICLAC register)

None of the cell lines used in this study are in the ICLAC register.

