## [Peer Review File · Nature Cell Biology]

Peer Review Information

Journal: Nature Cell Biology

Manuscript Title: Transcription-coupled repair of DNA-protein crosslinks depends on CSA and CSB

Corresponding author name(s): Dr Julian Stingele

Editorial Notes:

Reviewer Comments & Decisions:

Decision Letter, initial version:
--

*Please delete the link to your author homepage if you wish to forward this email to co-authors.

Dear Dr Stingle,

Thank you for submitting your manuscript, "Cockayne syndrome proteins CSA and CSB promote transcription-coupled repair of DNA-protein crosslinks independently of nucleotide excision repair", to Nature Cell Biology, and I am so very sorry for the long delay before communicating our decision to you.

The manuscript has now been seen by 2 referees, who are experts in DPCs (Referee #1); transcription and genome stability (Referee #2). Unfortunately, Referee #3, with expertise in NER, has had a personal emergency and it is not longer clear whether the referee will be able to share comments with us. We had been waiting for their feedback, but, given the time elapsed, we have now decided to move forward with a decision based on the comments we have received. We again apologize for the length of the process.

As you will see from their comments (attached below), the reviewers found this work of potential interest but have raised substantial concerns, which in our view would need to be addressed with considerable revisions before we can consider publication in Nature Cell Biology.

Nature Cell Biology editors discuss the referee reports in detail within the editorial team, including the chief editor, to identify key referee points that should be addressed with priority, as opposed to requests that are beyond the scope of the current study. To guide the scope of the revisions, I have listed these points below. Our standard revision period is six months, and we are committed to providing a fair and constructive peer-review process, so please feel free to contact me if you would like to discuss any of the referee comments further or if you anticipate any issues or delays addressing the reviews.

The reviewers' comments indicate to us that additional mechanistic studies are required to firm up the claims related to the new pathway, and these points need to be addressed experimentally to the best of your ability. Reconsideration of the study at the journal and re-engagement of the referees will depend on the strength of the revisions. We editorially agree with the reviewers that further mechanistic insights are warranted to delineate the transcription-coupled DPC repair pathway, as follows:

A- the reviewers both had questions about the factors involved: Rev#2 suggested further assessing CSB and CSA functions (comments pasted below), while Rev#1 asked about the possibility that SPRTN may play a role in the absence of CSB (point #3). We agree that following up on these questions is

important to define the functions of these factors and refine our understanding of the pathway.

Rev#2: "In other words, while the CSB ATPase mutant is sensitive to DPC agents, is it also repairing less DPCs in genes? Is the ATPase activity required for the transcription restart? Some characterization would help adding novelty to the manuscript. At the same time, what is the role of CSA? Data presented point to a potential role in the ubiquitilation of DPCs, and the authors suggest that these could be perhaps histones, according to some other published data from the lab. However, an effort to show that DNA protein crosslinked histones (or Pol II see later) are not repaired in CSA/CSB defective cells should be done."

B- Both reviewers questioned the link to transcription and its requirement, and also suggested deeper characterization of the DPC-seq datasets, which are important suggestions:

Rev#1 points #1-2

Rev#2: "First of all, I couldn't find how many times the experiment was performed, so some validation of the data would be required. More in the specifics, the authors show a metagene profile across all genes, showing a peak of DPCs at the TSS. Now, there are a lot of proteins at the TSS, so perhaps a WB to test whether indeed RNAPII is DPC could help supporting their findings. I guess the metagene profile is not specifically for transcribed genes, so using Pol II levels from published ChIP-Seq to identify relevant genes would be enough. This because I expect that on transcribed genes there should be a clearer difference between the WT and CSB mutant cells for the repair of DPCs. Moreover, as a control, I would like to see that the repair is CSB-dependent specifically on transcribed genes and not on not transcribed genes, also to have an idea of transcription-independent repair kinetics of DPCs. I would also look at genes of different lengths rather than only by transcription levels, as perhaps long genes (>100kB) may still have DPCs in the 3' end of the gene after the 6h, further supporting that repair follows Pol II transcription along genes. The use of the ATAC-Seq data is not very clear, is it specifically around the TSS or along the whole length of the gene? And if so, is it used as a reader of chromatin compactation? Not sure that it is sensitive enough in this sense.

However, the most striking thing was the fact that while 1154 genes show more DPCs in a CSB dependent manner, but 327 genes more in the WT. what are these 327 genes? why are they so different? What genes are these? Is it specific gene GO classes? Are these 327 genes affected in their transcription following CSB/-/? Or is it the 1154 genes that are specifically transcribed in response to damage in the absence of CSB? Basically, some characterization of the two groups of genes would be needed to determine why some genes are repaired faster without CSB (Pol II levels, ATAC-Seq, GO, gene length just to mention a few).

Finally, an attempt to define whether transcription shut down is specific to Pol II or affects other polymerases should be done, as CSA and CSB have a role in regulating also rRNA transcription, notice the difference between S4e and S4g between DPC and UV treatments. Last, some clarification of what the transcription shut down is would be helpful also to determine why CSB deficient cells are defective in the DPC repair. Following UV-induced DNA damage transcription is affected at different levels: 1) impairment of Pol II progression because of the lesions; 2) slow down of transcription elongation; 3) shut down of new initiation. All three on their own would induce a reduction in EU incorporation, and reduce reduce DNA damage repair ability, and CSB has roles both in the repair of UV-induced lesions and the transcription restart following UV damage (Proietti-De-Santis et al., 2006). Moreover, in the

Mulderrig et al., 2021 there is a clear indication in Fig 1 that formaldehyde, removes the hypophosphorylated I₁a form of Pol II, that is the initiating one, indicative of a shut down also of new gene transcription. Hence, defining better the transcription shut-down and restart can help defining and also why CSB deficient cells accumulate more DPCs and whether this is only because they repair less or whether they allow fewer Pol II back in the genes following damage and in this way sensing fewer lesions.”

C- All other referee concerns pertaining to strengthening existing data, providing controls, methodological details, clarifications and textual changes, ensuring reproducible and robust results with adequate sample size (e.g., see Rev#2's points) should also be addressed.

D- Finally, please pay close attention to our guidelines on statistical and methodological reporting (listed below) as failure to do so may delay the reconsideration of the revised manuscript. In particular, please provide:

We would be happy to consider a revised manuscript that would satisfactorily address these points, unless a similar paper is published elsewhere, or is accepted for publication in Nature Cell Biology in the meantime.

- ensure that it conforms to our format instructions and publication policies (see below and www.nature.com/nature/authors/).

- provide a point-by-point rebuttal to the full referee reports verbatim, as provided at the end of this letter.

- provide the completed Editorial Policy Checklist (found here <https://www.nature.com/authors/policies/Policy.pdf>), and Reporting Summary (found here <https://www.nature.com/authors/policies/ReportingSummary.pdf>). This is essential for reconsideration of the manuscript and these documents will be available to editors and referees in the event of peer review. For more information see <http://www.nature.com/authors/policies/availability.html> or contact me.

Nature Cell Biology is committed to improving transparency in authorship. As part of our efforts in this direction, we are now requesting that all authors identified as 'corresponding author' on published

papers create and link their Open Researcher and Contributor Identifier (ORCID) with their account on the Manuscript Tracking System (MTS), prior to acceptance. ORCID helps the scientific community achieve unambiguous attribution of all scholarly contributions. You can create and link your ORCID from the home page of the MTS by clicking on 'Modify my Springer Nature account'. For more information please visit www.springernature.com/orcid.

[Redacted]

We hope that you will find our referees' comments and editorial guidance helpful. Please do not hesitate to contact me if there is anything you would like to discuss. Thank you very much again for considering NCB for your work.

Best wishes,

Melina

Melina Casadio, PhD
Senior Editor, Nature Cell Biology
ORCID ID: <https://orcid.org/0000-0003-2389-2243>

Reviewers' Comments:

Reviewer #1:

Remarks to the Author:

Carnie and colleagues report a novel finding of a transcription-coupled (TC) repair mechanism for DNA-protein crosslinks (DPCs). Through CRISPR interference screenings, they discovered CSA and CSB as genes that protect cells from DPCs induced by formaldehyde (FA) and 5-aza-dC. In a series of gene knockout experiments, the authors identified a pathway that facilitates transcription recovery after FA exposure, which is dependent on the upstream TC-NER factors, CSB, CSA, and partially UVSSA, but not downstream factors like XPC, XPA, ERCC1, or XPG. They also developed a new technique (DPC-seq) for genome-wide mapping of DPC-containing genomic regions, which was used to demonstrate CSB-dependent DPC clearance in transcriptionally active loci. Based on these results, the authors conclude that the CSA and CSB-dependent TC-DPC repair pathway promotes transcription recovery and cell survival following DPC induction.

This study reports an important discovery regarding the novel functions of CSA and CSB in DPC repair, which distinguishes them from other downstream TC-NER factors. Genetic mutations in these two genes cause Cockayne syndrome, which has distinct features from Xeroderma pigmentosum caused by mutations in other genes in the TC-NER pathway. Therefore, this study is significant as it provides

a possible explanation for the difference. Overall, the data presented in the study are of high quality and have been interpreted properly. If the points listed below are addressed, I believe that this study is suitable for publication in Nature Cell Biology.

Major points

1. Could the response of CSB/CSA to FA be due to Pol II crosslinking to DNA rather than Pol II stalling at DPCs induced by FA?
2. Although DPC repair at transcribed genes appears to be linked to transcription, this relationship is only demonstrated through a correlation with transcriptional activity (Fig. 6a,b) and not examined directly. Would it be possible to test the requirement of transcription, for example, by using a Cdk7 inhibitor such as THZ1?
3. Fig. S9h: The impact of CSB knockout on the clearance of DPCs was surprisingly mild, even in highly transcribed genes. Could this be due to the existence of other DPC repair pathways such as SPRTN? Would it be possible to investigate this using SPRTN deltaC CSB^{-/-} cells?

Minor points

1. 5-aza-dC does not cause transcription defects (measured by EU in Fig. S4d), but RPE1 cells appear to be less sensitive to 5-aza-dC than HAP1 cells (Fig. 2e,g). Could the results be different in HAP1 cells?
2. Fig. 3a: The proposed model suggests that CSB interacts preferentially with elongating RPB1. Is the FA-induced interaction stronger with RPB1 CTD-pS2 than CTD-pS5?
3. Fig. 3e-j, Fig. S3d-g: It is not clear why the authors chose to examine RPB1 CTD-pS5 instead of pS2. Further clarification would be helpful.
4. Fig. 3e-j: Molecular weight markers for the RPB1 CTD-pS5 blots are missing.
5. Line 247, "Taken together, these data suggest that formaldehyde and 5-aza-dC-induced DPCs cause transcription stress that sets off a CSB-dependent response, which is crucial for effective recovery of RNA synthesis.": This does not seem to be an accurate summary of what is shown for 5-aza-dC, which did not have global effects on transcription rates (Fig. S4d).

Reviewer #2:

Remarks to the Author:

In the present work the authors present data about the role of CSB mainly, and of CSA, in the repair of transcription blocking DNA-protein crosslinks and the recovery of transcription shutdown induced by treatment with DPC-inducing agents. The data are very intriguing as they identify a role for CSA and CSB that is separated from the rest of the components of the transcription-coupled nucleotide excision repair pathway. Repair of these lesions in transcribed genes does not lead to nucleotide excision repair, making it a novel repair pathway rather than TC-NER dealing with another type of lesion. The role of CSA and CSB in the repair of DPC is on its own not novel, as previous papers have already presented a role for these two factors in the repair of these lesions (Mulderig et al., 2021; Burgos-Moron et al., 2018). While I praise the effort of the authors in proving that CSA and CSB play a role separate from that of other TC-NER factors and to me that makes it really interesting as mentioned above, they have not presented enough novel evidence about what mechanistically these two factors are doing. In other words, while the CSB ATPase mutant is sensitive to DPC agents, is it also repairing less DPCs in genes? Is the ATPase activity required for the transcription restart? Some characterization would help adding novelty to the manuscript. At the same time, what is the role of CSA? Data presented point to a potential role in the ubiquitilation of DPCs, and the authors suggest that these

could be perhaps histones, according to some other published data from the lab. However, an effort to show that DNA protein crosslinked histones (or Pol II see later) are not repaired in CSA/CSB defective cells should be done. This links me then to the DPC-Seq. The results shown are definitely interesting but some clarification is required. First of all, I couldn't find how many times the experiment was performed, so some validation of the data would be required. More in the specifics, the authors show a metagene profile across all genes, showing a peak of DPCs at the TSS. Now, there are a lot of proteins at the TSS, so perhaps a WB to test whether indeed RNAPII is DPC could help supporting their findings. I guess the metagene profile is not specifically for transcribed genes, so using Pol II levels from published ChIP-Seq to identify relevant genes would be enough. This because I expect that on transcribed genes there should be a clearer difference between the WT and CSB mutant cells for the repair of DPCs. Moreover, as a control, I would like to see that the repair is CSB-dependent specifically on transcribed genes and not on not transcribed genes, also to have an idea of transcription-independent repair kinetics of DPCs. I would also look at genes of different lengths rather than only by transcription levels, as perhaps long genes (>100kB) may still have DPCs in the 3' end of the gene after the 6h, further supporting that repair follows Pol II transcription along genes. The use of the ATAC-Seq data is not very clear, is it specifically around the TSS or along the whole length of the gene? And if so, is it used as a reader of chromatin compactation? Not sure that it is sensitive enough in this sense.

However, the most striking thing was the fact that while 1154 genes show more DPCs in a CSB dependent manner, but 327 genes more in the WT. what are these 327 genes? why are they so different? What genes are these? Is it specific gene GO classes? Are these 327 genes affected in their transcription following CSB-/-? Or is it the 1154 genes that are specifically transcribed in response to damage in the absence of CSB? Basically, some characterization of the two groups of genes would be needed to determine why some genes are repaired faster without CSB (Pol II levels, ATAC-Seq, GO, gene length just to mention a few).

Finally, an attempt to define whether transcription shut down is specific to Pol II or affects other polymerases should be done, as CSA and CSB have a role in regulating also rRNA transcription, notice the difference between S4e and S4g between DPC and UV treatments. Last, some clarification of what the transcription shut down is would be helpful also to determine why CSB deficient cells are defective in the DPC repair. Following UV-induced DNA damage transcription is affected at different levels: 1) impairment of Pol II progression because of the lesions; 2) slow down of transcription elongation; 3) shut down of new initiation. All three on their own would induce a reduction in EU incorporation, and reduce reduce DNA damage repair ability, and CSB has roles both in the repair of UV-induced lesions and the transcription restart following UV damage (Proietti-De-Santis et al., 2006). Moreover, in the Mulderrig et al., 2021 there is a clear indication in Fig 1 that formaldehyde, removes the hypophosphorylated IIa form of Pol II, that is the initiating one, indicative of a shut down also of new gene transcription. Hence, defining better the transcription shut-down and restart can help defining and also why CSB deficient cells accumulate more DPCs and whether this is only because they repair less or whether they allow fewer Pol II back in the genes following damage and in this way sensing fewer lesions.

Reviewer #3:

None

Methods should be written concisely, but should contain all elements necessary to allow interpretation and replication of the results. As a guideline, Methods sections typically do not exceed 3,000 words. The Methods should be divided into subsections listing reagents and techniques. When citing previous methods, accurate references should be provided and any alterations should be noted. Information must be provided about: antibody dilutions, company names, catalogue numbers and clone numbers for monoclonal antibodies; sequences of RNAi and cDNA probes/primers or company names and catalogue numbers if reagents are commercial; cell line names, sources and information on cell line identity and authentication. Animal studies and experiments involving human subjects must be reported in detail, identifying the committees approving the protocols. For studies involving human subjects/samples, a statement must be included confirming that informed consent was obtained. Statistical analyses and information on the reproducibility of experimental results should be provided in a section titled "Statistics and Reproducibility".

All Nature Cell Biology manuscripts submitted on or after March 21 2016 must include a Data availability statement at the end of the Methods section. For Springer Nature policies on data availability see <http://www.nature.com/authors/policies/availability.html>; for more information on this particular policy see <http://www.nature.com/authors/policies/data/data-availability-statements-data-citations.pdf>. The Data availability statement should include:

- Accession codes for primary datasets (generated during the study under consideration and designated as "primary accessions") and secondary datasets (published datasets reanalysed during the study under consideration, designated as "referenced accessions"). For primary accessions data should be made public to coincide with publication of the manuscript. A list of data types for which submission to community-endorsed public repositories is mandated (including sequence, structure, microarray, deep sequencing data) can be found here <http://www.nature.com/authors/policies/availability.html#data>.
- Unique identifiers (accession codes, DOIs or other unique persistent identifier) and hyperlinks for datasets deposited in an approved repository, but for which data deposition is not mandated (see here for details <http://www.nature.com/sdata/data-policies/repositories>).
- At a minimum, please include a statement confirming that all relevant data are available from the authors, and/or are included with the manuscript (e.g. as source data or supplementary information),

listing which data are included (e.g. by figure panels and data types) and mentioning any restrictions on availability.

- If a dataset has a Digital Object Identifier (DOI) as its unique identifier, we strongly encourage including this in the Reference list and citing the dataset in the Methods.

We recommend that you upload the step-by-step protocols used in this manuscript to the Protocol Exchange. More details can found at www.nature.com/protocolexchange/about.

All imaging data should be accompanied by scale bars, which should be defined in the legend. Cropped images of gels/blots are acceptable, but need to be accompanied by size markers, and to retain visible background signal within the linear range (i.e. should not be saturated). The boundaries of panels with low background have to be demarked with black lines. Splicing of panels should only be considered if unavoidable, and must be clearly marked on the figure, and noted in the legend with a statement on whether the samples were obtained and processed simultaneously. Quantitative comparisons between samples on different gels/blots are discouraged; if this is unavoidable, it should only be performed for samples derived from the same experiment with gels/blots were processed in parallel, which needs to be stated in the legend.

- We accept PowerPoint (.PPT) files if they are fully editable. However, please refrain from adding PowerPoint graphical effects to objects, as this results in them outputting poor quality raster art. Text

used for PowerPoint figures should be Helvetica (preferred) or Arial.

Unprocessed scans of all key data generated through electrophoretic separation techniques need to be presented in a supplementary figure that should be labelled and numbered as the final supplementary

figure, and should be mentioned in every relevant figure legend. This figure does not count towards the total number of figures and is the only figure that can be displayed over multiple pages, but should be provided as a single file, in PDF or TIFF format. Data in this figure can be displayed in a relatively informal style, but size markers and the figures panels corresponding to the presented data must be indicated.

The total number of Supplementary Figures (not including the “unprocessed scans” Supplementary Figure) should not exceed the number of main display items (figures and/or tables (see our Guide to Authors and March 2012 editorial <http://www.nature.com/ncb/authors/submit/index.html#suppinfo>; <http://www.nature.com/ncb/journal/v14/n3/index.html#ed>). No restrictions apply to Supplementary Tables or Videos, but we advise authors to be selective in including supplemental data.

GUIDELINES FOR EXPERIMENTAL AND STATISTICAL REPORTING

REPORTING REQUIREMENTS – To improve the quality of methods and statistics reporting in our papers we have recently revised the reporting checklist we introduced in 2013. We are now asking all life sciences authors to complete two items: an Editorial Policy Checklist (found here <https://www.nature.com/authors/policies/Policy.pdf>) that verifies compliance with all required editorial policies and a reporting summary (found here <https://www.nature.com/authors/policies/ReportingSummary.pdf>) that collects information on experimental design and reagents. These documents are available to referees to aid the evaluation of the manuscript. Please note that these forms are dynamic ‘smart pdfs’ and must therefore be downloaded and completed in Adobe Reader. We will then flatten them for ease of use by the reviewers. If you would like to reference the guidance text as you complete the template, please access these flattened versions at <http://www.nature.com/authors/policies/availability.html>.

Author Rebuttal to Initial comments

Reviewer Comments (reproduced in their entirety, our responses in green)

Reviewer 1

Remarks to the Author:

Carnie and colleagues report a novel finding of a transcription-coupled (TC) repair mechanism for DNA-protein crosslinks (DPCs). Through CRISPR interference screenings, they discovered CSA and CSB as genes that protect cells from DPCs induced by formaldehyde (FA) and 5-aza-dC. In a series of gene knockout experiments, the authors identified a pathway that facilitates transcription recovery after FA exposure, which is dependent on the upstream TC-NER factors, CSB, CSA, and partially UVSSA, but not downstream factors like XPC, XPA, ERCC1, or XPG. They also developed a new technique (DPC-seq) for genome-wide mapping of DPC-containing genomic regions, which was used to demonstrate CSB-dependent DPC clearance in transcriptionally active loci. Based on these results, the authors conclude that the CSA and CSB-dependent TC-DPC repair pathway promotes transcription recovery and cell survival following DPC induction.

This study reports an important discovery regarding the novel functions of CSA and CSB in DPC repair, which distinguishes them from other downstream TC-NER factors. Genetic mutations in these two genes cause Cockayne syndrome, which has distinct features from Xeroderma pigmentosum caused by mutations in other genes in the TC-NER pathway. Therefore, this study is significant as it provides a possible explanation for the difference. Overall, the data presented in the study are of high quality and have been interpreted properly. If the points listed below are addressed, I believe that this study is suitable for publication in Nature Cell Biology.

We thank Reviewer 1 for their supportive comments and constructive suggestions/requests.

Major points

1. Could the response of CSB/CSA to FA be due to Pol II crosslinking to DNA rather than Pol II stalling at DPCs induced by FA?

In the original manuscript, we speculated on this possibility but are now providing several lines of direct evidence indicating that RNAPII can indeed become crosslinked to DNA upon FA treatment - primarily at the transcription start site (TSS). Importantly, however, our data also indicate that the repair of RNAPII-DPCs at the TSS is CSB-independent.

To determine and monitor RNAPII-DPC formation directly, we developed a stringent CUT&Tag approach under high salt conditions and in the absence of any crosslinking/fixation step aside from our formaldehyde treatment. Using antibodies against RPB1 CTD-pS2 and -pS5, we found that formaldehyde induced clear RPB1 crosslinking at the TSS that resembled our DPC-seq metagene profiles. Importantly, upon a 6-hour release from formaldehyde, this peak returned to basal levels in a manner independent of CSB (new Extended Data Fig. 8b-c and shown below).

Extended Data Fig. 8b**Extended Data Fig. 8c**
Extended Data Fig. 8. (b-c) Metagene profiles from stringent CUT&Tag with antibodies against RPB1 CTD-pS5 (b) and -pS2 (c) in WT or *CSB*^{-/-} RPE1 cells treated with formaldehyde or treated and then released for 6 hours; n=3 biological replicates.

To confirm these findings via an orthogonal approach, we isolated DPCs from cells after formaldehyde treatment using the PxP (Purification of x-linked Proteins) assay followed by western blot against (phosphorylated) RPB1 or unbiased mass spectrometry. In agreement with our CUT&Tag data, we observed formation of crosslinks formed by RPB1 and other RNAPII subunits that were resolved over time in a *CSB*-independent manner (new Extended Data Fig. 8d, new Extended Data Fig. 7d and shown below).

Extended Data Fig. 8d**Extended Data Fig. 7d**
Extended Data Fig. 8. (d) PxP-WB with the indicated antibodies in WT and *CSB*^{-/-} RPE1 cells treated with formaldehyde and released for the indicated time points. Extended Data Fig. 7. (d) Heat map displaying Z-scored intensities for DNA-crosslinked proteins identified by PxP-MS after formaldehyde treatment and release for the indicated times in WT or *CSB*^{-/-} RPE1 cells. Intensities are displayed for three replicates per condition.

Together, these data suggest that RNAPII crosslinking upon FA treatment indeed occurs, but this seems to be restricted primarily to the TSS, where repair occurs independently of CSB. We have updated our Discussion to take into account these new data. Together with our DPC-seq data, our results demonstrate that CSB functions in the bodies of active genes, rather than acting on crosslinked RNAPII at the TSS. This conclusion is reminiscent of recent reports that in the context of UV-irradiation, RNAPII is removed and degraded from promoters in a CSB-independent manner (Steurer et al, *Nature Comms* 2022). Therefore, we conclude that most of the CSB-associated phenotypes arise from RNAPII stalling at DPCs within genes.

To understand to what degree these DPCs are formed by RNAPII itself or by other proteins such as histones, remains an interesting future question but this will likely require the development of novel experimental tools and methods. We did try to adopt our modified CUT&Tag approach to monitor histone-DPCs but were unable to find conditions sufficiently stringent to remove background (non-crosslinked) histones while also enabling subsequent antibody binding and tagmentation. At any rate, the fact that *CSB*^{-/-} cells are hypersensitive to 5-aza-dC-induced DNMT1-DPCs clearly demonstrates that CSB contributes to the repair of non-RNAPII DPCs.

2. Although DPC repair at transcribed genes appears to be linked to transcription, this relationship is only demonstrated through a correlation with transcriptional activity (Fig. 6a,b) and not examined directly. Would it be possible to test the requirement of transcription, for example, by using a Cdk7 inhibitor such as THZ1?

To address the reviewer's suggestion, we monitored the repair of formaldehyde-induced DPCs by DPC-seq in WT RPE1 cells in the presence or absence of the transcription inhibitor flavopiridol.

By using the existing RNAPII ChIP-seq data that we had used to infer transcriptional activity, we divided genes into quartiles of low-to-high RNAPII occupancy and found that flavopiridol specifically delayed the repair of DPCs in genes with high RNAPII occupancy (new Fig. 6f). We termed the subset of genes that showed substantially delayed DPC repair upon flavopiridol treatment as 'transcription-dependent' (new Fig. 6g and shown below). Finally, to determine whether CSB is specifically important for DPC repair in genes that show transcription-dependent repair, we assessed the DPC repair rate in WT and *CSB*^{-/-} cells in the sets of genes where DPC repair was perturbed by flavopiridol or not. To this end, we calculated fold-changes between FA+0h and FA+6h in the respective cell lines and gene sets (new Fig. 7f and shown below). Strikingly, *CSB*^{-/-} cells displayed a highly statistically significant defect in DPC repair but only in the transcription-dependent gene class.

Together, these data directly demonstrate the existence of transcription-coupled DPC repair.

Fig. 6f**Fig. 6g****Fig. 7f**
Figure 6. (f) Log₂ fold change of DPC-seq coverage per gene 6h/0h after FA treatment, with or without flavopiridol treatment, in genes grouped by RNAPII occupancy, statistics via paired Wilcoxon test * p<0.05, ** p<0.01, *** p<0.001. (g) DPC-seq coverage per gene 6 hours after formaldehyde treatment in WT RPE1 cells that are either not treated (NT) or treated with flavopiridol. In green, genes with significantly higher DPC-coverage in flavopiridol treated cells, indicating they undergo transcription-dependent DPC-repair. **Figure 7.** (f) Log₂ fold change of DPC-seq coverage per gene 6h/0h after FA treatment, in WT RPE1 cells vs *CSB*^{-/-} cells, in genes grouped by if they show transcription-dependent DPC-repair, statistics via paired Wilcoxon test *** p<0.001.

3. Fig. S9h: The impact of *CSB* knockout on the clearance of DPCs was surprisingly mild, even in highly transcribed genes. Could this be due to the existence of other DPC repair pathways such as *SPRTN*? Would it be possible to investigate this using *SPRTN* Δ*CSB*^{-/-} cells?

We tested whether *SPRTN* is responsible for the residual repair observed in *CSB*^{-/-} cells using the RRS (recovery of RNA synthesis) assay. However, we did not observe any additional transcription recovery defects after formaldehyde treatment in *SPRTN* Δ*CSB*^{-/-} cells compared with *CSB*^{-/-} cells (new Extended Data Fig. 7b and shown below). This was not unexpected because we performed these experiments with cells synchronised in G1-phase and *SPRTN* is primarily expressed in S-phase.

There are several possibilities that could explain the remaining clearance of DPCs in *CSB*^{-/-} cells:

- a) In addition to SPRTN-dependent proteolysis, DPCs are targeted by several other pathways that ultimately lead to replication-coupled and global-genome proteasomal degradation. Therefore, we explored whether the proteasome also contributes to DPC-clearance in genes. Indeed, using DPC-seq, we observed that proteasome inhibition caused a global delay in DPC repair in all gene bodies but most pronounced in highly accessible chromatin (new Extended Data Fig. 8i and shown below). Thus, global-genome proteasome-dependent DPC degradation may contribute to the loss of DPCs in the absence of CSB. Of note, this scenario is not mutually exclusive with our model that proteasomal degradation may also act downstream of CSB/CSA-dependent DPC ubiquitylation to enhance DPC repair in transcribed genes (as laid out in our updated Discussion).
- b) Importantly, in contrast to UV-induced damage, formaldehyde-induced DPCs are not chemically stable. Spontaneous hydrolysis contributes substantially to the loss of formaldehyde-induced DPCs in cells (Quievryn and Zhitkovich, *Carcinogenesis* 2000), which may contribute to the resolution of DPCs in transcribed genes in the absence of CSB.

In agreement with DPCs in transcribed genes being eventually resolved through spontaneous hydrolysis or a global-genome repair mechanism, transcription ultimately recovered in *CSB*^{-/-} cells after release from formaldehyde treatment, unlike after UVC (see Fig. 5a-b in the revised manuscript).

Importantly, while the DPC clearance defect observed in *CSB*^{-/-} cells is not absolute, the hypersensitivities to DPC-inducing agents of *CSB*^{-/-} cells are dramatic (highlighted by CSB and CSA scoring strongest amongst all 'DNA repair' factors in our CRISPRi screens). Thus, while CSB function may not be the most critical factor in DPC repair from a quantitative point-of-view, it is clearly required to counteract the most toxic consequences of DPC formation in human cells.

Extended Data Fig. 7b

Extended Data Fig. 8i

Fig. 5a

Fig. 5b

Extended Data Fig. 7. (b) RRS assays in WT *SPRTN-ΔC*, *CSB*^{-/-} and *SPRTN-ΔC/CSB*^{-/-} RPE1 cells treated with formaldehyde and released for the indicated times. Mean intensities of replicates are displayed as black dots with a mean of those averages shown; n=3 replicates error bars ± SEM. Extended Data Fig. 8. (i). Log₂ fold change of DPC-seq coverage per gene 6h/0h after FA treatment, with or without MG132 treatment, in genes with low, medium or high DNA accessibility as determined by ATAC-seq (GEO: GSE209659), statistics via paired Wilcoxon test *** p<0.001. Box-plot shows upper and lower quartile boundaries, a line at the median and total distribution. Figure 5. (a-b) RRS assays in WT, *CSB*^{-/-} or *CSA*^{-/-} RPE1 cells treated with formaldehyde (a) or UVC (b) and released for the indicated times.

Minor points

1. 5-aza-dC does not cause transcription defects (measured by EU in Fig. S4d), but RPE1 cells appear to be less sensitive to 5-aza-dC than HAP1 cells (Fig. 2e,g). Could the results be different in HAP1 cells?

We have tested EU incorporation in HAP1 cells upon 5-aza-dC as suggested (new Extended Data Figure 4e) but did not detect any appreciable transcription shutdown. We conclude that the transcription stress induced by 5-aza-dC is likely highly localised and is thus difficult to visualise globally by measuring EU incorporation.

Extended Data Fig. 4e

Extended Data Fig. 4. (e) RRSs assay in HAP1 cells released from double thymidine block into a 1-hour treatment with 5-aza-dC at the indicated doses.

2. Fig. 3a: The proposed model suggests that CSB interacts preferentially with elongating RPB1. Is the FA-induced interaction stronger with RPB1 CTD-pS2 than CTD-pS5?

To answer the reviewer's question, we conducted co-IP experiments with antibodies raised against RPB1 CTD-pS2 and RPB1 CTD-pS5. We found that after formaldehyde treatment CSB and CSA co-immunoprecipitated with RPB1 using both antibodies (see Fig. R1, below). This is not entirely unexpected, because although RPB1 CTD-pS2 is most associated with elongating RNAPII, RPB1 CTD-pS5 phosphorylation (while strongly enriched at the TSS) also occurs throughout gene bodies as part of elongating complexes and particularly at splice sites (Nojima et al, *Cell* 2015). Indeed, RPB1 CTD-pS2 and -pS5 are not mutually exclusive and one RPB1 CTD can bear pS2 and pS5 modifications. As such, IPs conducted using an antibody against pS5 would also pull down elongating RNAPII complexes. We have not included these new data in the revised manuscript but have added clarification to the Results section.

Figure R1.

Figure R1. Western blot using the indicated antibodies following immunoprecipitation of endogenous, chromatin-bound RPB1 CTD-pS2 and -pS5 from WT and CSB^{-/-} RPE1 cells after UVC or formaldehyde treatments, as indicated.

3. Fig. 3e-j, Fig. S3d-g: It is not clear why the authors chose to examine RPB1 CTD-pS5 instead of pS2. Further clarification would be helpful.

The antibody used for these experiments was raised against RPB1 CTD-pS5 but is often used by others in the field as a pan-RPB1 antibody (such as in Tufegzic Vidakovic et al, *Cell* 2020). In the revised manuscript, we assessed RPB1 stability in CSB^{-/-} and CSA^{-/-} cells also using an antibody against RPB1 CTD-pS2 and found no overt differences (new Extended Data Fig. 3j-m and shown below).

Extended Data Fig. 3. (j-k) RPB1 degradation in cycloheximide-treated WT and CSB^{-/-} (j) or CSA^{-/-} (k) cells treated with formaldehyde for the indicated time points. For (j) and (k), GAPDH blot images are also shown alongside blots in Figures 3g and 3h, respectively, due to detection of RPB1 CTD-pS2 and RPB1 CTD-pS5 from the same gel. (l-m) Quantification of (j) and (k), respectively; error bars \pm SEM, n=3 replicates.

4. Fig. 3e-j: Molecular weight markers for the RPB1 CTD-pS5 blots are missing.

This omission has been rectified.

5. Line 247, "Taken together, these data suggest that formaldehyde and 5-aza-dC-induced DPCs cause transcription stress that sets off a CSB-dependent response, which is crucial for effective recovery of RNA synthesis.": This does not seem to be an accurate summary of what is shown for 5-aza-dC, which did not have global effects on transcription rates (Fig. S4d).

We agree and reworded the sentence that now reads "Taken together, these data suggest that DPCs cause transcription stress that sets off a CSB-dependent response, which is crucial for effective recovery of RNA synthesis."

Reviewer 2 (our numbering of comments added)

Remarks to the Author:

In the present work the authors present data about the role of CSB mainly, and of CSA, in the repair of transcription blocking DNA-protein crosslinks and the recovery of transcription shutdown induced by treatment with DPC-inducing agents. The data are very intriguing as they identify a role for CSA and CSB that is separated from the rest of the components of the transcription-coupled nucleotide excision repair pathway. Repair of these lesions in transcribed genes does not lead to nucleotide excision repair, making it a novel repair pathway rather than TC-NER dealing with another type of lesion. The role of CSA and CSB in the repair of DPC is on its own not novel, as previous papers have already presented a role for these two factors in the repair of these lesions (Mulderigg et al., 2021; Burgos-Moron et al., 2018). While I praise the effort of the authors in proving that CSA and CSB play a role separate from that of other TC-NER factors and to me that makes it really interesting as mentioned above,

they have not presented enough novel evidence about what mechanistically these two factors are doing.

We thank Reviewer 2 for their supportive comments and constructive suggestions/requests.

1. In other words, while the CSB ATPase mutant is sensitive to DPC agents, is it also repairing less DPCs in genes? Is the ATPase activity required for the transcription restart? Some characterization would help adding novelty to the manuscript.

As suggested by the reviewer, we have determined the recovery of transcription after formaldehyde treatment in *CSB*^{-/-} RPE1 cells complemented by either GFP-CSB^{WT} or catalytically inactive GFP-CSB^{K538R}. We note that while GFP-CSB^{K538R} expressed to a lower level than GFP-CSB^{WT}, its expression was still at least as much as endogenous CSB (new Extended Data Fig. 5c and shown below). Cells expressing GFP-CSB^{K538R} displayed a partial transcription recovery defect relative to the WT protein (new Extended Data Fig. 5d and shown below). The partial functionality of CSB^{K538R} in these assays agrees with the partial complementation of *CSB*^{-/-} cells in our survival assays by this variant (Fig. 1g-i). We thus conclude that CSB's ATPase activity contributes to CSB's function in DPC repair, but that CSB has, in addition, a non-catalytic function in this process.

Extended Data Fig. 5c

Extended Data Fig. 5d

Extended Data Fig. 5c-d. (c) Western blot analysis of CSB and GFP-CSB expression in the indicated RPE1 cell lines. (d) RRS assays in cell lines from (c) treated with formaldehyde and released for the indicated times. Mean intensities of replicates are displayed as black dots with a mean of those averages shown; n=3 replicates, error bars \pm SEM.

2. At the same time, what is the role of CSA? Data presented point to a potential role in the ubiquitilation of DPCs, and the authors suggest that these could be perhaps histones, according to some other published data from the lab. However, an effort to show that DNA protein crosslinked histones (or Pol II see later) are not repaired in CSA/CSB defective cells should be done.

Assessing CSA-mediated DPC ubiquitylation and turnover is technically challenging because all DPCs are also targeted by competing ubiquitylation pathways, e.g. by the E3 ligases RNF4, RFWD3, and TRAIIP. This is highlighted by our new PXP-MS analysis – the first proteomic analysis of the repair of formaldehyde-induced DPCs – that compares the resolution of DPCs induced by formaldehyde over 6 hours in WT and *CSB*^{-/-} cells (see response to Reviewer 1, point 1, and new Extended Data Fig. 7d). These data show that CSB loss does not affect the repair of any DPC on a global level. Hence, the contribution of transcription-coupled repair can only be visualised by resolving DPC repair genome-wide, as done by us using DPC-Seq. Therefore, we expanded the use of DPC-Seq in the revised manuscript to strengthen several aspects of our study (as laid out in our response to the reviewer's additional points below).

Nonetheless, to address the reviewer's suggestion to better define the role of CSA and ubiquitylation in transcription-coupled DPC repair, we conducted additional experiments. We determined whether CSA's function in DPC repair can be entirely explained by its established role in RNAPII ubiquitylation. ELOF1 promotes CSA-dependent ubiquitylation of RNAPII at lysine K1268. Therefore, we tested transcription recovery and sensitivity phenotypes of both *ELOF1*^{-/-} RPE1 cells and HeLa cells with the endogenous *POLR2A* gene mutated to give rise to RPB1-K1268R. Both cell lines exhibited only partial sensitivity phenotypes and transcription recovery defects after formaldehyde compared with *CSB*^{-/-} cells (new Fig. 5c-d, new Extended Data Fig. 5g-h,j-k; data relating to RPB1-K1268R HeLa cells are shown below).

These observations, coupled with the mild defect in formaldehyde-induced RPB1 ubiquitylation seen in *CSB*^{-/-} cells in the original manuscript (Fig. 3c in the revised manuscript and shown below), led us to conclude that CSA's role in transcription-coupled DPC repair is not limited to RPB1 ubiquitylation. It therefore appears most likely that CSA ubiquitylates transcription-blocking DPCs. However, directly testing this idea will likely require the *in vitro* reconstitution of transcription-coupled DPC repair – an important future goal that lies beyond the scope of this study.

Fig. 5d

Extended Data Fig. 5j

Extended Data Fig. 5k

Fig. 3c

Fig. 5. (d) RRS assays with WT, *CSB*^{-/-} and RPB1-K1268R HeLa cells treated with formaldehyde and released for the indicated times. **Extended Data Fig. 5.** (j-k) Alamar blue viability assays with WT, *CSB*^{-/-} and RPB1-K1268R HeLa cells treated with formaldehyde (j) or 5-aza-dC (k); error bars ± SD, n=3 replicates. **Fig. 3c.** (c) Dsk2 pulldown in WT and *CSB*^{-/-} RPE1 cells synchronized in G1 by serum starvation and treated with 250μM FA for the indicated times.

3. This links me then to the DPC-Seq. The results shown are definitely interesting but some clarification is required. First of all, I couldn't find how many times the experiment was performed, so some validation of the data would be required. More in the specifics, the authors show a metagene profile across all genes, showing a peak of DPCs at the TSS. Now, there

are a lot of proteins at the TSS, so perhaps a WB to test whether indeed RNAPII is DPC could help supporting their findings.

All DPC-seq experiments are displayed as averages of 3 independent biological replicates. Each biological replicate consists of three technical replicates through the DPC purification stage, which were then pooled at equimolar ratios prior to NGS library preparation – we now provide this information in the figure legends and Methods section of our manuscript. Of note, we have found through additional DPC-seq experiments that the DPC-seq peak at the TSS is highly sensitive to the extent of DNA shearing that occurs through the manual handling and sonication steps of DPC-seq sample preparation (new Extended Data Fig. 8g-h and shown below), likely due to the relatively nucleosome-free nature of DNA at the TSS.

To address the reviewer's suggestion to directly test for the presence of RNAPII-DPCs, we provide several lines of new evidence in the revised manuscript (as also outlined in detail in response to Reviewer 1, point 1). In brief, we confirmed the presence of RNAPII-DPCs at the TSS by using an orthogonal approach. We used a modified, stringent form of CUT&Tag, that does not use sonication to fragment DNA. Our protocol employs high salt washes and no fixation aside from the formaldehyde treatment itself to remove as much non-crosslinked protein as possible from chromatin and thus enrich for DPCs. Consistent with RNAPII-DPC formation at the TSS, we observed dramatic enrichment of RNAPII CTD-pS5-DPCs and -pS2-DPCs at the TSS after formaldehyde treatment (new Extended Data Fig. 8e-f and shown below). DPC formation by RNAPII subunits was further confirmed by PxB-MS and PxB-WB analysis (response to Reviewer 1, point 1). Overall, we thus have clear evidence that RNAPII itself becomes crosslinked at the TSS after formaldehyde treatment.

Extended Data Fig. 8h.

Extended Data Fig. 8e.

Extended Data Fig. 8f.

Extended Data Fig. 8. (h) Metagene profiles of DPC-seq after formaldehyde treatment in RPE1 cells from samples in subjected to low or high shearing conditions; n=3 biological replicates. (e-f) Overlaid metagene profiles from formaldehyde-treated WT RPE1 cells subjected to DPC-seq or stringent CUT&Tag against RPB1 CTD-pS5 (e) or -pS2 (f); n=3 biological replicates for each approach.

4. I guess the metagene profile is not specifically for transcribed genes, so using Pol II levels from published ChIP-Seq to identify relevant genes would be enough. This because I expect that on transcribed genes there should be a clearer difference between the WT and CSB mutant cells for the repair of DPCs. Moreover, as a control, I would like to see that the repair

is CSB-dependent specifically on transcribed genes and not on not transcribed genes, also to have an idea of transcription-independent repair kinetics of DPCs.

As the reviewer points out, CSB's role in DPC repair should be restricted to transcriptionally active loci according to our model. To test the transcription-dependency of DPC repair and CSB's role therein more directly than by inferring transcriptional activity based on RNAPII occupancy, we performed DPC-seq experiments in the presence of flavopiridol, which enabled several important analyses:

- a. Validating the use of RNAPII occupancy as a surrogate for transcriptional activity using this assay, we found that flavopiridol specifically impaired DPC repair in transcriptionally active genes, as inferred from RNAPII occupancy. The effect of flavopiridol on DPC repair was most dramatic in the highest quartile of RNAPII occupancy (new Fig. 6f and shown below).
- b. We identified a large number of genes whose DPC repair was slowed upon treatment with flavopiridol throughout the recovery period. We classify these genes as 'transcription-dependent' (new Fig. 6g and shown below).
- c. Analysis of feature enrichment in genes that are repaired in a transcription-dependent and/or CSB-dependent manner highlighted strong similarities, most dramatically chromatin accessibility (ATAC) and RNAPII occupancy (new Fig. 7d and shown below).
- d. We assessed DPC repair rates in *CSB*^{-/-} cells compared to WT cells specifically in genes that showed delayed DPC repair in the presence of flavopiridol (and those genes who did not). We found that CSB loss had a dramatic effect on DPC repair specifically in genes whose DPC repair was also transcription-dependent (new Fig. 7f and shown below).

Figure 6. (f) Log₂ fold change of DPC-seq coverage per gene 6h/0h after FA treatment, with or without flavopiridol treatment, in genes grouped by RNA-Pol II occupancy, statistics via paired Wilcoxon test * p < 0.05, ** p < 0.01, *** p < 0.001. (g) DPC-seq coverage per gene 6 hours after formaldehyde treatment in WT RPE1 cells that are either not treated (NT) or treated with flavopiridol. In green, genes with significantly higher DPC-coverage in flavopiridol treated cells, indicating they undergo transcription-dependent DPC-repair. **Figure 7.** (d) The level of different genomic features in CSB- and transcription-dependent genes relative to the non-changing group. (f) Log₂ fold change of DPC-seq coverage per gene 6h/0h after FA treatment, in WT RPE1 cells vs CSB^{-/-} cells, in genes grouped by if they show transcription-dependent DPC-repair, statistics via paired Wilcoxon test *** p < 0.001. For all DPC-seq analyses, n=3 biological replicates.

Together, our new DPC-seq experiments examining the effect of transcription inhibition on DPC repair confirm CSB's role in a transcription-coupled DPC repair pathway.

5. I would also look at genes of different lengths rather than only by transcription levels, as perhaps long genes (>100kB) may still have DPCs in the 3' end of the gene after the 6h, further supporting that repair follows Pol II transcription along genes.

Our DPC-seq analyses suggest that there is a relationship between gene length and transcription/CSB-dependent DPC repair. Our feature enrichment analysis highlighted a mild enrichment for longer genes in CSB/transcription-dependent gene sets (new Fig. 7d and Extended Data Fig. 9f in the revised manuscript; Fig. 7d is shown in the response above and Extended Data Fig. 9f is shown below). However, since the original submission we have performed nascent RNA-seq (TTchem-seq) studies, the data from which we elaborate on below in response to point #9, but in this experiment we observed impaired transcription recovery in CSB^{-/-} cells after formaldehyde treatment across irrespective of gene length.

Overall, we think that the relationship between gene length and CSB-dependency for DPC repair suggested by our DPC-seq data is mild, and likely arises alongside transcriptional activity, since long genes tend to be more transcriptionally active.

6. The use of the ATAC-Seq data is not very clear, is it specifically around the TSS or along the whole length of the gene? And if so, is it used as a reader of chromatin compactation? Not sure that it is sensitive enough in this sense.

ATAC-seq was calculated across the entire length of the gene - as for DPC-seq coverage – with the purpose of determining how chromatin compaction impacts on repair, rather than specifically to look at the TSS, which we have now addressed using CUT&Tag as mentioned above.

Using the ATAC-seq data, we were able to draw several informative conclusions:

- a) Inaccessible chromatin experiences the most dramatic induction of DPCs, but repair in such contexts occurs across compaction states (Fig. 6d in the revised manuscript and shown below).
- b) The proteasome promotes DPC repair irrespective of chromatin compaction state (new Extended Data Fig. 8i and shown below).

c) Transcription-coupled DPC repair appears to operate primarily in highly accessible, RNAPII-occupied chromatin, but RNAPII occupancy is a better predictor of transcription- or CSB-dependent DPC repair (new Fig. 6i, new Fig. 7d-e and shown below; Fig. 7d shown in response to Reviewer 2 point 4).

Fig. 6d

Extended Data Fig. 8i

Fig. 6i

Fig. 7e

Fig. 6. (d) DPC-seq coverage per gene, with or without 6h recovery after treatment, in genes with low, medium or high DNA accessibility as determined by ATAC-seq (GEO: GSE209659), statistics via paired Wilcoxon test *** $p < 0.001$. Box-plot shows upper and lower quartile boundaries, a line at the median and total distribution. **Extended Data Fig. 8.** (i) Log_2 fold change of DPC-seq coverage per gene 6h/0h after FA treatment, with or without MG132 treatment, in genes with low, medium or high DNA accessibility as determined by ATAC-seq (GEO: GSE209659), $n=3$ biological replicates, statistics via paired Wilcoxon test *** $p < 0.001$. Box-plot shows upper and lower quartile boundaries, a line at the median and total distribution. **Fig. 6.** (i) Per gene RNAPII occupancy vs DNA accessibility, as determined via ATAC-seq in RPE1 cells (GEO: GSE209659) but only showing genes that show transcription-dependent DPC repair and a group of matched size that do not show transcription-dependent repair, also with the percentage of each group that are present in the shown quadrants. **Fig. 7.** (e) as for Fig. 6i but showing genes that exhibit CSB-dependent DPC repair. For all DPC-seq analyses, $n=3$ biological replicates.

7. However, the most striking thing was the fact that while 1154 genes show more DPCs in a CSB dependent manner, but 327 genes more in the WT. what are these 327 genes? why are they so different? What genes are these? Is it specific gene GO classes? Are these 327 genes affected in their transcription following CSB-/-? Or is it the 1154 genes that are specifically transcribed in response to damage in the absence of CSB? Basically, some characterization

of the two groups of genes would be needed to determine why some genes are repaired faster without CSB (Pol II levels, ATAC-Seq, GO, gene length just to mention a few).

As suggested by the reviewer, we have explored the CSB-dependent vs. -independent genes and performed several analyses:

- Feature enrichment analysis of CSB-dependent vs. independent genes, which shows that CSB-dependent genes tend to be more RNAPII-occupied, reside in more accessible chromatin environments (ATAC), longer, and be protein-coding (Fig. 7d and Extended Data Fig. 9c-f in the revised manuscript).
- As suggested, we have compared our CSB-dependent/-independent gene lists to existing data identifying transcripts upregulated in WT and *CSB*^{-/-} cells after UV treatment (TTchem-seq from Tufegdzic Vidaković et al, *Cell* 2020). This comparison highlighted no enrichment for CSB-dependence or -independence in differentially expressed transcripts after UV in WT or *CSB*^{-/-} cells (shown in Figure R2).

Fig. 7d.

Extended Data Fig. 9

Extended Data Fig. 9

Fig. R2.

Fig. 7. (d) The level of different genomic features in CSB- and transcription-dependent genes relative to the non-changing group. **Extended Data Fig. 9.** (c) Percentage of different gene types in genes that are dependent on CSB for DPC repair or where repair is not significantly changing based on CSB status.

(d) RNA-Pol II occupancy of genes that are dependent on CSB for DPC repair or where repair is not significantly changing based on CSB status, statistics via unpaired Wilcoxon test *** $p < 0.001$. (e) Same as (d) but for DNA accessibility. (f) Same as (d) but for gene length. Fig. R2. Gene counts of upregulated, downregulated or unaffected genes after UV treatment in WT or *CSB(ERCC6)*^{-/-} cells from published TTchem-seq data (Tufegdžić Vidaković et al, *Cell* 2020), with CSB-dependence and -independence as defined by DPC-seq highlighted.

Overall, our analyses have given further insight into the nature of genes whose repair is CSB-dependent, including their transcription-dependence based on the flavopiridol DPC-seq experiments already discussed. Of the 327 genes whose DPC repair was statistically significantly faster in *CSB*^{-/-} cells, we found no apparent biological explanation for these through our analyses and these gene set was not significantly enriched for any GO terms. We conclude that these genes are likely noise arising from the sequencing approach. Since NGS analysis is only qualitative in an internally relative/comparative sense, meaning that if a set of genes' read counts drops significantly, increased relative reads will be detected elsewhere. To reflect this, we have removed references to these CSB-independent genes in our figures.

We consider the most significant findings in relation to CSB-dependent genes to be their association with transcriptional activity and dependence on transcription for their DPC repair, underlining CSB as a critical factor mediating transcription-coupled DPC repair.

8. Finally, an attempt to define whether transcription shut down is specific to Pol II or affects other polymerases should be done, as CSA and CSB have a role in regulating also rRNA transcription, notice the difference between S4e and S4g between DPC and UV treatments.

As recommended by the reviewer, we assessed the impact of formaldehyde and CSB on RNAPII transcription. Using the RRS assay and immunostaining for the nucleolar marker NPM1, we quantified relative EU signal intensity across the nucleus and segmented into nucleolar and nucleoplasmic regions. We found that, as for RNAPII transcription, CSB loss compromised the recovery of rRNA synthesis in the nucleolus and therefore RNAPII transcription (new Figure 4a-d in the revised manuscript and shown below).

Regarding RNAPIII transcription, our DPC-seq datasets suggest that CSB does not significantly affect the DPC repair of tRNA genes, but we note that these analyses are challenging to the relatively unclear annotations of tRNA genes.

Fig. 4.

Fig. 4. (a-c) Quantification of Recovery of RNA Synthesis (RRS) assays paired with NPM1 staining to enable stratified quantification of relative EU intensity in nuclear (a), nucleolar (b) and nucleoplasmic (c) regions in WT or *CSB*^{-/-} cells treated with formaldehyde and released into fresh medium for the indicated time points; error bars \pm SEM $n=3$ replicates. (d) Representative images from RRS assays in (a-c).

While the mechanistic basis of CSB's role in RNAPII transcription recovery after formaldehyde treatment is unclear, this area represents an interesting area for future work that could have important implications for the understanding of Cockayne Syndrome.

9. Last, some clarification of what the transcription shut down is would be helpful also to determine why CSB deficient cells are defective in the DPC repair. Following UV-induced DNA damage transcription is affected at different levels: 1) impairment of Pol II progression because of the lesions; 2) slow down of transcription elongation; 3) shut down of new initiation. All three on their own would induce a reduction in EU incorporation, and reduce DNA damage repair ability, and CSB has roles both in the repair of UV-induced lesions and the transcription restart following UV damage (Proietti-De-Santis et al., 2006). Moreover, in the Mulderig et al., 2021 there is a clear indication in Fig 1 that formaldehyde, removes the hypophosphorylated IIa form of Pol II, that is the initiating one, indicative of a shut down also of new gene transcription. Hence, defining better the transcription shut-down and restart can help defining and also why CSB deficient cells accumulate more DPCs and whether this is only because they repair less or whether they allow fewer Pol II back in the genes following damage and in this way sensing fewer lesions.

The nature of transcriptional shutdown and recovery poses an interesting and challenging question, and we note that the basis of this is complex and not completely understood even in the well-studied context of UV-induced DNA lesions. However, we agree with the reviewer that this is an important area to explore and the majority of Figure 4 in the revised manuscript aims at addressing this question.

We took several approaches to examine the sources of transcription shutdown in the context of DPC induction, focussing mainly on the distinction between perturbed elongation and the inhibition of initiation.

- a) First, we used an RT-qPCR approach with normalisation to a spiked-in chicken transcript to examine the impact of formaldehyde on mRNA production. We performed this assay against six unstable transcripts in WT and *CSB*^{-/-} cells and found that one hour of formaldehyde treatment caused a dramatic shutdown in mRNA production relative to untreated controls, suggesting that active elongation is impaired. Consistent with a role for CSB in responding to DPC-blocked RNAPII, *CSB*^{-/-} cells exhibited delayed recovery of production of all these transcripts (new Fig. 4e and shown below).
- b) As suggested, we examined the loss and recovery of initiating, hypophosphorylated RPB1 in WT and *CSB*^{-/-} cells subjected to formaldehyde treatment. As shown in Mulderrig et al, *Nature* 2021, initiating RPB1 was depleted relative to elongating RPB1 after 3 hours of treatment in WT cells and recovered to equal levels by 6-8 hours. In contrast, *CSB*^{-/-} cells took longer for initiating RPB1 to recover relative to elongating RPB1 but did recover eventually (new Fig. 4f and shown below). This suggested that *CSB*^{-/-} cells might suffer from impaired new initiation after DPC induction. To test this further, we examined the induction and degradation of the inhibitor of initiation ATF3 and found that formaldehyde induced ATF3 expression that persisted in *CSB*^{-/-} cells compared to WT cells (new Fig. 4g and shown below). These data suggested that prolonged formaldehyde can indeed perturb transcription initiation, particularly in *CSB*^{-/-} cells.
- c) To better understand the basis of formaldehyde-induced transcription shutdown in an unbiased manner, we performed TTchem-seq to analyse the synthesis of nascent transcripts. Immediately after formaldehyde treatment, nascent transcript production disappeared from gene bodies and shifted to the TSS, consistent with DPC-blocked RNAPII in gene bodies failing to transcribe, rather than an acute shutdown of initiation. As expected, following release from formaldehyde, *CSB*^{-/-} cells displayed impaired recovery of transcription using this method (new Fig. 4i-j and shown below).

Fig. 4.

Fig. 4. (e) RT-qPCR for the indicated targets normalised to a chicken spike-in mRNA in WT or CSB^{-/-} cells treated with formaldehyde and released into fresh medium for 6 hours. Values were normalised to transcript levels in untreated conditions; error bars \pm SEM, n=4 replicates. * p<0.05 based on multiple Mann-Whitney tests. (f) Western blot analysis of hyperphosphorylated (elongating) and hypophosphorylated (initiating) RPB1, denoted by * and #, respectively, upon formaldehyde treatment for the indicated time points. (g) Western blot analysis of ATF3 induction and degradation in WT or CSB^{-/-} RPE1 cells upon treatment with formaldehyde for the indicated times, or with UVC and allowed to recover for 24 hours. (h) Nascent RNA-seq heatmaps of 3-100kb genes in WT and CSB^{-/-} cells treated with formaldehyde and released for 9 hours. (i) Example genome browser plot from (h) of the *MAP3K14* locus. (j) Metagene profiles from (h) of genes 25-50kb, 50-100kb and >100kb in length, shown in full (left panels), +/- 5kb around the TSS (centre panels) or beginning from 5kb into the gene body (right panels).

We thank the reviewer for their suggestion to examine the mode of transcription shutdown after formaldehyde treatment. Our TTchem-seq data demonstrate that the acute transcription shutdown upon formaldehyde treatment, seen across several orthogonal approaches, arises likely not from acute initiation defects but from RNAPII stalling at DPCs in gene bodies. Indeed, this complements data in the manuscript demonstrating the appearance of detectable RPB1

polyubiquitylation within 15 minutes of formaldehyde treatment (Fig. 3c). That prolonged formaldehyde exposure likely causes transcription initiation defects that are eventually reversible in *CSB*^{-/-} cells is an interesting discovery that is reminiscent of similar findings in the context of UV-induced transcription shutdown and is worthy of further investigation beyond the scope of this manuscript.

Decision Letter, first revision:

Our ref: NCB-A51027A

19th January 2024

Dear Dr. Stingele,

Thank you for submitting your revised manuscript "Cockayne syndrome proteins CSA and CSB promote transcription-coupled repair of DNA-protein crosslinks independently of nucleotide excision repair" (NCB-A51027A). It has now been seen by the original referees and their comments are below. The reviewers find that the paper has improved in revision, and therefore we'll be happy in principle to publish it in Nature Cell Biology, pending minor revisions to satisfy the referees' final requests and to comply with our editorial and formatting guidelines.

We are now performing detailed checks on your paper and will send you a checklist detailing our editorial and formatting requirements in about ~2 weeks. Please do not upload the final materials and make any revisions until you receive this additional information from us.

Thank you again for your interest in Nature Cell Biology Please do not hesitate to contact me if you have any questions.

Sincerely,

Melina

Melina Casadio, PhD
Senior Editor, Nature Cell Biology
ORCID ID: <https://orcid.org/0000-0003-2389-2243>

Reviewer #1 (Remarks to the Author):

The authors have addressed all the points I raised during the initial review. The newly added data indicate that crosslinking of RNAPII by FA indeed occurs but is repaired in a CSB-independent manner. This is an important distinction from the CSB-dependent DPC repair mechanism, which responds to RNAPII stalling at DPCs. The authors have also provided additional evidence supporting the dependency of transcription-coupled DPC repair on transcription using transcription inhibitor flavopiridol. Further analyses of DPC-seq data have demonstrated that CSB knockout impacts DPC repair specifically in genes where DPC repair is transcription-dependent, thereby reinforcing the proposed model of CSB-dependent transcription-coupled DPC repair. With these and other improvements, I believe that the manuscript is now suitable for publication in Nature Cell Biology.

Reviewer #2 (Remarks to the Author):

The authors have provided a greatly improved manuscript with additional data to support their conclusions, and I am happy to support the publication of the manuscript on Nature Cell Biology as it is.

Decision Letter, final checks:

Our ref: NCB-A51027A

31st January 2024

Dear Dr. Stingele,

Thank you for your patience as we've prepared the guidelines for final submission of your Nature Cell Biology manuscript, "Cockayne syndrome proteins CSA and CSB promote transcription-coupled repair of DNA-protein crosslinks independently of nucleotide excision repair" (NCB-A51027A). Please carefully follow the step-by-step instructions provided in the attached file, and add a response in each row of the table to indicate the changes that you have made. Please also check and comment on any additional marked-up edits we have proposed within the text. Ensuring that each point is addressed will help to ensure that your revised manuscript can be swiftly handed over to our production team.

In recognition of the time and expertise our reviewers provide to Nature Cell Biology's editorial process, we would like to formally acknowledge their contribution to the external peer review of your manuscript entitled "Cockayne syndrome proteins CSA and CSB promote transcription-coupled repair of DNA-protein crosslinks independently of nucleotide excision repair". For those reviewers who give their assent, we will be publishing their names alongside the published article.

Nature Cell Biology offers a Transparent Peer Review option for new original research manuscripts submitted after December 1st, 2019. As part of this initiative, we encourage our authors to support increased transparency into the peer review process by agreeing to have the reviewer comments, author rebuttal letters, and editorial decision letters published as a Supplementary item. When you submit your final files please clearly state in your cover letter whether or not you would like to participate in this initiative. Please note that failure to state your preference will result in delays in accepting your manuscript for publication.

Cover suggestions

COVER ARTWORK: We welcome submissions of artwork for consideration for our cover. For more information, please see our guide for cover artwork.

Nature Cell Biology has now transitioned to a unified Rights Collection system which will allow our Author Services team to quickly and easily collect the rights and permissions required to publish your work. Approximately 10 days after your paper is formally accepted, you will receive an email in providing you with a link to complete the grant of rights. If your paper is eligible for Open Access, our Author Services team will also be in touch regarding any additional information that may be required to arrange payment for your article.

Please note that *Nature Cell Biology* is a Transformative Journal (TJ). Authors may publish their research with us through the traditional subscription access route or make their paper immediately open access through payment of an article-processing charge (APC). Authors will not be required to make a final decision about access to their article until it has been accepted. Find out more about Transformative Journals

Please use the following link for uploading these materials:
[Redacted]

Best regards,

Jonathon Comfort
Staff
Nature Cell Biology

On behalf of

Melina Casadio, PhD
Senior Editor, Nature Cell Biology
ORCID ID: <https://orcid.org/0000-0003-2389-2243>

Reviewer #1:

Remarks to the Author:

The authors have addressed all the points I raised during the initial review. The newly added data indicate that crosslinking of RNAPII by FA indeed occurs but is repaired in a CSB-independent manner. This is an important distinction from the CSB-dependent DPC repair mechanism, which responds to RNAPII stalling at DPCs. The authors have also provided additional evidence supporting the dependency of transcription-coupled DPC repair on transcription using transcription inhibitor flavopiridol. Further analyses of DPC-seq data have demonstrated that CSB knockout impacts DPC repair specifically in genes where DPC repair is transcription-dependent, thereby reinforcing the proposed model of CSB-dependent transcription-coupled DPC repair. With these and other improvements, I believe that the manuscript is now suitable for publication in Nature Cell Biology.

Reviewer #2:

Remarks to the Author:

The authors have provided a greatly improved manuscript with additional data to support their conclusions, and I am happy to support the publication of the manuscript on Nature Cell Biology as it is.

Final Decision Letter:

Dear Dr Stinglele,

I am pleased to inform you that your manuscript, "Transcription-coupled repair of DNA-protein crosslinks depends on CSA and CSB", has now been accepted for publication in Nature Cell Biology. Congratulations on this beautiful study!

Please note that *Nature Cell Biology* is a Transformative Journal (TJ). Authors may publish their research with us through the traditional subscription access route or make their paper immediately open access through payment of an article-processing charge (APC). Authors will not be required to make a final decision about access to their article until it has been accepted. Find out more about Transformative Journals

If you have not already done so, we strongly recommend that you upload the step-by-step protocols used in this manuscript to the Protocol Exchange (www.nature.com/protocolexchange), an open online resource established by Nature Protocols that allows researchers to share their detailed experimental know-how. All uploaded protocols are made freely available, assigned DOIs for ease of citation and are fully searchable through nature.com. Protocols and Nature Portfolio journal papers in which they are used can be linked to one another, and this link is clearly and prominently visible in the online versions of both papers. Authors who performed the specific experiments can act as primary authors for the Protocol as they will be best placed to share the methodology details, but the Corresponding Author of the present research paper should be included as one of the authors. By uploading your Protocols to Protocol Exchange, you are enabling researchers to more readily reproduce or adapt the methodology you use, as well as increasing the visibility of your protocols and papers. You can also establish a dedicated page to collect your lab Protocols. Further information can be found at www.nature.com/protocolexchange/about

With kind regards,

Melina

Melina Casadio, PhD
Senior Editor, Nature Cell Biology
ORCID ID: <https://orcid.org/0000-0003-2389-2243>
